# Noise-Robust Density Estimation for Tabular Data Anomaly Detection

**Dazhi Fu** [1]  **Zhao Zhang** [2]  **Jicong Fan** [1]

## Abstract

Density-based anomaly detection methods often provide accurate and interpretable predictions but their performance can be severely degraded by inherent noise in the data, such as changes arising from environmental conditions during data collection or background noise. To deal with such noise, we present noise-robust density estimation (NRDE) for tabular data anomaly detection. We aim to estimate the density of pure data with the influence of noise isolated, which is a non-trivial task since the data-generating process is completely unknown. Specifically, NRDE learns a Jacobian-regularized normalizing flow to estimate the sources of data and categorizes sources into two groups, where one group generates pure data and the other generates noise. After generating pure data, we can use the density of such pure data to detect anomalies caused solely by the sources of pure data. Therefore, NRDE is robust to inherent noise. We provide theoretical results to support the effectiveness of NRDE and compare NRDE with 17 baselines on 47 benchmark datasets under different settings, including vanilla anomaly detection, anomaly detection with anomaly contamination, anomaly detection on noisy data, and transductive outlier detection. Our code is available at https://github.com/fudazhiaka/Noise-Robust-Density-Estimation.

## 1. Introduction

In an increasingly data-driven world, the problem of identifying unusual patterns or deviations from expected behavior—known as anomaly detection—has become paramount across diverse domains. Anomaly detection (Chandola et al., 2009; Pang et al., 2021; Ruff et al., 2021), sometimes also referred to as novelty or outlier detection (Breunig et al., 2000; Pimentel et al., 2014; Fu & Fan, 2026), involves the identification of data points, events, or observations that significantly differ from the majority of the data. These anomalies can signal critical incidents such as fraud (Ahmed et al., 2016), security breaches (Breier & Branišová, 2017), system failures (Du et al., 2017), or novel insights, making their accurate detection essential for timely intervention and decision-making.

In the past few years, many deep learning-based anomaly detection methods have been proposed (Cai & Fan, 2022; Shenkar & Wolf, 2022; Xu et al., 2023; Fu et al., 2024; Yin et al., 2024; Xiao et al., 2025a; Dai et al., 2025; Xiao et al., 2025b). For instance, DeepSVDD (Ruff et al., 2018b) assumes that representations of normal data can be enclosed within a small hypersphere and representations of anomalous data lie outside the hypersphere, where the representations are given by a neural network. ADERH (Durani et al., 2026) constructs ensembles of hyperspheres from randomly paired samples to identify rare points. Although these methods often demonstrate impressive performance in various scenarios, several of them require making assumptions on the structure or distribution of normal and anomalous data, which may not hold or are difficult to guarantee by the training process. For instance, Zhang et al. (2024) analyzed the limitations of the hypersphere assumption in high-dimensional spaces and proposed to project normal data into the region bounded by two hyperspheres. Moreover, some of these methods are proposed to solve the one-class classification problem (Schölkopf et al., 2001), which relies on the assumption that training data originates from a single class or exhibits a single manifold structure. Consequently, these methods can be ineffective when training data encompasses multiple clusters or lies on multiple disconnected manifolds (Khayatkhoei et al., 2018).

It should be noted that density-based methods make no assumptions about the shape or distribution of the data and are capable of modeling complex data structures. This flexibility allows them to be effective even when training data encompasses multiple classes. Traditional density-based methods include Kernel Density Estimation (KDE) (Parzen, 1962), Gaussian Mixture Models (GMM), etc. These methods often suffer from the curse of dimensionality and are not

---

[1]School of Data Science, The Chinese University of Hong Kong, Shenzhen, China [2]School of Computer Science and Information Engineering, Hefei University of Technology, Hefei, China. Correspondence to: Jicong Fan <fanjicong@cuhk.edu.cn>.

*Proceedings of the 43rd International Conference on Machine Learning*, Seoul, South Korea. PMLR 306, 2026. Copyright 2026 by the author(s).

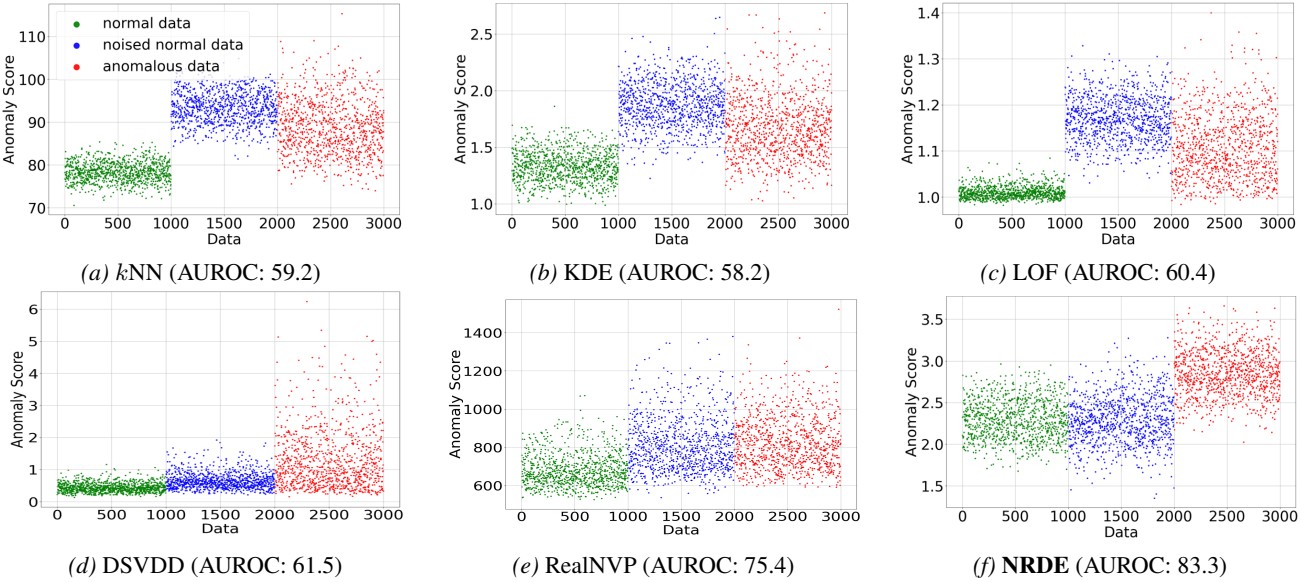

*Figure 1.* Detection performance on a synthetic dataset. A larger anomaly score indicates that the sample is more likely to be anomalous. The data were generated from a few data sources and many noise sources. Points marked in green, blue, and red represent **normal data**, **noisy normal data** (caused by noise change), and **anomalous data**, respectively. See (8) for definitions. The five compared methods detect most of the noisy normal samples as anomalies, while our NRDE is robust to changes in noise. More details about this experiment are in Appendix F.

effective on complex data. To address the problem, several deep learning based density estimation methods have been proposed. For instance, DAGMM (Zong et al., 2018) combines deep auto-encoders with GMM, utilizing the output density given by GMM in a low-dimensional space to detect anomalies. Normalizing flow (Kobyzev et al., 2020), an effective generative model, is also effective in estimating the density of complex data, and hence is useful for anomaly detection. Some flow-based image anomaly detection methods (Gudovskiy et al., 2022) first employ feature extractors to derive semantic representations and then implement normalizing flow to detect anomalies. In this work, we focus on tabular data since data of other types can be converted into tabular formats using feature extractors.

For standard anomaly detection, density-based methods, including normalizing flow and other shallow and deep models, are sensitive to changes in inherent noise in the data, yielding high false-positive rates. It is noteworthy that such inherent noise can be quite different from artificial noise like Gaussian noise, since it could represent minor changes in equipment or the environment during data collection. More specifically, real data contains inherent noise and can be described as being generated by $\mathbf{x} = G([\mathbf{s}_D; \mathbf{s}_N])$, where $\mathbf{s}_D$ and $\mathbf{s}_N$ denote the pure data source and noise source, respectively, and $G$ is the generating function. The changes of $\mathbf{x}$ caused by $\mathbf{s}_N$ should not be treated as anomalies, or at least should be distinguished from the anomalies of interest, and we call such data noisy normal data for convenience. For instance, in a vehicle monitoring system,

changes in background noise may alter the observed data, but we are only concerned with the status of the vehicle itself. Similarly, in medical diagnosis, we hope that changes in instruments and equipment or the occasional noise do not affect the diagnostic results for diseases. In Figure 1, we use a synthetic dataset to show the influence of inherent noise on the performance of five anomaly detection methods and our proposed method **N**oise-**R**obust **D**ensity **E**stimation (NRDE). We observe that the five methods fail to distinguish between noisy normal data and real anomalies, exhibiting high false-positive rates and low AUROC values, whereas NRDE is robust to changes in the inherent noise in the data and performs the best.

NRDE trains a neural network to estimate the density of pure data with the influence of noise isolated. Specifically, we propose a Jacobian-regularized normalizing flow that explicitly separates latent sources into data and noise components, allowing us to isolate noise and estimate the density of pure data. As a result, we can detect anomalies that are caused by data sources without being affected by the noise. The framework of NRDE is shown in Figure 2. Our contributions are summarized as follows:

- We propose a novel density-based AD method, NRDE, for tabular data based on a Jacobian-regularized normalizing flow.
- NRDE separates the sources into pure-data sources and noise sources and performs density estimation for the pure data only, making it robust to changes in noise.

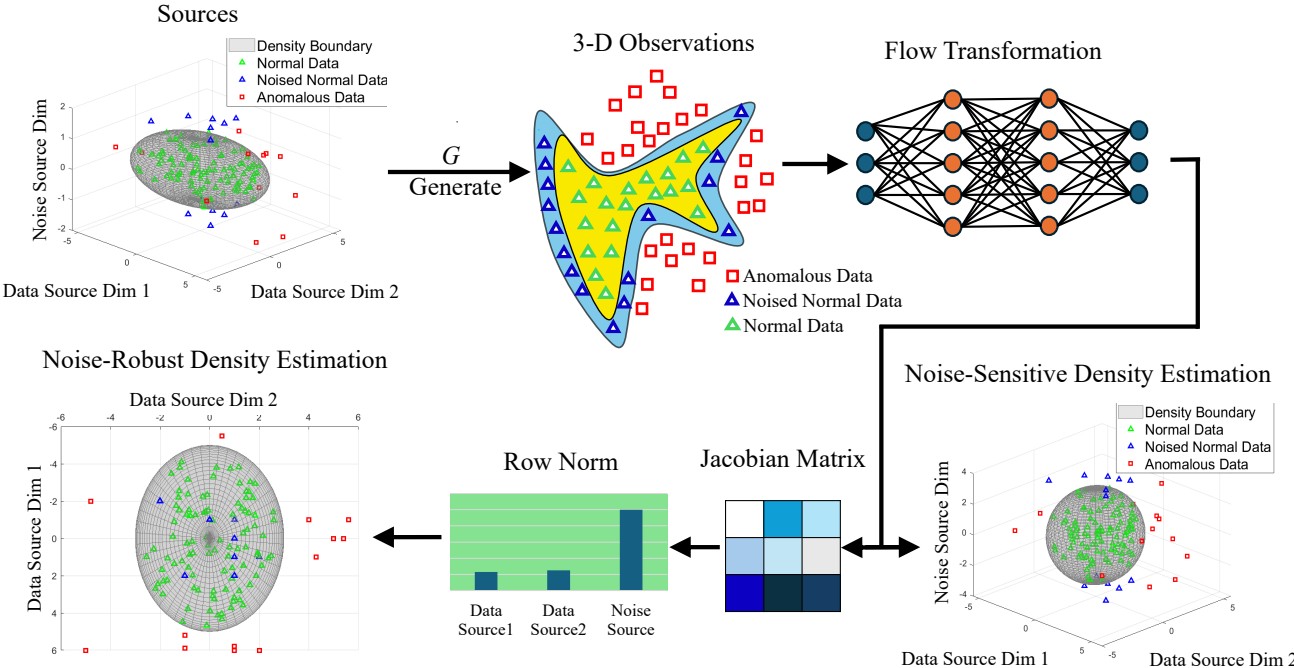

*Figure 2.* Architecture of the proposed NRDE. NRDE estimates the density of pure data by utilizing a normalizing flow with Jacobian regularization, where the influence of noise sources is isolated. Therefore, NRDE is robust to changes in inherent noise in the data.

- We provide some theoretical analysis for NRDE to support its effectiveness.
- We conduct experiments on 47 datasets to compare NRDE against 17 baseline methods. While primary evaluation is performed under the standard anomaly detection setting, our experimental setup also includes anomaly detection on noisy data, anomaly detection with contaminated data, and outlier detection.

## 2. Related Work and Preliminary Knowledge

### 2.1. Generative Models for Anomaly Detection

Deep generative models (Schlegl et al., 2019; Xia et al., 2022; Liu et al., 2025) are useful in anomaly detection due to their ability to model complex data structures. For instance, RobustRealNVP (Liu et al., 2022) ignores low-density points that are likely to be anomalies, by discarding the gradient produced by these points in the training stage, and therefore obtains a robust density function. In (Rozner et al., 2023), the authors found that densities around normal samples are relatively stable and proposed to use an autoregressive probabilistic model to maximize the density of training samples while minimizing their density variance. DTE (Livernoche et al., 2023) estimates the distribution over diffusion time for a given input and uses the mode or mean of this distribution as the anomaly score. TCCM (Li et al., 2025) employs a one-step flow-matching model to learn velocity fields between the normal data distribution and a simple reference distribution, and uses the discrep-

ancy between the predicted and true velocity vectors as the anomaly score. Unfortunately, these works do not address the problem of sensitivity to changes in inherent noise in standard anomaly detection shown by Figure 1.

### 2.2. ICA and Normalizing Flow

Independent Component Analysis (ICA) (Hyvärinen & Oja, 2000) assumes that observed data is generated by an unknown mixing process of several independent components (sources) drawn from simple distributions, and tries to obtain these components. By categorizing the mixing process, we can divide ICA methods into linear ICA and nonlinear ICA. Linear ICA assumes that the mixing process is linear and the sources are non-Gaussian, and often solves the problem by maximizing the non-Gaussianity. As for nonlinear ICA, the mixing process is assumed to be nonlinear, and the main problem faced by the field is that the model is unidentifiable or the sources are inseparable. In other words, there are infinitely many ways to transform the data into independent components, which may still be a mixture of underlying sources. By utilizing additional structure in the data or introducing auxiliary variables, many methods (Hyvärinen & Pajunen, 1999; Hyvarinen & Morioka, 2016; Zheng et al., 2022) have been developed. It is noteworthy that prior studies on ICA (Jung et al., 2000; Salimi-Khorshidi et al., 2014; Pruim et al., 2015) have shown that source components are not equally informative: some components correspond to meaningful signal sources, while others mainly capture

noise, nuisance, or background factors.

Here, we briefly review the foundational concept of normalizing flows, which is closely related to ICA. Given a set of observations, each of which, denoted as $\mathbf{x}$, is drawn from a complex distribution $\mathcal{X}$ in $\mathbb{R}^d$, normalizing flow aims to learn a function $F_{\mathcal{W}} : \mathbb{R}^d \to \mathbb{R}^d$ composed of a sequence of invertible mappings $\{f_{\mathcal{W}_t}\}_{t=1}^T$, i.e., $F_{\mathcal{W}} = f_{\mathcal{W}_T} \circ \cdots \circ f_{\mathcal{W}_2} \circ f_{\mathcal{W}_1}$, that transforms the complex distribution $\mathcal{X}$ into a simpler one, denoted as $\mathcal{Z}$, such as a standard Gaussian $\mathcal{N}(\mathbf{0}, \mathbf{I})$. Here, $T$ is the number of mappings and $\mathcal{W} = \{\mathcal{W}_1, \ldots, \mathcal{W}_T\}$ denotes the set of all neural network parameters. Because $F_{\mathcal{W}}$ is invertible, the density of $\mathbf{x}$, namely $p_{\mathcal{X}}(\mathbf{x})$, can be computed using the change-of-variables formula:

$$p_{\mathcal{X}}(\mathbf{x}) = p_{\mathcal{Z}}\big(F_{\mathcal{W}}(\mathbf{x})\big)|\det\big(\nabla_{\mathbf{x}} F_{\mathcal{W}}(\mathbf{x})\big)|, \qquad (1)$$

where $\det\big(\nabla_{\mathbf{x}} F_{\mathcal{W}}(\mathbf{x})\big)$ is the determinant of the Jacobian matrix of $F_{\mathcal{W}}$ evaluated at $\mathbf{x}$. One example of a coupling normalizing flow is RealNVP, proposed by (Dinh et al., 2016), where $f_{\mathcal{W}_i}$ is called the coupling transformation. Denoting by $\mathbf{x}^{(i)} \in \mathbb{R}^d$ the input of $f_{\mathcal{W}_i}$, $\mathbf{x}^{(i)}$ is usually split into two parts, i.e., $\mathbf{x}_\alpha^{(i)} = [x_{\alpha_1}^{(i)}, x_{\alpha_2}^{(i)}, \ldots, x_{\alpha_{q_i}}^{(i)}]^\top$ and $\mathbf{x}_\beta^{(i)} = [x_{\beta_1}^{(i)}, x_{\beta_2}^{(i)}, \ldots, x_{\beta_{d-q_i}}^{(i)}]^\top$, where $1 < q_i < d$. Then the output $\mathbf{y}^{(i)}$ of $f_{\mathcal{W}_i}$ is given as

$$\mathbf{y}_\alpha^{(i)} = \mathbf{x}_\alpha^{(i)}, \quad \mathbf{y}_\beta^{(i)} = \mathbf{x}_\beta^{(i)} \odot \exp(h_{i1}(\mathbf{x}_\alpha^{(i)})) + h_{i2}(\mathbf{x}_\alpha^{(i)}), \tag{2}$$

where $h_{i1} : \mathbb{R}^{q_i} \to \mathbb{R}^{d-q_i}$ and $h_{i2} : \mathbb{R}^{q_i} \to \mathbb{R}^{d-q_i}$ are two multilayer neural networks.

## 3. Proposed Method

### 3.1. Formulation of Noise-Robust Anomaly Detection

Let $\mathcal{D} = \{\mathbf{x}^{(1)}, \mathbf{x}^{(2)}, \ldots, \mathbf{x}^{(n)}\}$ be a set of $d$-dimensional training data drawn from an unknown distribution $\mathcal{X}$. The primary goal of anomaly detection (AD) is to learn a model $\Phi : \mathbb{R}^d \to \mathbb{R}$ from the training set $\mathcal{D}$, which can quantify the degree of anomaly or the dissimilarity of a new sample $\mathbf{x}_{\text{new}}$ relative to the distribution $\mathcal{X}$.

From a generative perspective, as described in ICA (Hyvärinen & Oja, 2000), an observation $\mathbf{x}$ can be regarded as being generated by applying an unknown invertible linear or nonlinear transformation, denoted as $G : \mathbb{R}^d \to \mathbb{R}^d$, to some unknown source $\mathbf{s} \in \mathbb{R}^d$, i.e.,

$$\mathbf{x} = G(\mathbf{s}), \tag{3}$$

where $\mathbf{s} \sim \mathcal{S}$. As mentioned in other ICA and normalizing flow techniques, the source distribution $\mathcal{S}$ is usually simple and each dimension of $\mathcal{S}$ is independent. For instance[1],

consider $\mathcal{S} = \mathcal{N}(\boldsymbol{\mu}, \boldsymbol{\Sigma})$, where $\boldsymbol{\mu} = [\mu_1, \mu_2, \ldots, \mu_d]^\top$, $\boldsymbol{\Sigma} = \text{diag}(\sigma_1^2, \sigma_2^2, \ldots, \sigma_d^2)$, and $\sigma_1 \geq \sigma_2 \geq \cdots \geq \sigma_d$. For convenience, we consider that the primary distinction among these sources resides in their variances, leading to the specification $\mathcal{S} = \mathcal{N}(\mathbf{0}, \boldsymbol{\Sigma})$. Based on $G$, the ideal normalizing flow $F_{\mathcal{W}}^*$ can be formulated as:

$$F_{\mathcal{W}}^*(\mathbf{x}) := \boldsymbol{\Sigma}^{-\frac{1}{2}} G^{-1}(\mathbf{x}), \tag{4}$$

where $\mathbf{z} = F_{\mathcal{W}}^*(\mathbf{x}) = \boldsymbol{\Sigma}^{-\frac{1}{2}}\mathbf{s} \sim \mathcal{N}(\mathbf{0}, \mathbf{I})$.

Inspired by prior studies (Salimi-Khorshidi et al., 2014; Pruim et al., 2015), we split the source $\mathbf{s}$ into two distinct parts:

$$\mathbf{s} = [\mathbf{s}_D; \mathbf{s}_N], \tag{5}$$

where $\mathbf{s}_D \in \mathbb{R}^m$ denotes the pure data (or signal) source and $\mathbf{s}_N \in \mathbb{R}^{d-m}$ denotes the noise source. It is natural to assume that the variances of $\mathbf{s}_D$ are much greater than those of $\mathbf{s}_N$, namely,

$$\sigma_1 \geq \sigma_2 \cdots \geq \sigma_m > c\sigma_{m+1} \geq c\sigma_{m+2} \cdots \geq c\sigma_d, \tag{6}$$

where $c$ is some constant much greater than 1. The data with noise removed, i.e., the pure data, is

$$\mathbf{x}_{\text{pure}} = G([\mathbf{s}_D; \mathbf{0}]). \tag{7}$$

Thus, the inherent noise in the data is $\boldsymbol{\epsilon} := \mathbf{x} - \mathbf{x}_{\text{pure}}$. Letting $\mathcal{T}$ denote the signal source distribution deemed as normal, we have the following categorization for the data:

$$
\begin{aligned}
\text{pure normal data}: \ & \mathbf{x}_{\text{pure}} = G([\mathbf{s}_D; \mathbf{0}]), \ \mathbf{s}_D \sim \mathcal{T}, \\
\text{noisy normal data}: \ & \mathbf{x}_{\text{norm}} = G([\mathbf{s}_D; \mathbf{s}_N]), \ \mathbf{s}_D \sim \mathcal{T}, \\
\text{anomalous data}: \ & \mathbf{x}_{\text{anom}} = G([\hat{\mathbf{s}}_D; \tilde{\mathbf{s}}_N]), \ \hat{\mathbf{s}}_D \not\sim \mathcal{T},
\end{aligned} \tag{8}
$$

where $\mathbf{s}_N \neq \mathbf{0}$ and $\tilde{\mathbf{s}}_N$ is arbitrary. In this work, given the observation $\mathbf{x}$, the task of noise-robust anomaly detection is to recover $\mathbf{x}_{\text{pure}}$ and evaluate whether $\mathbf{x}_{\text{pure}}$ is normal or anomalous, which is determined by $\mathbf{s}_D$ only and is irrelevant to $\mathbf{s}_N$.

Rationale for the assumption in (6): Prior studies typically rely on external structural information, metadata, and labeled data to train classifiers for identifying data sources. In contrast, we focus solely on the signal itself. This assumption is reasonable because a meaningful signal, by definition, should contain structured information and variation that differentiates it from the background. Noise, often arising from random and uncorrelated processes, tends to have its energy dispersed without a dominant structure. Therefore, the variance of the signal, which captures its total power and variability, is expected to be higher than that of the noise. This

---

[1]Although the standard linear ICA requires an assumption that the sources are non-Gaussian, the Gaussian assumption in this work makes sense because $G$ may first convert each source to non-Gaussian and then perform mixing.

is a common and often necessary condition for the signal to be detectable and analyzable amid random fluctuations. For instance, in machine learning and statistics, PCA (Jolliffe & Cadima, 2016) assumes the most important data patterns are the directions with the highest variance, effectively treating them as the "signal" and discarding low-variance "noise." In signal processing, denoising filters work by removing low-power (low-variance) frequencies assumed to be noise, while preserving high-power (high-variance) frequencies considered to be the signal. More experimental results to verify our assumption and motivation are provided in Appendix H. For instance, the performance of baselines on pure data is provided in Table 13.

### 3.2. Signal and Noise Isolation

To realize the aforementioned noise-robust anomaly detection, we need to calculate $p_{\mathcal{X}}(\mathbf{x}_{\text{pure}})$ or $p_{\bar{\mathcal{X}}}(\mathbf{x}_{\text{pure}})$, where $\mathcal{X}$ denotes the distribution of $\mathbf{x}$ and $\bar{\mathcal{X}}$ denotes the distribution of $\mathbf{x}_{\text{pure}}$ defined on the $m$-dimensional manifold embedded in $\mathbb{R}^d$. When $p_{\mathcal{X}}(\mathbf{x}_{\text{pure}})$ or $p_{\bar{\mathcal{X}}}(\mathbf{x}_{\text{pure}})$ is smaller, $\mathbf{x}_{\text{pure}}$, as well as the corresponding noisy counterpart $\mathbf{x}$, is more likely to be anomalous.

Let $F_{\mathcal{W}}$ be the flow model learned from $\mathcal{D}$ and suppose $\mathbf{x}_{\text{pure}}$ can be identified from $\mathbf{x}$, we can obtain

$$p_{\mathcal{X}}\left(\mathbf{x}_{\text{pure}}\right) = p_{\mathcal{Z}}\left(F_{\mathcal{W}}(\mathbf{x}_{\text{pure}})\right)\left|\det\left(\nabla_{\mathbf{x}_{\text{pure}}}F_{\mathcal{W}}(\mathbf{x}_{\text{pure}})\right)\right|. \tag{9}$$

Using (4), we have the ideal case for $p_{\mathcal{X}}(\mathbf{x}_{\text{pure}})$, i.e.,

$$\begin{aligned}
&\log p_{\mathcal{X}}^*(\mathbf{x}_{\text{pure}})\\
&= \log p_{\mathcal{Z}}\left(F_{\mathcal{W}}^*(\mathbf{x}_{\text{pure}})\right) + \log\left|\det\left(\nabla_{\mathbf{x}_{\text{pure}}}F_{\mathcal{W}}^*(\mathbf{x}_{\text{pure}})\right)\right|\\
&= \log\left|\det\left(\nabla_{\mathbf{x}_{\text{pure}}}F_{\mathcal{W}}^*(\mathbf{x}_{\text{pure}})\right)\right| - \sum_{i=1}^{m}\frac{s_i^2}{2\sigma_i^2} - \frac{d}{2}\log(2\pi),
\end{aligned} \tag{10}$$

where we have used the fact that $F_{\mathcal{W}}^*(\mathbf{x}_{\text{pure}}) = [\mathbf{z}_D; \mathbf{0}]$ and $\mathbf{z} = \mathbf{\Sigma}^{-\frac{1}{2}}\mathbf{s}$.

Similarly, for $p_{\bar{\mathcal{X}}}(\mathbf{x}_{\text{pure}})$, we have

$$p_{\bar{\mathcal{X}}}^*\left(\mathbf{x}_{\text{pure}}\right) = \frac{\mathcal{N}(\mathbf{z}_D; \mathbf{0}, \mathbf{I}_m)}{\sqrt{\det\left[J_{F_{\mathcal{W}}^{*-1}}(\mathbf{z}_D)^\top J_{F_{\mathcal{W}}^{*-1}}(\mathbf{z}_D)\right]}}, \tag{11}$$

where $J_{F_{\mathcal{W}}^{*-1}}(\mathbf{z}_D)$ denotes $\nabla_{\mathbf{z}_D}F_{\mathcal{W}}^{*-1}([\mathbf{z}_D; \mathbf{0}])$. The detailed derivation is provided in Appendix A.3.

The challenge is that we may never obtain $F_{\mathcal{W}}^*$. The learned $F_{\mathcal{W}}$ from $\mathcal{D}$ can only ensure that $\mathbf{z} = F_{\mathcal{W}}(\mathbf{x}) \sim \mathcal{N}(\mathbf{0}, \mathbf{I})$. It is difficult to determine which of $z_1, \ldots, z_d$ correspond to $\mathbf{s}_D$ and which of $z_1, \ldots, z_d$ correspond to $\mathbf{s}_N$. Moreover, the number of data sources $m$ is unknown and is not easy to estimate. In the following, we show how to address these problems.

Note that (4) indicates that

$$\frac{\partial z_j}{\partial \mathbf{x}} = \sigma_j^{-1} \times \frac{\partial G_j^{-1}(\mathbf{x})}{\partial \mathbf{x}}. \tag{12}$$

We assume that

$$\gamma - \delta \leq \left\|\frac{\partial G_j^{-1}(\mathbf{x})}{\partial \mathbf{x}}\right\| \leq \gamma + \delta, \quad \forall j \in [d], \tag{13}$$

where $\gamma$ and $\delta$ are some positive constants and $\delta \ll \gamma$. This assumption is reasonable because $G$ usually mixes the sources randomly and uniformly. Moreover, this formulation generalizes the widely adopted assumption in linear ICA (Hyvärinen et al., 2001), which assumes $\mathbf{W}^\top\mathbf{W} = \mathbf{I}$ in $G(\mathbf{s}) = \mathbf{W}\mathbf{s}$, corresponding to the special case $\gamma = 1$ and $\delta = 0$. Combining (12) and (13), we have

$$(\gamma - \delta)\left\|\frac{\partial z_j}{\partial \mathbf{x}}\right\|^{-1} \leq \sigma_j \leq (\gamma + \delta)\left\|\frac{\partial z_j}{\partial \mathbf{x}}\right\|^{-1}. \tag{14}$$

If $(\gamma - \delta)\left\|\frac{\partial z_j}{\partial \mathbf{x}}\right\|^{-1} > c(\gamma + \delta)\left\|\frac{\partial z_{j'}}{\partial \mathbf{x}}\right\|^{-1}$ or $\left\|\frac{\partial z_{j'}}{\partial \mathbf{x}}\right\| > c\frac{\gamma+\delta}{\gamma-\delta}\left\|\frac{\partial z_j}{\partial \mathbf{x}}\right\|$ equivalently, then $\sigma_j > c\sigma_{j'}$. This indicates that we may compare $\left\|\frac{\partial z_1}{\partial \mathbf{x}}\right\|, \ldots, \left\|\frac{\partial z_d}{\partial \mathbf{x}}\right\|$ to distinguish between $\mathbf{s}_D$ and $\mathbf{s}_N$. However, a clear gap may not exist between $\left\|\frac{\partial z_1}{\partial \mathbf{x}}\right\|, \ldots, \left\|\frac{\partial z_d}{\partial \mathbf{x}}\right\|$. An intuitive example is shown in Figure 3. The reason is that the source $\mathbf{s}$ in (3) is not identifiable and there are many equivalent problems (Hyvärinen & Pajunen, 1999; Hyvarinen et al., 2019; Zheng et al., 2022). For instance, let $\mathbf{R}$ be an orthonormal matrix and suppose $F_{\mathcal{W}}(\mathbf{x}) = \mathbf{R}F_{\mathcal{W}}^*(\mathbf{x})$ is a normalizing flow learned from $\mathcal{D}$. In this case, $F_{\mathcal{W}}(\mathbf{x}) \sim \mathcal{N}(\mathbf{0}, \mathbf{I})$ and the estimated density remains unchanged. However, $F_{\mathcal{W}}(\mathbf{x})$ becomes a combination of $\mathbf{z}$, and the row norms of the Jacobian matrix do not explicitly reflect the variances of sources.

However, we can exploit the prior knowledge in (6) to train $F_{\mathcal{W}}$ close to $F_{\mathcal{W}}^*$ and consider the following optimization

$$\begin{aligned}
&\underset{\mathcal{W}, A, B}{\text{maximize}} \sum_{\mathbf{x} \in \mathcal{D}} \log\left(p_{\mathcal{Z}}(F_{\mathcal{W}}(\mathbf{x}))|\det(\nabla_{\mathbf{x}}F_{\mathcal{W}}(\mathbf{x}))|\right)\\
&\text{subject to } \min_{j \in A}\left\|\frac{\partial z_j}{\partial \mathbf{x}}\right\|^{-1} > c'\max_{j \in B}\left\|\frac{\partial z_j}{\partial \mathbf{x}}\right\|^{-1}, \forall \mathbf{x} \in \mathcal{D}\\
&\qquad A \cup B = [d], \ A \cap B = \emptyset, \ |A| = m
\end{aligned} \tag{15}$$

where $c' = c\frac{\gamma+\delta}{\gamma-\delta}$ and $A$ corresponds to $\mathbf{s}_D$ and $B$ corresponds to $\mathbf{s}_N$. It is very difficult to solve (15) because $c, \gamma, \delta$ are unknown and the constraints are related to every $\mathbf{x}$ and the min and max operations. We also need to know $m$.

### 3.3. Jacobian-Regularized Normalizing Flow

The constraints in (15) indicate that some rows of the Jacobian matrix $\nabla_{\mathbf{x}}F_{\mathcal{W}}(\mathbf{x})$ have much smaller norms than other

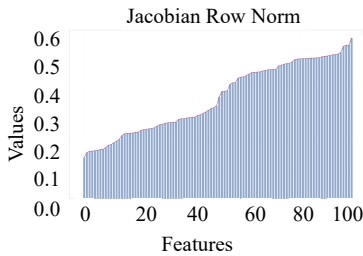 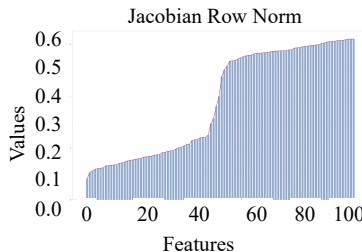

*Figure 3.* Visualization of row norms of the Jacobian matrix on a synthetic dataset with 50 pure data sources and 50 noise sources. The left panel shows the unregularized case, while the right panel shows the regularized case. More visualization results on real datasets are shown in Appendix I.1.

rows, which is a kind of sparsity. Therefore, we propose to regularize $\nabla_{\mathbf{x}} F_{\mathcal{W}}(\mathbf{x})$ during the optimization of $F_{\mathcal{W}}$ and hence solve

$$
\begin{aligned}
\operatorname*{minimize}_{\mathcal{W}} \quad & \frac{1}{n} \sum_{\mathbf{x} \in \mathcal{D}} - \log\left( p_{\mathcal{Z}}\left(F_{\mathcal{W}}(\mathbf{x})\right) | \det\left(\nabla_{\mathbf{x}} F_{\mathcal{W}}(\mathbf{x})\right)|\right) \\
& + \lambda \mathcal{R}\left(\frac{1}{n} \sum_{\mathbf{x} \in \mathcal{D}} |\nabla_{\mathbf{x}} F_{\mathcal{W}}(\mathbf{x})|\right),
\end{aligned}
\tag{16}
$$

where $\mathcal{R}$ denotes a sparse regularizer on matrices and $\lambda > 0$ is a hyperparameter. Instead of imposing regularization on each individual $\mathbf{x} \in \mathcal{D}$, we regularize the average absolute Jacobian, since the categories of sources should remain consistent across $\mathcal{D}$. We use the following $\mathcal{R}$:

$$
\mathcal{R}(\mathbf{Q}) = \sum_{i=1}^{d} \sqrt{\|\mathbf{q}_{i:}\|_1}, \tag{17}
$$

where $\mathbf{q}_{i:}$ denotes the $i$th row of $\mathbf{Q} \in \mathbb{R}^{d \times d}$. Note that $\mathcal{R}^2(\mathbf{Q})$ is the $\ell_{1,1/2}$ quasi-norm, which is sharper than the $\ell_{2,1}$ norm widely used in sparse optimization. Figure 3 illustrates the effect of $\mathcal{R}$. More details about $\mathcal{R}$ are provided in Appendix C.

An alternative to (17) is using $\mathbf{R}(\mathbf{Q}) = \sum_{j \in [B]} \|\mathbf{q}_{j:}\| - \sum_{j \in [A]} \|\mathbf{q}_{j:}\|$, where $A$ is the index set of the $m$ rows of $\mathbf{Q}$ with smaller norms and $B$ is the index set of the $d - m$ rows of $\mathbf{Q}$ with larger norms determined in each iteration. This method requires a good estimate of $m$ and is sensitive to the initialization. The performance is not as good as (17).

Although solving (16) makes sense, to obtain the pure data $\mathbf{x}_{\text{pure}}$, we will have to estimate $m$. Here we provide a simple way to determine $\hat{m}$, an approximation of $m$ as follows: we first compute the average absolute Jacobian

$$
J = \frac{1}{n} \sum_{\mathbf{x} \in \mathcal{D}} |\nabla F_{\mathcal{W}}(\mathbf{x})|. \tag{18}
$$

We then take the row norms $\{\|J_i\|\}_{i=1}^{d}$ and sort them in ascending order, denoted by $\{\|J_{\hat{i}}\|\}_{i=1}^{d}$. Then we measure

the variance gap by computing $\Delta_i = \frac{J_{(i\hat{+}1)} - J_{\hat{i}}}{J_{(i\hat{+}1)}}$, and find $m$ by

$$
\hat{m} = \arg\max_i \Delta_i, \tag{19}
$$

and

$$
\|J_i\| \approx \frac{\gamma + \delta}{\sigma_i}. \tag{20}
$$

Note that $\|J_i\|$ is smaller for sources with larger variance $\sigma_i$, which is more likely to be a data source. After determining $\hat{m}$, we can select $\hat{m}$ sources with the smallest row norms as $\mathbf{s}_{\hat{D}}$ or $\mathbf{z}_{\hat{D}}$ and generate the pure data $\hat{\mathbf{x}}_{\text{pure}}$ by:

$$
\hat{\mathbf{x}}_{\text{pure}} = F_{\mathcal{W}}^{-1}([\mathbf{z}_{\hat{D}}; \mathbf{0}]). \tag{21}
$$

Now, one can use the density of pure data $p_{\mathcal{X}}(\hat{\mathbf{x}}_{\text{pure}})$ defined in (9) or $p_{\bar{\mathcal{X}}}(\hat{\mathbf{x}}_{\text{pure}})$ defined in (11) as its anomaly score. For clarity, we refer to the two methods as NRDE and NRDE-M (manifold), respectively, throughout the experiments. More details about $p_{\bar{\mathcal{X}}}(\mathbf{x}_{\text{pure}})$ is provided in Appendix A.3.

In summary, we train a Jacobian-regularized normalizing flow via (16). After the model is well-trained, we can determine the signal sources using (19). Then, for any test data $\mathbf{x}^{\text{new}}$, we can generate its pure data $\hat{\mathbf{x}}_{\text{pure}}^{\text{new}}$ using (21) and then use the density of pure data $p_{\mathcal{X}}(\hat{\mathbf{x}}_{\text{pure}}^{\text{new}})$ or $p_{\bar{\mathcal{X}}}(\hat{\mathbf{x}}_{\text{pure}}^{\text{new}})$ as the anomaly score to determine whether $\mathbf{x}_{\text{new}}$ is anomalous or not. More details about the algorithm of NRDE are shown in Appendix F.

### 3.4. Theoretical Guarantees

We provide the following theoretical guarantee for $p_{\mathcal{X}}(\hat{\mathbf{x}}_{\text{pure}})$ to approximate $p_{\mathcal{X}}(\mathbf{x}_{\text{pure}})$ and detect anomalies successfully.

**Theorem 3.1.** *Let $F_{\mathcal{W}} = f_{\mathcal{W}_T} \circ \cdots \circ f_{\mathcal{W}_2} \circ f_{\mathcal{W}_1}$ denote an ideal sequence of coupling normalizing flows, where the Lipschitz constant of each $f_{\mathcal{W}_i}$ is bounded above by $\tau_i^+$ and below by $\tau_i^-$ and the Lipschitz constant of $h_{i1}$ is bounded above by $\tau_{i\alpha}$. Suppose $p_{\mathcal{X}}(\mathbf{x}_{pure})$ and $p_{\mathcal{X}}(\hat{\mathbf{x}}_{pure})$ are the estimated densities of pure data $\mathbf{x}_{pure}$ and $\hat{\mathbf{x}}_{pure}$ respectively.*

*Then the following inequality holds:*

$$| \log p_{\mathcal{X}} (\hat{\mathbf{x}}_{pure}) - \log p_{\mathcal{X}} (\mathbf{x}_{pure})| \leq \eta \| [\mathbf{z}_D; \mathbf{0}] - [\mathbf{z}_{\hat{D}}; \mathbf{0}] \|$$
$$+ \frac{|\|\mathbf{z}_{\hat{D}}\|^2 - \|\mathbf{z}_D\|^2|}{2},$$

where $\eta = \sum_{i=1}^{T} (\tau_{i\alpha} \sqrt{d} \prod_{j=1}^{i-1} \tau_j^+) \prod_{k=1}^{T} \frac{1}{\tau_k}$. This theorem indicates that our method can approximate the density of the pure data and the estimation error mainly originates from the difference between estimated data sources $\mathbf{z}_{\hat{D}}$ and true data sources $\mathbf{z}_D$.

**Assumption 3.2.** For any $\mathbf{x}^a, \mathbf{x}^b \in \mathbb{R}^d$, there exists a constant $\varphi$ such that $|p_{\mathcal{X}} (\hat{\mathbf{x}}_{\text{pure}}^a) - p_{\mathcal{X}} (\hat{\mathbf{x}}_{\text{pure}}^b)| \geq \varphi \|\mathbf{x}^a - \mathbf{x}^b\|$. Moreover, if $\mathbf{x}^c$ is an anomaly, $p_{\mathcal{X}} (\hat{\mathbf{x}}_{\text{pure}}^c) \leq \max_{\mathbf{x} \in \mathcal{D}} p_{\mathcal{X}} (\hat{\mathbf{x}}_{\text{pure}})$

This assumption is reasonable since $\varphi$ can be calculated as $\inf_{\mathbf{x} \in \text{dom}(p_{\mathcal{X}})} \|\nabla_{\mathbf{x}} p_{\mathcal{X}} (\hat{\mathbf{x}}_{\text{pure}})\|$ and if the pure data of an anomaly has a density exceeding the maximum density observed in the pure data of all training data, its detection becomes considerably more challenging or even impossible.

**Theorem 3.3.** *Let $\mathbf{x}^c$ be an anomaly. Suppose that $\mathbf{x}^a, \mathbf{x}^b \in \mathcal{D}$ such that $\arg\max_{\mathbf{x}} p_{\mathcal{X}} (\hat{\mathbf{x}}_{pure}) = \mathbf{x}^a$ and $\arg\min_{\mathbf{x}} p_{\mathcal{X}} (\hat{\mathbf{x}}_{pure}) = \mathbf{x}^b$ and $p_{\mathcal{X}} (\hat{\mathbf{x}}_{pure}^a) = \varsigma_1, p_{\mathcal{X}} (\hat{\mathbf{x}}_{pure}^b) = \varsigma_2$. Then, under Assumption 3.2, if $\|\mathbf{x}^c - \mathbf{x}^a\| > \frac{\varsigma_1 - \varsigma_2}{\varphi}$, $\mathbf{x}^c$ can be detected as an anomaly.*

Theorem 3.3 shows that our proposed method can detect anomalies that are significantly distant from normal data. The proofs for the theorems are provided in Appendix A. We also compare the time complexity of density-based methods in Appendix B, while the actual training-time comparison is included in Appendix I.7.

## 4. Numerical Results

### 4.1. Experimental Settings

**Datasets** In our experiments, we evaluate the performance of 17 baseline methods on 47 widely used real-world datasets spanning multiple domains in a popular benchmark for anomaly detection proposed by (Han et al., 2022). Statistical information about these datasets is provided in Appendix G. In anomaly detection tasks, we follow the protocol of (Zong et al., 2018; Bergman & Hoshen, 2020; Shenkar & Wolf, 2022; Xu et al., 2023) by randomly partitioning normal samples in the datasets: 50% for training, while the remaining 50% are combined with all anomalous samples to form the test set. For outlier detection, the model is trained on the entire dataset to identify outliers within it, which is a transductive learning setting. Since our primary focus is on standard anomaly detection, we randomly select 10 datasets for evaluation under other anomaly detection

settings including outlier detection, AD on noisy data, and AD with anomaly contamination.

**Baselines** The 17 baselines include TCCM (Li et al., 2025), ADERH (Durani et al., 2026), DTE (Livernoche et al., 2023), MCM (Yin et al., 2024), DPAD (Fu et al., 2024), SLAD (Xu et al., 2023), ECOD (Li et al., 2022), ICL (Shenkar & Wolf, 2022), NAD (Qiu et al., 2021), DSVDD (Ruff et al., 2018a), RealNVP (Dinh et al., 2016), IF (Liu et al., 2008), AE (Hinton & Salakhutdinov, 2006), OC-SVM (Schölkopf et al., 2001), LOF (Breunig et al., 2000), $k$NN (Ramaswamy et al., 2000), KDE (Parzen, 1962). For TCCM, ADERH, DTE, MCM, DPAD, SLAD, ICL, and NAD, we use the code provided by the authors of the papers. For other methods, we use the code from the Python library PyOD (Chen et al., 2024). For all baseline methods and across all experiments, we perform grid search to identify the best-performing hyperparameter configuration on each dataset, unless stated otherwise. The detailed hyperparameter configuration for grid search is in Appendix E.

It should be mentioned that for standard anomaly detection, in addition to grid search, we also compare all methods under two other hyperparameter selection strategies: one is based on AutoUAD (Dai & Fan, 2025), and the other is based on the recommended hyperparameter settings in the original papers or code. The results are shown in Appendix I.2, including Table 15, Table 16, Table 17, and Table 18.

**Evaluation Metrics** We use the Area Under the Receiver Operating Characteristic Curve (AUROC) and the Area Under the Precision-Recall Curve (AUPRC) as evaluation metrics, following (Han et al., 2022; Xu et al., 2023; Yin et al., 2024; Li et al., 2025). These two metrics do not rely on specific decision thresholds and are capable of comprehensively assessing the performance of different methods. More details are provided in Appendix D.

### 4.2. Results of Standard Anomaly Detection

Table 1 reports the detailed performance of different methods on 14 datasets, and the average AUROC and AUPRC results across 47 datasets, with detailed results for each dataset available in Appendix I.2. NRDE achieves the best performance, outperforming the second-best method by more than 2%. Compared to RealNVP, NRDE demonstrates a significant improvement. Additionally, NRDE and NRDE-M outperform other baseline methods on a larger number of datasets. For example, on yeast, vertebral, and WPBC, baseline methods attain AUROC scores around 50%, indicating anomaly detection is particularly challenging for these datasets. In contrast, our method significantly outperforms baselines, highlighting its effectiveness in complex datasets. Notably, KDE and $k$NN—two traditional methods—outperform many other baseline methods. We attribute this phenomenon to two main factors. First, as mentioned

*Table 1.* AUROC (%) and AUPRC (%) with standard deviation of each method on 14 tabular datasets from ADBench. The detailed results on 47 datasets is shown in Apppendix I.2. Due to the page limitation, We only report the average performance, average rank, and paired $p$-values over all 47 datasets. The best and second-best results are highlighted in red and orange, respectively. Hyperparameter of each method on each dataset is tuned using grid search.

| AUROC | KDE (1962) | kNN (2000) | LOF (2000) | OC-SVM (2001) | IF (2008) | AE (2006) | DSVDD (2018) | RealNVP (2016) | NAD (2021) | ECOD (2022) | ICL (2022) | SLAD (2023) | DPAD (2024) | MCM (2024) | DTE-C (2024) | TCCM (2025) | ADERH (2025) | NRDE | NRDE-M |
|---|---|---|---|---|---|---|---|---|---|---|---|---|---|---|---|---|---|---|---|
| annthyroid | 91.4±0.0 | 94.1±0.0 | 95.4±0.0 | 91.7±0.0 | 93.1±0.0 | 93.5±2.3 | 92.3±1.6 | 96.1±0.5 | 87.7±3.2 | 78.7±0.0 | 83.5±0.5 | 87.3±0.4 | 82.8±4.8 | 98.2±0.0 | 97.9±0.1 | 94.1±0.8 | 91.7±0.4 | 97.3±0.5 | 98.4±0.3 |
| cover | 95.5±0.0 | 98.5±0.0 | 99.3±0.0 | 98.8±0.0 | 92.1±0.0 | 98.4±0.4 | 49.8±0.5 | 83.8±0.1 | 82.2±7.6 | 92.0±1.1 | 67.4±3.4 | 73.1±5.8 | 50.1±0.0 | 95.1±0.0 | 98.1±0.1 | 98.7±1.0 | 93.1±0.1 | 99.9±0.0 | 99.6±0.1 |
| mammography | 89.8±0.0 | 87.8±0.0 | 85.0±0.0 | 88.9±0.0 | 89.0±0.0 | 91.5±0.2 | 89.6±0.2 | 89.5±0.4 | 76.1±0.3 | 90.7±0.0 | 76.7±1.8 | 76.0±0.4 | 78.2±8.9 | 91.6±0.0 | 89.3±0.8 | 86.3±0.5 | 87.5±0.0 | 91.7±0.1 | 92.4±0.7 |
| PageBlocks | 95.0±0.0 | 96.2±0.0 | 97.5±0.0 | 96.1±0.0 | 94.1±0.0 | 97.0±0.5 | 96.4±0.1 | 92.0±0.9 | 92.9±0.1 | 91.4±0.0 | 91.7±0.9 | 90.0±1.6 | 90.8±4.1 | 79.8±0.0 | 96.7±0.3 | 95.7±1.4 | 94.4±0.0 | 93.5±0.5 | 97.2±0.5 |
| Pima | 78.5±0.0 | 79.0±0.0 | 77.3±0.0 | 76.5±0.0 | 76.9±0.0 | 76.2±0.8 | 75.7±0.4 | 79.3±0.6 | 63.5±1.2 | 61.5±0.0 | 70.2±0.3 | 64.7±7.2 | 75.0±0.7 | 70.3±0.8 | 73.1±1.5 | 74.9±3.5 | 77.2±0.2 | 80.0±0.3 | 81.1±0.8 |
| smtp | 88.2±0.0 | 93.6±0.0 | 94.9±0.0 | 87.2±0.0 | 93.5±0.0 | 94.6±0.0 | 91.9±0.1 | 93.4±0.1 | 95.3±0.2 | 87.9±0.0 | 65.0±15.0 | 96.2±1.2 | 84.1±0.8 | 88.4±0.0 | 95.5±0.0 | 91.8±4.6 | 93.0±0.3 | 90.6±1.1 | 96.3±0.3 |
| SpamBase | 86.0±0.0 | 84.5±0.0 | 82.4±0.0 | 81.7±0.0 | 79.1±0.0 | 82.6±0.4 | 83.3±0.1 | 80.1±0.3 | 80.9±0.5 | 66.0±0.0 | 82.3±0.2 | 84.8±1.8 | 71.2±3.1 | 73.6±0.0 | 85.4±0.7 | 86.4±0.0 | 84.5±0.4 | 84.5±0.3 | 82.0±0.3 |
| speech | 71.4±0.0 | 48.5±0.0 | 57.7±0.0 | 46.3±0.0 | 49.2±0.0 | 47.5±0.0 | 47.6±1.5 | 50.0±0.0 | 58.5±1.0 | 46.1±0.0 | 54.9±4.6 | 55.4±0.3 | 54.3±3.5 | 52.7±0.0 | 52.9±2.0 | 55.1±0.3 | 49.2±0.1 | 58.7±2.7 | 62.8±0.4 |
| Stamps | 95.1±0.0 | 92.3±0.0 | 91.9±0.0 | 92.9±0.0 | 95.2±0.0 | 93.6±1.5 | 90.6±0.4 | 93.6±1.4 | 79.7±2.7 | 86.7±0.0 | 89.0±0.6 | 81.5±1.1 | 92.5±1.5 | 89.8±0.0 | 89.6±2.2 | 90.8±1.2 | 91.9±0.1 | 96.9±1.6 | 97.4±1.0 |
| vertebral | 43.5±0.0 | 45.7±0.0 | 58.0±0.0 | 59.3±0.0 | 49.6±0.0 | 55.8±0.7 | 50.9±1.3 | 53.6±4.8 | 61.5±4.7 | 41.8±0.0 | 60.8±0.7 | 46.2±3.4 | 48.3±9.2 | 53.0±0.0 | 66.6±3.5 | 69.3±5.0 | 36.7±0.2 | 82.8±4.3 | 80.0±0.7 |
| Waveform | 76.9±0.0 | 76.4±0.0 | 76.7±0.0 | 72.7±0.0 | 73.6±0.0 | 71.6±0.8 | 73.1±2.9 | 72.5±1.6 | 75.3±1.0 | 60.0±0.0 | 64.3±1.5 | 52.2±0.6 | 82.3±1.9 | 77.8±0.0 | 64.9±0.1 | 75.7±1.8 | 77.4±0.4 | 86.7±1.3 | 80.2±2.8 |
| wine | 92.2±0.0 | 93.3±0.0 | 92.2±0.0 | 92.2±0.0 | 93.8±0.0 | 92.5±2.7 | 90.2±4.2 | 92.6±1.6 | 97.4±1.8 | 73.0±0.0 | 89.2±3.7 | 95.0±4.0 | 84.2±7.7 | 100.0±0.0 | 94.7±1.7 | 94.1±0.8 | 92.8±0.1 | 99.8±0.3 | 99.4±0.2 |
| WPBC | 52.5±0.0 | 51.7±0.0 | 53.5±0.0 | 51.2±0.0 | 54.6±0.0 | 52.6±1.2 | 49.8±3.5 | 58.0±1.0 | 57.5±0.9 | 47.0±0.0 | 57.8±0.8 | 49.8±1.8 | 51.0±4.8 | 57.3±0.0 | 53.8±1.3 | 51.2±0.6 | 50.1±0.0 | 77.5±5.0 | 64.7±2.5 |
| yeast | 43.2±0.0 | 47.6±0.0 | 49.5±0.0 | 48.3±0.0 | 47.4±0.0 | 50.0±0.2 | 50.9±0.3 | 51.0±1.4 | 60.2±2.3 | 45.3±0.0 | 57.6±3.4 | 55.3±1.3 | 55.3±4.2 | 50.1±0.0 | 52.0±2.4 | 59.0±1.0 | 40.1±0.2 | 61.2±2.5 | 63.4±0.9 |
| Avg. AUROC | 85.3 | 86.1 | 86.3 | 81.1 | 82.1 | 85.9 | 81.3 | 82.3 | 81.0 | 74.2 | 81.2 | 82.0 | 77.8 | 85.9 | 86.2 | 86.7 | 81.7 | 88.5 | 86.7 |
| Avg. Rank | 8.18 | 7.44 | 7.82 | 11.53 | 10.93 | 7.86 | 10.68 | 11.50 | 11.02 | 16.18 | 11.34 | 11.61 | 13.76 | 7.71 | 9.12 | 8.37 | 11.83 | 5.79 | 7.34 |
| p-value | 0.037 | 0.041 | 0.094 | 0.000 | 0.000 | 0.007 | 0.000 | 0.000 | 0.000 | 0.000 | 0.000 | 0.000 | 0.000 | 0.010 | 0.015 | 0.037 | 0.000 | - | 0.008 |
| **AUPRC** | | | | | | | | | | | | | | | | | | | |
| annthyroid | 66.2±0.0 | 72.0±0.0 | 77.0±0.0 | 66.9±0.0 | 65.8±0.0 | 69.3±7.3 | 69.5±3.0 | 77.0±2.8 | 43.4±7.1 | 40.8±0.0 | 48.1±2.6 | 60.1±0.5 | 54.7±6.6 | 85.5±0.0 | 84.3±0.3 | 73.8±0.8 | 65.1±1.3 | 75.6±3.0 | 85.7±2.1 |
| cover | 34.2±0.0 | 72.0±0.0 | 84.8±0.0 | 48.2±0.0 | 19.5±0.0 | 2.2±0.0 | 9.3±1.8 | 46.6±0.4 | 18.4±0.0 | 9.0±3.7 | 4.6±1.2 | 2.0±0.0 | 21.1±0.0 | 71.8±0.7 | 93.4±0.1 | 16.2±0.1 | | 96.5±0.1 | 84.6±6.8 |
| mammography | 41.6±0.0 | 41.9±0.0 | 35.0±0.0 | 41.2±0.0 | 47.1±0.0 | 59.7±1.9 | 38.5±0.8 | 44.8±2.2 | 11.1±0.4 | 54.0±0.0 | 30.0±0.8 | 20.2±1.3 | 36.6±7.2 | 43.9±0.0 | 48.0±0.3 | 39.0±2.3 | 38.6±0.3 | 49.6±6.8 | 59.4±3.5 |
| PageBlocks | 84.8±0.0 | 86.6±0.0 | 89.9±0.0 | 86.2±0.0 | 76.8±0.0 | 88.2±1.2 | 87.5±0.4 | 74.5±1.8 | 78.5±0.7 | 66.4±0.0 | 81.4±1.2 | 74.7±2.3 | 78.8±4.4 | 54.1±0.0 | 86.4±0.6 | 86.2±2.7 | 79.6±0.3 | 79.5±0.9 | 90.1±1.2 |
| Pima | 77.0±0.0 | 77.9±0.0 | 76.0±0.0 | 76.0±0.0 | 78.2±0.0 | 78.1±0.0 | 75.2±1.0 | 77.6±0.8 | 63.0±1.5 | 65.7±0.0 | 69.4±2.0 | 66.6±7.2 | 74.3±0.5 | 73.8±0.0 | 73.4±1.4 | 74.1±2.1 | 75.6±0.3 | 79.5±0.9 | 80.5±1.0 |
| smtp | 58.8±0.0 | 42.0±0.0 | 40.4±0.0 | 61.4±0.0 | 1.2±0.0 | 40.8±0.6 | 28.4±3.8 | 32.0±2.0 | 48.1±1.9 | 52.6±0.0 | 12.3±12.2 | 43.9±0.0 | 50.1±0.9 | 65.0±0.0 | | 52.8±12.3 | 43.6±6.1 | 57.6±0.9 | 57.9±0.5 |
| SpamBase | 88.3±0.0 | 86.6±0.0 | 83.8±0.0 | 85.1±0.0 | 84.1±0.0 | 85.5±0.3 | 86.5±0.1 | 80.5±0.4 | 83.8±0.8 | 68.9±0.0 | 87.0±0.2 | 87.9±0.2 | 74.3±2.6 | 78.8±0.0 | 87.2±0.3 | 88.9±0.2 | 84.2±0.1 | 85.3±0.6 | 80.9±0.6 |
| speech | 7.5±0.0 | 3.7±0.0 | 6.9±0.0 | 3.6±0.0 | 4.0±0.0 | 3.8±0.1 | 3.1±0.1 | 3.2±0.0 | 4.6±0.8 | 3.4±0.0 | 4.2±0.9 | 3.9±0.3 | 4.9±1.3 | 5.9±0.0 | 4.0±0.5 | 5.0±0.1 | 3.9±0.1 | 4.9±0.7 | 5.1±0.2 |
| Stamps | 63.7±0.0 | 54.2±0.0 | 55.4±0.0 | 56.3±0.0 | 69.4±0.0 | 59.5±4.2 | 49.6±1.1 | 61.8±6.1 | 33.2±3.4 | 45.2±0.0 | 61.3±4.7 | 36.9±2.0 | 57.2±3.9 | 51.5±0.0 | 54.1±3.7 | 55.0±2.5 | 53.6±0.1 | 78.4±13.0 | 88.3±4.8 |
| vertebral | 19.7±0.0 | 20.3±0.0 | 29.6±0.0 | 26.4±0.0 | 22.2±0.0 | 24.7±0.1 | 23.1±0.5 | 25.3±2.2 | 33.8±3.2 | 19.9±0.0 | 30.3±0.8 | 20.3±1.0 | 23.7±5.3 | 23.2±0.0 | 33.0±4.2 | 35.5±5.4 | 17.4±0.0 | 51.4±7.4 | 54.9±2.3 |
| Waveform | 27.6±0.0 | 25.4±0.0 | 33.8±0.0 | 11.2±0.0 | 10.8±0.0 | 15.8±1.0 | 10.3±0.8 | 11.3±0.6 | 49.0±0.3 | 7.6±0.0 | 40.3±1.6 | 6.8±0.1 | 32.4±9.8 | 20.4±0.0 | 11.5±0.0 | 19.6±1.7 | 15.4±0.0 | 32.3±0.9 | 14.8±1.4 |
| wine | 58.2±0.0 | 61.5±0.0 | 52.7±0.0 | 58.7±0.0 | 69.1±0.0 | 65.3±10.9 | 59.0±13.4 | 56.6±7.1 | 88.4±7.7 | 30.5±0.0 | 60.3±8.8 | 73.1±21.0 | 47.1±14.4 | 100.0±0.0 | 72.3±9.7 | 66.2±4.8 | 59.8±0.5 | 99.1±1.5 | 97.2±1.2 |
| WPBC | 38.4±0.0 | 38.2±0.0 | 39.0±0.0 | 37.8±0.0 | 39.9±0.0 | 39.8±1.8 | 39.0±2.0 | 43.8±1.0 | 45.5±3.2 | 35.4±0.0 | 47.4±2.7 | 38.9±1.1 | 40.2±3.0 | 42.5±0.0 | 40.0±2.1 | 37.8±0.6 | 36.9±0.0 | 68.7±6.3 | 52.7±6.4 |
| yeast | 48.2±0.0 | 49.6±0.0 | 52.4±0.0 | 51.1±0.0 | 52.3±0.0 | 52.1±0.1 | 51.3±0.0 | 51.7±0.6 | 56.6±1.1 | 50.0±0.0 | 55.6±1.9 | 54.4±0.6 | 54.8±3.1 | 51.7±0.0 | 52.2±0.8 | 56.9±1.4 | 46.1±0.4 | 58.5±2.9 | 61.1±1.3 |
| Avg. AUPRC | 63.0 | 63.6 | 64.5 | 56.9 | 54.3 | 63.9 | 57.8 | 52.8 | 54.2 | 45.0 | 56.8 | 56.5 | 55.6 | 65.1 | 61.8 | 66.0 | 55.6 | 68.1 | 66.3 |
| Avg. Rank | 8.29 | 8.27 | 8.37 | 11.19 | 10.73 | 7.74 | 10.88 | 12.85 | 10.90 | 15.34 | 10.09 | 11.43 | 12.35 | 7.37 | 9.79 | 7.71 | 12.51 | 6.82 | 7.36 |
| p-value | 0.021 | 0.015 | 0.078 | 0.000 | 0.000 | 0.041 | 0.000 | 0.000 | 0.003 | 0.000 | 0.001 | 0.000 | 0.000 | 0.162 | 0.005 | 0.249 | 0.000 | - | 0.149 |

*Table 2.* Average AUROC (%) on anomaly detection with injected noise experiments over 10 tabular datasets of ADBench.

| Metric | KDE | KNN | LOF | OCSVM | IF | AE | DSVDD | RealNVP | NAD | ECOD | ICL | SLAD | DPAD | MCM | DTE-C | TCCM | ADERH | NRDE | NRDE-M |
|---|---|---|---|---|---|---|---|---|---|---|---|---|---|---|---|---|---|---|---|
| Avg. AUROC | 70.3 | 69.2 | 68.9 | 67.3 | 65.4 | 66.9 | 69.4 | 62.4 | 60.8 | 50.1 | 69.3 | 69.5 | 68.6 | 65.5 | 61.7 | 67.8 | 65.1 | 81.8 | 77.5 |

*Table 3.* Average AUROC (%) on outlier detection over 10 tabular datasets. * indicates that we report the performance in their paper.

| Metric | KDE | KNN | LOF | OCSVM | IF | AE | DSVDD | RealNVP | NAD | ECOD | ICL | SLAD | DPAD | MCM | DTE-C | TCCM | ADERH | ODIM* | NRDE | NRDE-M |
|---|---|---|---|---|---|---|---|---|---|---|---|---|---|---|---|---|---|---|---|---|
| Avg. AUROC | 40.7 | 68.2 | 58.7 | 65.4 | 69.2 | 67.1 | 63.8 | 66.6 | 66.2 | 62.7 | 57.6 | 64.4 | 61.5 | 76.1 | 67.7 | 69.2 | 69.1 | 65.2 | 79.4 | 78.2 |

earlier, tabular data typically consists of features that inherently provide excellent semantic differences. As a result, even Euclidean distance can capture meaningful distinctions between samples. Second, as demonstrated in (Jiang, 2017; Gu et al., 2019), these two methods provide more explicit predictions for datasets with lower dimensions and more samples, which aligns with the experimental results and the curse of dimensionality.

## 4.3. Results of AD with Anomaly Contamination

In practical AD scenarios, the training set typically contains a small but unavoidable proportion of anomalous samples. To evaluate the robustness and performance of all methods in this scenario, We introduce varying proportions of anomalies into the training set and conduct experiments on these contaminated datasets. The contamination ratio ranges from 1% to 10% of the training set size. We report the average performance of all methods in Figure 4, while the detailed experimental results for each dataset are in Appendix I.3. From the figure, we observe that as the anomaly ratio increases, the performance of all methods decreases. In this

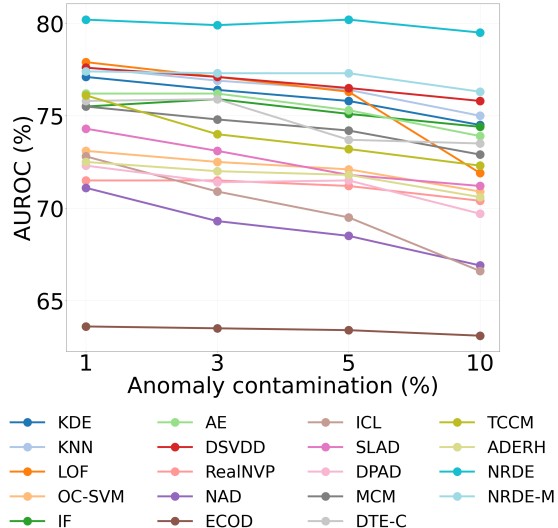

*Figure 4.* Average AUROC values across 10 datasets of AD experiments with anomaly contamination, contamination ratio ranging from 1% to 10%.

scenario, NRDE consistently achieves superior performance

over other methods, demonstrating its robustness to anomalies in the training set. It should be noted that the AUROC performance of NRDE is affected, with a less significant performance drop than that observed in other methods.

### 4.4. Results of Anomaly Detection on Noisy Data

In real-world AD, observations can be corrupted by noise. To assess the robustness of different methods under such conditions, we inject additive Gaussian noise into both the training set and the anomalous test samples, i.e., $\mathcal{N}(\mathbf{0}, 0.1\mathbf{I}_d)$, while perturbing normal test samples with a stronger noise level $\mathcal{N}(\mathbf{0}, 0.2\mathbf{I}_d)$. This setting reduces the separability between normal and anomalous test data, making the detection task more challenging than an equal-noise injection setting. Note that we first normalize the data using z-score normalization and then inject the perturbations. Table 2 reports the average results where NRDE consistently maintains superior performance compared to competing approaches, highlighting its robustness to noise perturbations. We also consider a symmetric variant where anomalous test samples receive stronger noise. In this setting, most methods exhibit improved performance compared to the standard anomaly detection setup, suggesting that the task is less challenging. NRDE still outperforms other methods under this setting. The detailed results are shown in Appendix I.4.

### 4.5. Results of Outlier Detection

To evaluate the effectiveness of anomaly detection methods in the outlier detection (transductive) setting, we conduct experiments on several datasets where all samples are used for both training and testing. Under this setting, we compare NRDE with the recently proposed ODIM (Kim et al., 2023), which is specifically designed for outlier detection. As shown in Table 3, NRDE outperforms competing methods, indicating strong robustness to the presence of anomalies during training. Detailed results for each dataset are provided in I.5.

### 4.6. More Results

Further experiments validating our assumptions and motivation are presented in Appendix H. Additional visualization results are provided in Appendix I.1. Ablation studies and hyperparameter (architecture) analysis are reported in Appendix I.6. The training time comparison is included in Appendix I.7, and the training dynamics are shown in Appendix I.8.

## 5. Conclusion

We proposed a novel and effective method NRDE for anomaly detection in tabular data. Our key observation is that data is typically generated by independent sources, which can be categorized into pure data sources and noise sources. By distinguishing these sources using the Jacobian matrix, we can approximate the density of the pure data which is unaffected by noise. This allows NRDE to be robust to noise and effectively identify both anomalous data and noisy normal data. We provided theoretical analysis on the estimation error, the reliability of our proposed method, and the time complexity of density-based approaches. Numerical experiments demonstrated that NRDE outperforms 17 baseline methods across 47 real-world datasets. Furthermore, NRDE exhibits robustness to anomalies in the training set and noise inside the data.

## Acknowledgements

This work was partially supported by the General Program of the Natural Science Foundation of Guangdong Province under Grant No.2024A1515011771, the National Natural Science Foundation of China under Grant No.62376236, and the Shenzhen Stability Science Program 2023.

## Impact Statement

This paper presents work whose goal is to advance the field of Machine Learning. There are many potential societal consequences of our work, none which we feel must be specifically highlighted here.

## Limitations

NRDE leverages the variance across sources to distinguish noise from signal, with noise being isolated through the proposed procedure. This design is primarily grounded in Assumption 6. While this assumption holds for many real-world datasets, there may exist scenarios where it is violated, in which case the effectiveness of NRDE may degrade. We provide a discussion and empirical verification of this limitation in Appendix H. In addition, NRDE may incur substantial computational overhead on extremely high-dimensional data due to the computation of Jacobian matrices. A practical remedy is to first apply dimensionality reduction techniques, as discussed in Appendix I.9.

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

*Table 4.* Organization of the Appendix.

# A. Proofs of Theorems

## A.1. Proof for Theorem 3.1

**Lemma A.1.** *(Behrmann et al., 2021) Let $f_{\mathcal{W}_i}$ be a coupling flow, the Lipschitz constant of the forward $f_{\mathcal{W}_i}$ can be locally bounded for $\mathbf{x} \in [a, b]^d$ as:*

$$Lip(f_{\mathcal{W}_i}) \leq \max(1, c_g) + M, \tag{22}$$

*Where $\exp(h_{i1}(\mathbf{x})) \leq c_g$ and $M = \max(|a|, |b|) \cdot c_{g'} \cdot Lip(h_{i1}) + Lip(h_{i2})$. Similarly, the Lipschitz constant of the reverse*

$f_{\mathcal{W}_i}^{-1}$ *can be locally bounded for* $\mathbf{y}_i \in [a^*, b^*]^d$ *as:*

$$Lip(f_{\mathcal{W}_i}^{-1}) \leq \max(1, c_{\frac{1}{g}}) + M^*, \tag{23}$$

*Where* $M^* = \max(|a^*|, |b^*|) \cdot c_{(\frac{1}{g}')} \cdot Lip(h_{i1}) \cdot c_t + c_{\frac{1}{g}} \cdot Lip(h_{i2})$

*Proof.* According to Lemma A.1, here we can assume that $\mathbf{x}_i, \mathbf{y}_i$ are both bounded since data is preprocessed and normalized, then we have the bi-Lipschitz constant of $f_{\mathcal{W}_i}$ are bounded as:

$$\begin{aligned}
\tau_i^- \|\mathbf{x}^{(i)} - \hat{\mathbf{x}}^{(i)}\| &\leq \|f_{\mathcal{W}_i}(\mathbf{x}^{(i)}) - f_{\mathcal{W}_i}(\hat{\mathbf{x}}^{(i)})\| \leq \tau_i^+ \|\mathbf{x}^{(i)} - \hat{\mathbf{x}}^{(i)}\|, \\
\frac{1}{\tau_i^+} \|f_{\mathcal{W}_i}(\mathbf{x}^{(i)}) - f_{\mathcal{W}_i}(\hat{\mathbf{x}}^{(i)})\| &\leq \|\mathbf{x}^{(i)} - \hat{\mathbf{x}}^{(i)}\| \leq \frac{1}{\tau_i^-} \|f_{\mathcal{W}_i}(\mathbf{x}^{(i)}) - f_{\mathcal{W}_i}(\hat{\mathbf{x}}^{(i)})\|
\end{aligned} \tag{24}$$

The determinant of $\nabla_{\mathbf{x}^{(i)}} f_{\mathcal{W}_i}(\mathbf{x}^{(i)})$ can be calculated as:

$$\log |\det \nabla_{\mathbf{x}^{(i)}} f_{\mathcal{W}_i}(\mathbf{x}^{(i)})| = h_{i1}(\mathbf{x}_\alpha^{(i)}) \cdot \mathbf{1}, \tag{25}$$

Suppose $h_{i1}(\mathbf{x}) = \mathbf{W}_{i,L}(\phi(\cdots \phi(\mathbf{W}_{i,2}\phi(\mathbf{W}_{i,1}\mathbf{x}))\cdots))$ and $h_{i2}(\mathbf{x}) = \hat{\mathbf{W}}_{i,L}\left(\phi\left(\cdots \phi\left(\hat{\mathbf{W}}_{i,2}\phi\left(\hat{\mathbf{W}}_{i,1}\mathbf{x}\right)\right)\cdots\right)\right)$ are two neural networks comprising $L$ layers and $\phi$ represents the activation function. Consider different $\mathbf{x}^{(i)}, \hat{\mathbf{x}}^{(i)}$, denote $\rho$ the Lipschitz constant of $\phi$, we have:

$$\begin{aligned}
&|\log|\det \nabla_{\mathbf{x}^{(i)}} f_{\mathcal{W}_i}(\mathbf{x}^{(i)})| - \log|\det \nabla_{\hat{\mathbf{x}}^{(i)}} f_{\mathcal{W}_i}(\hat{\mathbf{x}}^{(i)})|| \\
=&\|h_{i1}(\mathbf{x}_\alpha^{(i)}) - h_{i1}(\hat{\mathbf{x}}_\alpha^{(i)})\|_1 \\
\leq&\sqrt{d}\|h_{i1}(\mathbf{x}_\alpha^{(i)}) - h_{i1}(\hat{\mathbf{x}}_\alpha^{(i)})\| \\
\leq&\sqrt{d}\rho^{L-1}\prod_{l=1}^{L}\|\mathbf{W}_{i,l}\|_2\|\mathbf{x}_\alpha^{(i)} - \hat{\mathbf{x}}_\alpha^{(i)}\| \\
\leq&\sqrt{d}\rho^{L-1}\prod_{l=1}^{L}\|\mathbf{W}_{i,l}\|_2\|\mathbf{x}^{(i)} - \hat{\mathbf{x}}^{(i)}\| \\
=&\tau_{i\alpha}\sqrt{d}\|\mathbf{x}^{(i)} - \hat{\mathbf{x}}^{(i)}\| \\
\leq&\tau_{i\alpha}\sqrt{d}\prod_{j=1}^{i-1}\tau_j^+\|\mathbf{x} - \hat{\mathbf{x}}\|
\end{aligned} \tag{26}$$

Where $\tau_{i\alpha} = \rho^{L-1}\prod_{l=1}^{L}\|\mathbf{W}_{i,l}\|_2$ is the Lipschitz constant of $h_{i1}$. Then, we can conclude that $\log|\det(\nabla_x F_{\mathcal{W}}(\mathbf{x}))|$ has a Lipschitz constant:

$$\begin{aligned}
&|\log|\det(\nabla_{\mathbf{x}} F_{\mathcal{W}}(\mathbf{x}))| - \log|\det(\nabla_{\hat{\mathbf{x}}} F_{\mathcal{W}}(\hat{\mathbf{x}}))|| \\
=&|\sum_{i=1}^{T}\left(\log|\det \nabla_{\mathbf{x}^{(i)}} f_{\mathcal{W}_i}(\mathbf{x}^{(i)})| - \log|\det \nabla_{\hat{\mathbf{x}}^{(i)}} f_{\mathcal{W}_i}(\hat{\mathbf{x}}^{(i)})|\right)| \\
\leq&\sum_{i=1}^{T}|\log|\det \nabla_{\mathbf{x}^{(i)}} f_{\mathcal{W}_i}(\mathbf{x}^{(i)})| - \log|\det \nabla_{\hat{\mathbf{x}}^{(i)}} f_{\mathcal{W}_i}(\hat{\mathbf{x}}^{(i)})|| \\
\leq&\sum_{i=1}^{T}\tau_{i\alpha}\sqrt{d}\|\mathbf{x}^{(i)} - \hat{\mathbf{x}}^{(i)}\| \\
\leq&\sum_{i=1}^{T}(\tau_{i\alpha}\sqrt{d}\prod_{j=1}^{i-1}\tau_j^+)\|\mathbf{x} - \hat{\mathbf{x}}\|
\end{aligned} \tag{27}$$

Suppose the true data sources are $[\mathbf{s}_D; \mathbf{0}]$ or $[\mathbf{z}_D; \mathbf{0}]$, and the estimated data sources using $\hat{m}$ are $[\mathbf{s}_{\hat{D}}; \mathbf{0}]$ or $[\mathbf{z}_{\hat{D}}; \mathbf{0}]$. Then according to (24), we have the estimated error between $\mathbf{x}_{\mathrm{pure}}$ and $\hat{\mathbf{x}}_{\mathrm{pure}}$ is bounded by:

$$\|\mathbf{x}_{\mathrm{pure}} - \hat{\mathbf{x}}_{\mathrm{pure}}\| \leq \prod_{i=1}^{T} \frac{1}{\tau_i^{-}} \|[\mathbf{z}_D; \mathbf{0}] - [\mathbf{z}_{\hat{D}}; \mathbf{0}]\| \tag{28}$$

Thus, the estimated error between $\log p_{\mathcal{X}}(\hat{\mathbf{x}}_{\mathrm{pure}})$ and $\log p_{\mathcal{X}}(\mathbf{x}_{\mathrm{pure}})$ is bounded by:

$$
\begin{aligned}
& |\log p_{\mathcal{X}}(\hat{\mathbf{x}}_{\mathrm{pure}}) - \log p_{\mathcal{X}}(\mathbf{x}_{\mathrm{pure}})| \\
& = |\log|\det(\nabla_{\mathbf{x}_{\mathrm{pure}}} F_{\mathcal{W}}(\mathbf{x}_{\mathrm{pure}}))| - \frac{\|[\mathbf{z}_D; \mathbf{0}]\|^2}{2} - \log|\det(\nabla_{\hat{\mathbf{x}}_{\mathrm{pure}}} F_{\mathcal{W}}(\hat{\mathbf{x}}_{\mathrm{pure}}))| + \frac{\|[\mathbf{z}_{\hat{D}}; \mathbf{0}]\|^2}{2} \\
& \leq |\log|\det(\nabla_{\mathbf{x}_{\mathrm{pure}}} F_{\mathcal{W}}(\mathbf{x}_{\mathrm{pure}}))| - \log|\det(\nabla_{\hat{\mathbf{x}}_{\mathrm{pure}}} F_{\mathcal{W}}(\hat{\mathbf{x}}_{\mathrm{pure}}))|| + \frac{|\|\mathbf{z}_{\hat{D}}\|^2 - \|\mathbf{z}_D\|^2|}{2} \\
& \leq \sum_{i=1}^{T} (\tau_{i\alpha} \sqrt{d} \prod_{j=1}^{i-1} \tau_j^{+}) \|\hat{\mathbf{x}}_{\mathrm{pure}} - \mathbf{x}_{\mathrm{pure}}\| + \frac{|\|\mathbf{z}_{\hat{D}}\|^2 - \|\mathbf{z}_D\|^2|}{2} \\
& \leq \sum_{i=1}^{T} (\tau_{i\alpha} \sqrt{d} \prod_{j=1}^{i-1} \tau_j^{+}) \prod_{k=1}^{T} \frac{1}{\tau_k^{-}} \|[\mathbf{z}_D; \mathbf{0}] - [\mathbf{z}_{\hat{D}}; \mathbf{0}]\| + \frac{|\|\mathbf{z}_{\hat{D}}\|^2 - \|\mathbf{z}_D\|^2|}{2}
\end{aligned}
\tag{29}
$$

This finishes the proof. □

### A.2. Proof for Theorem 3.3

*Proof.* By Assumption 3.2, we have that:

$$|p_{\mathcal{X}}(\hat{\mathbf{x}}_{\mathrm{pure}}^c) - p_{\mathcal{X}}(\hat{\mathbf{x}}_{\mathrm{pure}}^a)| \geq \varphi \|\mathbf{x}^a - \mathbf{x}^c\| \tag{30}$$

If $\|\mathbf{x}^a - \mathbf{x}^c\| > \frac{\varsigma_1 - \varsigma_2}{\varphi}$, then we have:

$$
\begin{aligned}
p_{\mathcal{X}}(\hat{\mathbf{x}}_{\mathrm{pure}}^a) - p_{\mathcal{X}}(\hat{\mathbf{x}}_{\mathrm{pure}}^c) &> \varsigma_1 - \varsigma_2 \\
\varsigma_1 - p_{\mathcal{X}}(\hat{\mathbf{x}}_{\mathrm{pure}}^c) &> \varsigma_1 - \varsigma_2 \\
p_{\mathcal{X}}(\hat{\mathbf{x}}_{\mathrm{pure}}^c) &< \varsigma_2 \\
p_{\mathcal{X}}(\hat{\mathbf{x}}_{\mathrm{pure}}^c) &< \min_{\mathbf{x} \in \mathcal{D}} p_{\mathcal{X}}(\hat{\mathbf{x}}_{\mathrm{pure}})
\end{aligned}
\tag{31}
$$

Now we have the density of $\hat{\mathbf{x}}_{\mathrm{pure}}^c$ is even smaller than the smallest density of pure data from $\mathcal{D}$, thus it can be detected as an anomaly. This finishes the proof. □

### A.3. Derivation of $p_{\bar{\mathcal{X}}}(\mathbf{x}_{\mathbf{pure}})$ and Its Connection with $p_{\mathcal{X}}(\mathbf{x}_{\mathbf{pure}})$

The support of $\mathbf{x}_{\mathrm{pure}}$ is an $m$-dimensional manifold $M$ embedded in $\mathbb{R}$. Let $g(\mathbf{z}_D) := F^{-1}(\mathbf{z}_D, \mathbf{0})$ The induced Riemannian metric on the manifold is given by:

$$\mathbf{M}(\mathbf{z}_D) = J_g(\mathbf{z}_D)^{\top} J_g(\mathbf{z}_D) \tag{32}$$

where $J_g$ denote the Jacobian of $g$, i.e., $\nabla_{\mathbf{z}_D} g(\mathbf{z}_D)$. The volume element on the manifold, relative to the parameter space $\mathbf{s}_D$ is

$$dV = \sqrt{\det\left[J_g(\mathbf{z}_D)^T J_g(\mathbf{z}_D)\right]} d\mathbf{z}_D. \tag{33}$$

The probability in the latent space is:

$$\mathbb{P}(\mathbf{z}_D \in B) = \int_B p_{\mathcal{Z}}^D(\mathbf{z}_D) d\mathbf{z}_D. \tag{34}$$

This probability must equal the probability on the manifold $M$. For a measurable set $A \subset M$ :

$$\mathbb{P}\left(\mathbf{x}_{\text{pure}} \in A\right) = \int_{g^{-1}(A)} p_{\mathcal{Z}}^{D}\left(\mathbf{z}_{D}\right) d\mathbf{z}_{D} \tag{35}$$

We change the variable of integration from $\mathbf{z}_D$ to $\mathbf{x}_{\text{pure}} \in M$ and use the manifold volume element to obtain

$$d\mathbf{z}_{D} = \frac{d\mathcal{H}^{m}(\mathbf{x}_{\text{pure}})}{\sqrt{\det\left[J_{g}\left(\mathbf{z}_{D}\right)^{\top} J_{g}\left(\mathbf{z}_{D}\right)\right]}}, \tag{36}$$

where $d\mathcal{H}^m$ is the $m$-dimensional Hausdorff measure on $M$. Substituting this into the integral, we have

$$\mathbb{P}\left(\mathbf{x}_{\text{pure}} \in A\right) = \int_{A} \frac{p_{\mathcal{Z}}^{D}\left(\mathbf{z}_{D}\right)}{\sqrt{\det\left[J_{g}\left(\mathbf{z}_{D}\right)^{\top} J_{g}\left(\mathbf{z}_{D}\right)\right]}} d\mathcal{H}^{m}(\mathbf{x}) \tag{37}$$

Therefore, the probability density function on the manifold $M$ with respect to the Hausdorff measure is as follows

$$p_{\bar{\mathcal{X}}}\left(\mathbf{x}_{\text{pure}}\right) = \frac{p_{\mathcal{Z}}^{D}\left(\mathbf{z}_{D}\right)}{\sqrt{\det\left[J_{g}\left(\mathbf{z}_{D}\right)^{\top} J_{g}\left(\mathbf{z}_{D}\right)\right]}} = \frac{\mathcal{N}\left(\mathbf{z}_{D}; \mathbf{0}, \mathbf{I}_{m}\right)}{\sqrt{\det\left[J_{g}\left(\mathbf{z}_{D}\right)^{\top} J_{g}\left(\mathbf{z}_{D}\right)\right]}} \tag{38}$$

where the second equality used the fact that $\mathbf{s}_D$ and $\mathbf{s}_N$ are independent.

On the other hand, we have

$$\begin{aligned} p_{\mathcal{X}}\left(\mathbf{x}_{\text{pure}}\right) &= p_{\mathcal{Z}}\left(\mathbf{z}_{D}, \mathbf{0}\right)\left|\det J_{F}\left(\mathbf{x}_{\text{pure}}\right)\right| \\ &= c\mathcal{N}\left(\mathbf{z}_{D}; \mathbf{0}, \mathbf{I}_{m}\right)\left|\det J_{F}\left(\mathbf{x}_{\text{pure}}\right)\right| \end{aligned} \tag{39}$$

where $c = \frac{1}{2\pi^{(d-m)/2}}$. It follows that

$$\begin{aligned} p_{\mathcal{X}}\left(\mathbf{x}_{\text{pure}}\right) &= p_{\bar{\mathcal{X}}}\left(\mathbf{x}_{\text{pure}}\right) \times c\left|\det J_{F}\left(\mathbf{x}_{\text{pure}}\right)\right| \sqrt{\det\left[J_{g}\left(\mathbf{z}_{D}\right)^{\top} J_{g}\left(\mathbf{z}_{D}\right)\right]} \\ &= p_{\bar{\mathcal{X}}}\left(\mathbf{x}_{\text{pure}}\right) \times c\left|\det J_{F^{-1}}\left(\mathbf{z}_{D}, \mathbf{0}\right)\right|^{-1} \sqrt{\det\left[J_{g}\left(\mathbf{z}_{D}\right)^{\top} J_{g}\left(\mathbf{z}_{D}\right)\right]} \end{aligned} \tag{40}$$

## B. Time Complexity of Density-based Methods

Suppose that $F_{\mathcal{W}}$ is a sequence of $T$ flows defined in (2), and $h_{i1}, h_{i2}$ are two MLPs of $L$ layers parameterized by $\{\mathbf{W}_{i,j}\}_{j=1}^{L}, \{\hat{\mathbf{W}}_{i,j}\}_{j=1}^{L}$, where $\mathbf{W}_{i,j}, \hat{\mathbf{W}}_{i,j} \in \mathbb{R}^{d_{i,j} \times d_{i,j-1}}$, $j \in [L]$. Consider a batch of $B$ data points, the time complexity of our method per iteration is $\mathcal{O}(B \sum_{i=1}^{T}(d_{i,L} \sum_{j=0}^{L-2} d_{i,j}d_{i,j+1}))$, and the space complexity is $\mathcal{O}(B \sum_{i=1}^{T} \sum_{j=0}^{L} d_{i,j}d_{i,j+1}))$ which primarily arises from the computation of the Jacobian matrix. Here, we also compare the testing time complexity of a few representative density-based methods. We assume that DAGMM (Zong et al., 2018) contains $K$ Gaussians and the encoder and decoder have $\hat{L}$ layers, with $ith$ layer of encoder being $\mathbf{W}_{E,i} \in \mathbb{R}^{d_i \times d_{i-1}}$ and $ith$ layer of decoder being $\mathbf{W}_{D,i} \in \mathbb{R}^{d_{L+1-i} \times d_{L-i}}$. For DPAD (Fu et al., 2024), we assume that the size of its neural network is the same as that of the encoder of DAGMM. Suppose we have one testing data, the time complexity of density-based methods is shown in Table 5.

We notice that traditional density-based methods, such as KNN and KDE, require comparing test data against the entire training set to generate anomaly scores. Consequently, these methods become computationally inefficient as dataset sizes grow, since the time complexity grows linearly with the number of training samples. DPAD encounters a similar issue due to its reliance on KNN, although it mitigates this by employing a neural network for dimensionality reduction. In contrast, methods like DAGMM, RealNVP, and our proposed NRDE primarily utilize neural network outputs for anomaly scoring, which do not depend on the training set.

*Table 5.* Time complexity comparison of density-based methods in testing stage.

| | Testing Complexity |
|---|---|
| KNN | $\mathcal{O}(nd)$ |
| KDE | $\mathcal{O}(nd)$ |
| LOF | $\mathcal{O}(nd)$ |
| DAGMM | $\mathcal{O}\left( \sum_{l=1}^{\hat{L}} d_{l-1}d_l + d_{\hat{L}}^3 \right)$ |
| DPAD | $\mathcal{O}(\sum_{l=1}^{\hat{L}} d_{l-1}d_l + d_{\hat{L}}n)$ |
| RealNVP | $\mathcal{O}\left( \sum_{i=1}^{T} \sum_{j=0}^{L-1} d_{i,j}d_{i,j+1} \right)$ |
| NRDE | $\mathcal{O}\left( \sum_{i=1}^{T} \sum_{j=0}^{L-1} d_{i,j}d_{i,j+1} \right)$ |

# C. Properties of the regularizer $\mathcal{R}()$

Briefly speaking, our objective is to construct a Jacobian matrix in which the row norms exhibit a clear separation—some being significantly larger than others—so that we can distinguish between pure data sources and noise sources. Consider the derivative $\frac{\partial \mathcal{R}(Q)}{\partial Q_{i,j}} = \frac{sign(Q_{i,j})}{2\sqrt{||Q_{i:}||}}$, where $||Q_{i:}||$ is the $\ell_1$ norm. In this formulation, rows with larger norms receive a smaller penalty from the regularizer, whereas rows with smaller norms receive a larger penalty. This naturally encourages row-wise sparsity and separation. Moreover, unlike the conventional $\ell_{2,1}$ norm, where smaller entries in the same row receive a smaller penalty, our regularizer $\mathcal{R}$ imposes the same penalty on all entries within a given row—avoiding the vanishing-penalty problem for small entries—thereby enhancing both separation and sparsity. Thus, in theory, the formulation is suitable for our task.

# D. Implementation Details

*Table 6.* Network architecture of MLP used in normalizing flow (RealNVP).

| Tabular |
|---|
| Dimension_input=$2d$ |
| Dimension_firstlayer=$b$ |
| Linear($2d, b$), LeakyReLU() |
| Linear($b, 2d$) |

All experiments are conducted using the PyTorch framework on a system equipped with an NVIDIA RTX 4090 GPU and an Intel Core i9-12900K CPU. Each experiment is performed 5 times to obtain the mean value and standard deviation. To ensure a consistent network architecture for fair comparison, we employ two 2-layer multilayer perceptrons (MLPs) for RealNVP and NRDE, corresponding to a parameter setting of $T = 2$ in (1) where LeakyReLU is used as the activation function. Note that the outputs of $h_{i1}$ and $h_{i2}$ are actually the split output from the same MLP. The detailed network architecture is shown in Table 6, where $b = 2048$. Additionally, we use Adam as our optimizer and set the batch size to 512 for all experiments, while the number of training epochs is set to 100. Since the scale of the Jacobian norm in different datasets can be largely different, as shown in Figure 6, we use grid search to determine the best hyperparameter configuration on each dataset for NRDE: $lr \in \{0.001, 0.005, 0.01\}$ and $\lambda \in \{0.01, 0.1, 1\}$. To provide a fair comparison between our proposed NRDE and other baseline methods, the same hyperparameter-tuning strategy is also used for all other baseline methods.

# E. Hyperparameter configurations of baseline methods

In our experiments, we perform a grid search on each dataset to identify the best hyperparameter configuration for each baseline method, and use these configurations in the following evaluations. The detailed hyperparameter configurations for different methods are summarized in Table 7. Note that ECOD has no hyperparameters.

*Table 7.* Hyperparameter combinations of traditional and deep AD methods used for grid search.

| Method | Hyperparameter 1 | Hyperparameter 2 | Total Methods |
|---|---|---|---|
| KDE | gamma: [1, 5, 10] | kernel: ['linear', 'rbf', 'sigmoid'] | 9 |
| LOF | n_neighbors: [5, 10, 20, 30, 50] | - | 5 |
| kNN | n_neighbors: [5, 10, 20, 30, 50] | - | 5 |
| OCSVM | nu (train error tol): [0.1, 0.3, 0.5, 0.7, 0.9] | kernel: ['linear', 'rbf', 'sigmoid'] | 15 |
| IF | n_estimators: [50, 100, 150, 200] | max_features: [1, 2, 3] | 12 |
| AE | lr: [0.01, 0.005, 0.001] | epoch_num: [10, 50, 100] | 9 |
| DSVDD | lr: [0.001, 0.01, 0.005] | epoch_num: [100, 50, 10] | 9 |
| RealNVP | lr: [0.001, 0.01, 0.005] | - | 3 |
| NeutralAD | lr: [0.001, 0.01, 0.005] | rep_dim: [128, 64, 256] | 9 |
| ICL | lr: [0.001, 0.01, 0.005] | rep_dim: [128, 64, 256] | 9 |
| ECOD | - | - | 1 |
| SLAD | lr: [0.001, 0.01, 0.005] | n_ensemble: [20, 10, 30] | 9 |
| DPAD | gamma: [0.1, 0.5, 1] | lambda: [0.01, 0.1, 1] | 9 |
| DTE-C | lr: [0.0001, 0.0005, 0.001] | T: [400, 200, 600] | 9 |
| MCM | lr: [0.01, 0.005, 0.001] | lambda: [0.1, 1, 10] | 9 |
| TCCM | epoch_num: [100, 50, 200] | lr: [0.001, 0.005, 0.01] | 9 |
| ADERH | n_estimators: [256, 128, 512] | n: [10, 20, 30] | 9 |

# F. Algorithm Details

The detailed algorithm for our proposed NRDE is illustrated in Algorithm 1.

---

**Algorithm 1** Training and Testing Procedure of NRDE

---

**Training stage of NRDE and NRDE-M:**
**Input:** $\mathcal{D} = \{\mathbf{x}_1, \mathbf{x}_2, \ldots, \mathbf{x}_n\}$, $\lambda > 0$, training epoch $B$
   **output:** $F_{\mathcal{W}}, \hat{m}$
   Initialize the parameters of flow network $\mathcal{W}$
   **for** $b = 1, \ldots, B$ **do**
      **for** each batch $\hat{\mathcal{D}}$ **do**
         Obtain the flow output $\{F_{\mathcal{W}}(\mathbf{x})\}_{\mathbf{x} \in \hat{\mathbf{D}}}$
         Update parameters $\mathcal{W}$ using (16)
      **end for**
   **end for**
   Obtain $\hat{m}$ using (19)
**Testing stage of NRDE and NRDE-M:**
**Input:** $\mathbf{x}^{\text{new}}, F_{\mathcal{W}}, \hat{m}$
   **output:** $p_{\mathcal{X}}\left(\hat{\mathbf{x}}_{\text{pure}}^{\text{new}}\right)$ or $p_{\bar{\mathcal{X}}}(\hat{\mathbf{x}}_{\text{pure}}^{\text{new}})$
   Obtain $\mathbf{z}^{\text{new}} = F_{\mathcal{W}}(\mathbf{x}^{\text{new}})$
   Obtain pure data $\hat{\mathbf{x}}_{\text{pure}}^{\text{new}}$ using (21)
   Obtain anomaly score: $p_{\mathcal{X}}\left(\hat{\mathbf{x}}_{\text{pure}}^{\text{new}}\right)$ in (9) or $p_{\bar{\mathcal{X}}}(\hat{\mathbf{x}}_{\text{pure}}^{\text{new}})$ in (11)

---

The synthetic data is generated using Algorithm 2. We primarily use Gaussian or uniform distributions to generate data, where the variances of the data sources are significantly larger than those of the noise sources. Specifically, $\mathcal{S}_D = \text{Unif}\left([-10, 50]^m\right), \mathcal{S}_N = \text{Unif}\left([-40, -20]^{d-m}\right), \hat{\mathcal{S}}_D = \text{Unif}\left([10, 30]^m\right), \hat{\mathcal{S}}_N = \text{Unif}\left([-10, 10]^{d-m}\right)$. Both the training and testing normal data are generated using the normal data generative process. For noisy normal data, the data sources are distributed according to $\mathcal{S}_D$, while the noise sources are distributed according to variables distributed in $\mathbf{S}_N$ and are perturbed by $\hat{\mathbf{S}}_N$, introducing anomalies in the noise sources. In the case of anomalous data, the generative process closely resembles that of noisy normal data, where the noise sources are distributed in $\mathcal{S}_N$. However, the data sources are perturbed by variables distributed in $\mathbf{S}_D$ and perturbed by $\hat{\mathbf{S}}_D$, leading to anomalies in the data sources. It is noteworthy that normal

---

**Algorithm 2** Data Generative Process

---

**Normal data generation:**

**Input:** data source distribution $\mathcal{S}_D$, noise source distribution $\mathcal{S}_N$, number of data sources $m$, data dimension $d$, mixing matrix $W \in \mathbb{R}^{d \times d}$

    **output**: normal data $\mathbf{x}_i$

    Obtain data sources $\mathbf{s}_D \in \mathbb{R}^m$ by $\mathbf{s}_D \sim \mathcal{S}_D$, obtain noise sources $\mathbf{s}_N \in \mathbb{R}^{d-m}$ by $\mathbf{s}_N \sim \mathcal{S}_N$

    Generate data using $\mathbf{x} = W[\mathbf{s}_D; \mathbf{s}_N]^T$

**Noisy normal data generation:**

**Input:** number of data $\hat{n}$, data source distribution $\mathcal{S}_D$, noise source distribution $\mathcal{S}_N$, noise perturbation distribution $\hat{\mathcal{S}}_N$, number of data sources $m$, data dimension $d$, mixing matrix $W \in \mathbb{R}^{d \times d}$

    **output**: noisy normal data $\tilde{\mathbf{x}}$

    Obtain data sources $\mathbf{s}_D \in \mathbb{R}^m$ by $\mathbf{s}_D \sim \mathcal{S}_D$, obtain noise sources $\mathbf{s}_N \in \mathbb{R}^{d-m}$ by $\mathbf{s}_N \sim \mathcal{S}_N$, obtain noise perturbation $\hat{\mathbf{s}}_N \in \mathbb{R}^{d-m}$ by $\hat{\mathbf{s}}_N \sim \hat{\mathcal{S}}_N$

    Generate data using $\tilde{\mathbf{x}} = W[\mathbf{s}_D; \mathbf{s}_N + \hat{\mathbf{s}}_N]^T$

**Anomalous data generation:**

**Input:** number of data $\hat{n}$, data source distribution $\mathcal{S}_D$, data perturbation distribution $\hat{\mathcal{S}}_D$, noise source distribution $\mathcal{S}_N$, number of data sources $m$, data dimension $d$, mixing matrix $W \in \mathbb{R}^{d \times d}$

    **output**: anomalous data $\hat{\mathbf{x}}$

    Obtain data sources $\mathbf{s}_D \in \mathbb{R}^m$ by $\mathbf{s}_D \sim \mathcal{S}_D$, obtain data perturbation $\hat{\mathbf{s}}_D \in \mathbb{R}^m$ by $\hat{\mathbf{s}}_D \sim \hat{\mathcal{S}}_D$, obtain noise sources $\mathbf{s}_N \in \mathbb{R}^{d-m}$ by $\mathbf{s}_N \sim \mathcal{S}_N$

    Generate data using $\hat{\mathbf{x}} = W[\mathbf{s}_D + \hat{\mathbf{s}}_D; \mathbf{s}_N]^T$

---

data is different from pure normal data in that normal data also contains noise components, whose magnitude is much smaller than that of noisy normal data.

Data shown in Figure 1 is generated using $d = 100, m = 10$. Note that here $W$ is an orthogonal matrix. Since pure normal data are difficult to obtain in real datasets, the normal samples shown in Figure 1 correspond to noisy normal data in (8) whose noise sources are normal. In contrast, the noisy normal samples in Figure 1 correspond to noisy normal data in (8) whose noise sources are anomalous. All models are trained using 10000 normal data, and tested using 1000 normal data, 1000 noisy normal data, and 1000 anomalous data.

## G. Statistics of Datasets

In our experiments, we evaluate the performance of 14 methods on 47 widely used real-world datasets spanning multiple domains, including healthcare, audio, language processing, and finance, in a popular benchmark for anomaly detection (Han et al., 2022). The statistics of these datasets are shown in Table 8. These datasets encompass a wide range of sample sizes and feature dimensions, from small to large, providing comprehensive metrics and evaluations for the methods.

## H. Experiments on Synthetic and Real Datasets to Verify Assumptions and Motivations

It is noteworthy that verifying the variance-related assumption on real datasets is inherently challenging because the underlying generative process $G$ is not accessible during training. A practical way to indirectly verify our assumptions is to compare NRDE's performance with that of other baselines on the 47 datasets, particularly with RealNVP (normalizing flow), which estimates the density of data with noise. The fact that NRDE outperforms RealNVP on most datasets indicates that our motivation and underlying assumptions hold for these real datasets. In this section, we include several experiments on both synthetic and real datasets to further verify our assumptions and motivations.

### H.1. Performance Results When the Variance Difference Assumption Is Violated

Here, we analyze the performance of NRDE on synthetic datasets where the variance difference is not satisfied. Suppose that the variance of pure data sources is $\sigma_d^2$ and that the variance of noise sources is $\sigma_n^2$. We report the performance results on synthetic datasets with different $\frac{\sigma_d^2}{\sigma_n^2}$ in Table 9. As $\frac{\sigma_d^2}{\sigma_n^2}$ decreases, the performance decline of NRDE supports our assumptions and motivations.

*Table 8.* Statistics of 47 real-world datasets in ADBench.

| Data | # Samples | # Features | # Anomaly | % Anomaly | Category |
|---|---|---|---|---|---|
| ALOI | 49534 | 27 | 1508 | 3.04 | Image |
| annthyroid | 7200 | 6 | 534 | 7.42 | Healthcare |
| backdoor | 95329 | 196 | 2329 | 2.44 | Network |
| breastw | 683 | 9 | 239 | 34.99 | Healthcare |
| campaign | 41188 | 62 | 4640 | 11.27 | Finance |
| cardio | 1831 | 21 | 176 | 9.61 | Healthcare |
| Cardiotocography | 2114 | 21 | 466 | 22.04 | Healthcare |
| celeba | 202599 | 39 | 4547 | 2.24 | Image |
| census | 299285 | 500 | 18568 | 6.20 | Sociology |
| cover | 286048 | 10 | 2747 | 0.96 | Botany |
| donors | 619326 | 10 | 36710 | 5.93 | Sociology |
| fault | 1941 | 27 | 673 | 34.67 | Physical |
| fraud | 284807 | 29 | 492 | 0.17 | Finance |
| glass | 214 | 7 | 9 | 4.21 | Forensic |
| Hepatitis | 80 | 19 | 13 | 16.25 | Healthcare |
| http | 567498 | 3 | 2211 | 0.39 | Web |
| InternetAds | 1966 | 1555 | 368 | 18.72 | Image |
| Ionosphere | 351 | 32 | 126 | 35.90 | Oryctognosy |
| landsat | 6435 | 36 | 1333 | 20.71 | Astronautics |
| letter | 1600 | 32 | 100 | 6.25 | Image |
| Lymphography | 148 | 18 | 6 | 4.05 | Healthcare |
| magic.gamma | 19020 | 10 | 6688 | 35.16 | Physical |
| mammography | 11183 | 6 | 260 | 2.32 | Healthcare |
| mnist | 7603 | 100 | 700 | 9.21 | Image |
| musk | 3062 | 166 | 97 | 3.17 | Chemistry |
| optdigits | 5216 | 64 | 150 | 2.88 | Image |
| PageBlocks | 5393 | 10 | 510 | 9.46 | Document |
| pendigits | 6870 | 16 | 156 | 2.27 | Image |
| Pima | 768 | 8 | 268 | 34.90 | Healthcare |
| satellite | 6435 | 36 | 2036 | 31.64 | Astronautics |
| satimage-2 | 5803 | 36 | 71 | 1.22 | Astronautics |
| shuttle | 49097 | 9 | 3511 | 7.15 | Astronautics |
| skin | 245057 | 3 | 50859 | 20.75 | Image |
| smtp | 95156 | 3 | 30 | 0.03 | Web |
| SpamBase | 4207 | 57 | 1679 | 39.91 | Document |
| speech | 3686 | 400 | 61 | 1.65 | Linguistics |
| Stamps | 340 | 9 | 31 | 9.12 | Document |
| thyroid | 3772 | 6 | 93 | 2.47 | Healthcare |
| vertebral | 240 | 6 | 30 | 12.50 | Biology |
| vowels | 1456 | 12 | 50 | 3.43 | Linguistics |
| Waveform | 3443 | 21 | 100 | 2.90 | Physics |
| WBC | 223 | 9 | 10 | 4.48 | Healthcare |
| WDBC | 367 | 30 | 10 | 2.72 | Healthcare |
| Wilt | 4819 | 5 | 257 | 5.33 | Botany |
| wine | 129 | 13 | 10 | 7.75 | Chemistry |
| WPBC | 198 | 33 | 47 | 23.74 | Healthcare |
| yeast | 1484 | 8 | 507 | 34.16 | Biology |

*Table 9.* AUROC performance of NRDE on synthetic datasets with different $\frac{\sigma_d^2}{\sigma_n^2}$ ratios.

| $\sigma_d^2/\sigma_n^2$ | 9 | 6 | 4 | 2 | 1 | 0.5 |
|---|---|---|---|---|---|---|
| NRDE | 87.3 | 80.9 | 77.8 | 75.4 | 71.0 | 65.8 |

## H.2. Performance Comparison Between NRDE and Optimal Baselines

On synthetic datasets where the number of data sources $m$ is known, we compare the performance of NRDE with KDE-O, DSVDD-O, and KNN-O, which are evaluated on datasets generated by data sources only and with NRDE-O, where only the $m$ sources with the smallest Jacobian row norms are used to obtain the pure data $\mathbf{x}_{\text{pure}}$. The results are shown in Table 10. Since NRDE is an approximation of NRDE-O, the fact that its performance is close to, but not as good as, NRDE-O and other ideal baselines supports our claim and motivation.

*Table 10.* AUROC (%) performance of NRDE and other baselines on the synthetic dataset.

| Method | NRDE | NRDE-O | KDE-O | KNN-O | DSVDD-O |
|---|---|---|---|---|---|
| AUROC | 80.9 | 86.0 | 87.2 | 90.2 | 87.5 |

## H.3. Performance of NRDE Under Opposite Assumptions

If we make an opposite assumption that the variances of data sources should be smaller, then we will select the sources with the largest Jacobian norms as our data sources. This method is denoted as NRDE-CON. The performance of NRDE-CON and NRDE on several datasets is shown in Table 11, where the results support the assumption in our paper.

*Table 11.* AUROC (%) performance of NRDE-CON and NRDE on real datasets.

| | WPBC | Cardiotocography | vertebral | Waveform | Speech |
|---|---|---|---|---|---|
| NRDE-CON | 59.5 | 76.5 | 70.1 | 72.1 | 51.7 |
| NRDE | 64.4 | 87.1 | 82.7 | 86.7 | 58.6 |

## H.4. A Practical Diagnostic to Verify the Variance Difference on Real Data

In this subsection, we provide a simple way to empirically verify the variance difference underlying our assumption as follows. We first train an *unregularized* normalizing flow $F_{\mathcal{W}}$ on the real dataset $\mathcal{D}$ and compute the average absolute Jacobian

$$J = \frac{1}{n} \sum_{\mathbf{x} \in \mathcal{D}} \left| \nabla F_{\mathcal{W}}(\mathbf{x}) \right|.$$

We then take the row norms $\{\|J_i\|\}_{i=1}^d$ and sort them in ascending order, denoted by $\{\|J_{\hat{i}}\|\}_{i=1}^d$.

One simple way to measure the gap is to compute the gap $\Delta_i = \frac{J_{(i\hat{+}1)} - J_{\hat{i}}}{J_{(i\hat{+}1)}}$, since the number of data sources $m$ cannot be obtained, we use the expectation of this gap $\mathbb{E}(\Delta) = \sum_{i=1}^{d-1} \Delta_i/(d-1)$ to measure the variance difference where a large value of $\mathbb{E}(\Delta)$ indicates that the variance difference is pronounced and that NRDE is particularly appropriate in such cases.

We conduct experiments to measure $\mathbb{E}(\Delta)$ on datasets where NRDE shows performance improvements and on datasets where it exhibits performance drops compared to other density-based methods. The results in Table 12 show that datasets with performance improvements tend to have larger values of $\mathbb{E}(\Delta)$, illustrating that $\mathbb{E}(\Delta)$ is an effective diagnostic for quantifying variance differences.

## H.5. Performance of Baselines Trained and Evaluated on Pure Data

To further validate our motivation of isolating noise and the variance assumption in (6), we use the proposed NRDE to generate pure data for both the training and test sets via (21). We then train several simple baseline methods using pure training data and evaluate them on the pure test data, and compare their performance with that obtained using the original data.

*Table 12.* Average gap values on datasets showing significant performance improvement or drop compared to other density-based methods.

| Datasets (Improvement) | $\mathbb{E}(\Delta)$ |
|---|---|
| annthyroid | 0.23 |
| smtp | 0.58 |
| vertebral | 0.16 |
| Pima | 0.35 |
| Cardiotocography | 0.14 |
| **Datasets (Drop)** | $\mathbb{E}(\Delta)$ |
| Ionosphere | 0.04 |
| landsat | 0.03 |
| letter | 0.03 |
| optdigits | 0.07 |
| pendigits | 0.06 |

As shown in Table 13, we compare the experimental results of baseline methods obtained using their default hyperparameter configurations on pure data with those achieved using best-performing hyperparameter configurations on the original data. The results show that substantially better performance is consistently achieved on the pure data, even without extensive hyperparameter tuning. Notably, this improvement persists even on datasets where NRDE itself is less effective (e.g., *landsat* and *magic.gamma*). These observations suggest that in the pure data space, separability between normal and anomalous samples is substantially enhanced, thereby providing empirical support for our variance-related assumption and the effectiveness of the proposed noise isolation strategy.

*Table 13.* AUROC and AUPRC performance of different methods when trained and evaluated on pure data generated by NRDE. The numbers in parentheses denote the performance improvement or degradation compared to training and testing on the original data.

| AUROC | KDE | KNN | ECOD | NRDE |
|---|---|---|---|---|
| annthyroid | 95.0 (+3.6) | 96.2 (+2.1) | 91.4 (+12.7) | 97.3 |
| Cardiotocography | 86.3 (+4.9) | 83.9 (+11.4) | 83.1 (+4.6) | 87.1 |
| landsat | 73.7 (+1.0) | 75.7 (+0.4) | 51.2 (+14.6) | 70.4 |
| magic.gamma | 79.1 (+3.4) | 86.6 (+3.7) | 70.3 (+6.9) | 82.5 |
| mammography | 90.3 (+0.5) | 88.5 (+0.7) | 92.6 (+1.9) | 91.7 |
| Stamps | 96.1 (+1.0) | 94.6 (+2.3) | 92.3 (+5.6) | 96.9 |
| vertebral | 68.5 (+25.0) | 68.4 (+22.7) | 76.4 (+34.6) | 82.8 |
| Waveform | 83.4 (+6.5) | 82.9 (+6.5) | 63.4 (+3.4) | 86.7 |
| Wilt | 79.7 (+42.6) | 72.7 (+7.5) | 70.8 (+30.5) | 83.6 |
| WPBC | 63.3 (+10.8) | 67.1 (+15.4) | 54.9 (+7.9) | 64.4 |
| AUPRC | KDE | KNN | ECOD | NRDE |
| annthyroid | 75.6 (+9.4) | 79.3 (+7.3) | 60.1 (+19.3) | 85.7 |
| Cardiotocography | 78.1 (+6.6) | 80.7 (+16.8) | 75.7 (+10.0) | 77.5 |
| landsat | 55.0 (+0.2) | 58.6 (+0.6) | 33.9 (+6.1) | 58.2 |
| magic.gamma | 83.5 (+3.1) | 88.3 (+2.9) | 74.3 (+6.7) | 81.9 |
| mammography | 50.0 (+8.4) | 45.5 (+3.6) | 57.9 (+3.9) | 59.4 |
| Stamps | 84.0 (+20.3) | 87.5 (+33.3) | 57.1 (+11.9) | 88.3 |
| vertebral | 32.3 (+12.6) | 43.5 (+23.2) | 59.9 (+40.4) | 54.9 |
| Waveform | 33.4 (+5.8) | 32.0 (+6.6) | 8.4 (+0.8) | 14.8 |
| Wilt | 42.2 (+34.8) | 35.1 (+22.2) | 20.1 (+12.0) | 25.8 |
| WPBC | 60.2 (+21.8) | 55.5 (+17.3) | 48.0 (+12.6) | 52.7 |

# I. More Experimental Results

## I.1. Jacobian Row Norm Visualization

In this subsection, we present visualizations of the Jacobian row norms on both synthetic and real-world datasets. Figures 5 and 6 illustrate these results. Notably, even without regularization, the row norms already exhibit clear separability; this distinction becomes even more pronounced when the regularizer is applied.

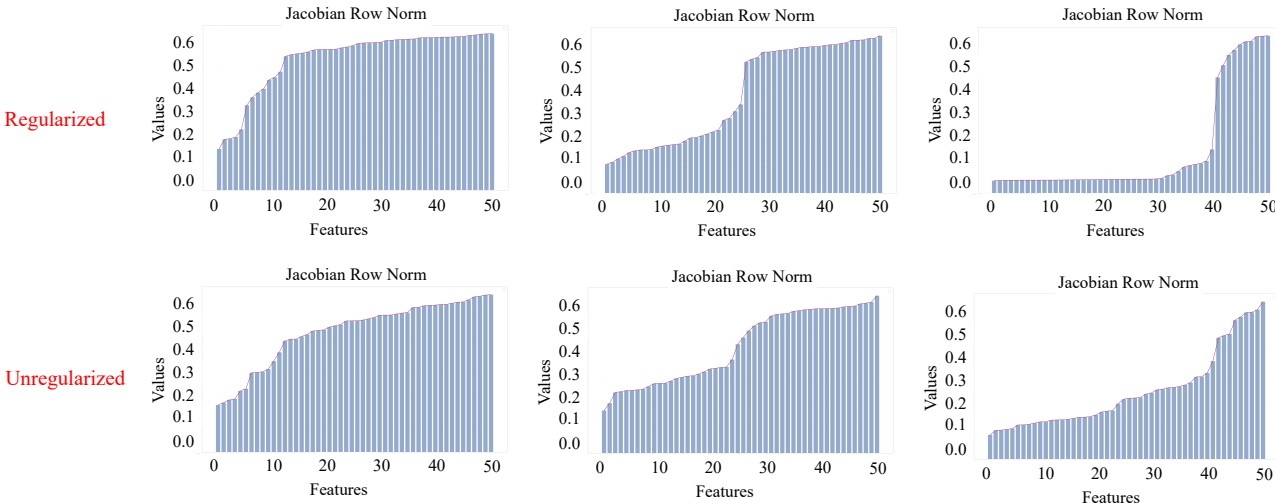

*Figure 5.* Visualization of the Jacobian matrix row norms on several synthetic datasets with 50 total sources. From left to right, the number of noise dimensions is 40, 25, and 10, respectively. The top row corresponds to the regularized case, whereas the bottom row illustrates the unregularized case.

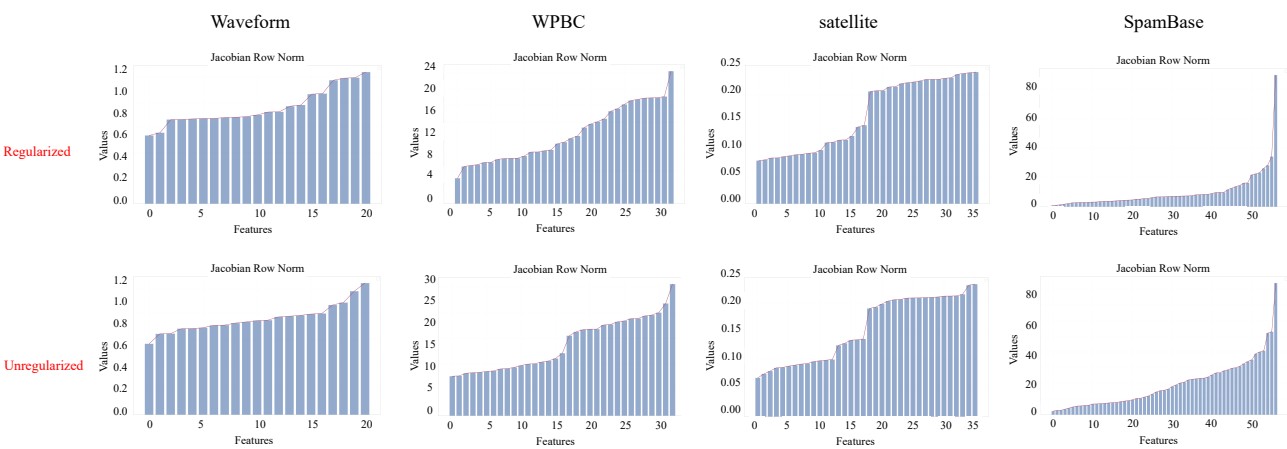

*Figure 6.* Visualization of the Jacobian matrix row norms on several real datasets. The top row corresponds to the regularized case, whereas the bottom row illustrates the unregularized case.

## I.2. Standard Anomaly Detection

In this subsection, we provide the detailed experimental results of AD on 47 real-world datasets. Table 14 shows the detailed AUROC and AUPRC results on 47 datasets. Additionally, box plots of all 19 methods on all the 47 benchmark datasets are provided in Figure 7. Meanwhile, we also provide the experimental results under default hyperparameter setting and under AutoUAD(Dai & Fan, 2025)-tuning setting in Table 15, Table 16 and Table 17, Table 18 respectively.

*Table 14.* Average AUROC (%) and AUPRC (%) with the standard deviation of each method on 47 tabular datasets of ADBench. The best and second-best results are highlighted in red and orange, respectively. Hyperparameter of each methods on each dataset is tuned using grid search.

| AUROC | KDE (1962) | kNN (2000) | LOF (2000) | OC-SVM (2001) | IF (2008) | AE (2006) | DSVDD (2018) | RealNVP (2016) | NAD (2021) | ECOD (2022) | ICL (2022) | SLAD (2023) | DPAD (2024) | MCM (2024) | DTE-C (2024) | TCCM (2025) | ADERH (2025) | NRDE | NRDE-M |
|---|---|---|---|---|---|---|---|---|---|---|---|---|---|---|---|---|---|---|---|
| ALOI | 56.3 ± 0.0 | 66.7 ± 0.0 | 78.6 ± 0.0 | 55.4 ± 0.0 | 53.1 ± 0.0 | 56.7 ± 0.3 | 55.4 ± 0.1 | 56.2 ± 2.5 | 61.3 ± 1.0 | 53.0 ± 0.0 | 58.4 ± 1.3 | 54.7 ± 0.1 | 50.6 ± 0.5 | 59.7 ± 0.0 | 54.5 ± 0.1 | 56.2 ± 0.5 | 55.3 ± 0.0 | 57.2 ± 0.4 | 53.4 ± 0.5 |
| annthyroid | 91.4 ± 0.0 | 94.1 ± 0.0 | 95.4 ± 0.0 | 91.7 ± 0.0 | 93.1 ± 0.0 | 93.5 ± 2.3 | 92.3 ± 1.6 | 96.1 ± 0.5 | 87.7 ± 3.2 | 78.7 ± 0.0 | 83.5 ± 0.5 | 87.3 ± 0.4 | 82.8 ± 4.8 | 98.2 ± 0.9 | 97.9 ± 0.1 | 94.1 ± 0.8 | 91.7 ± 0.4 | 97.3 ± 0.5 | 98.4 ± 0.3 |
| backdoor | 90.4 ± 0.0 | 94.5 ± 0.0 | 95.8 ± 0.0 | 66.8 ± 0.0 | 78.1 ± 0.0 | 94.8 ± 2.0 | 95.1 ± 0.7 | 91.8 ± 0.4 | 95.2 ± 0.1 | 84.6 ± 0.0 | 96.5 ± 0.2 | 74.6 ± 0.1 | 91.9 ± 1.5 | 97.3 ± 0.0 | 91.7 ± 0.0 | 96.1 ± 0.4 | 76.2 ± 1.3 | 92.1 ± 0.2 | 91.7 ± 2.7 |
| breastw | 99.1 ± 0.0 | 99.2 ± 0.0 | 98.4 ± 0.0 | 99.0 ± 0.0 | 99.6 ± 0.0 | 99.2 ± 0.2 | 99.2 ± 0.1 | 98.0 ± 0.0 | 80.1 ± 2.8 | 99.3 ± 0.0 | 92.6 ± 1.4 | 99.3 ± 0.1 | 98.9 ± 0.3 | 99.4 ± 0.0 | 97.0 ± 0.8 | 99.1 ± 0.2 | 98.8 ± 0.1 | 98.9 ± 0.3 | 98.9 ± 0.2 |
| campaign | 77.3 ± 0.0 | 78.5 ± 0.0 | 70.0 ± 0.0 | 77.2 ± 0.0 | 82.3 ± 0.0 | 82.6 ± 0.0 | 77.5 ± 0.4 | 79.6 ± 0.3 | 73.7 ± 0.1 | 78.3 ± 0.0 | 80.7 ± 0.8 | 76.0 ± 0.1 | 55.0 ± 5.0 | 86.4 ± 0.0 | 79.7 ± 1.1 | 79.5 ± 1.3 | 76.1 ± 0.2 | 84.9 ± 1.9 | 79.6 ± 5.4 |
| cardio | 96.6 ± 0.0 | 94.2 ± 0.0 | 95.3 ± 0.0 | 97.5 ± 0.0 | 96.2 ± 0.0 | 92.4 ± 3.2 | 96.4 ± 0.1 | 94.1 ± 0.4 | 81.5 ± 2.3 | 93.4 ± 0.0 | 91.2 ± 0.2 | 90.3 ± 0.0 | 87.9 ± 5.1 | 95.5 ± 0.0 | 93.7 ± 0.5 | 96.0 ± 0.2 | 94.6 ± 0.1 | 95.3 ± 1.1 | 96.1 ± 0.7 |
| Cardiotocography | 81.4 ± 0.0 | 72.5 ± 0.0 | 78.2 ± 0.0 | 83.9 ± 0.0 | 88.0 ± 0.0 | 80.7 ± 2.8 | 83.7 ± 0.5 | 77.9 ± 1.8 | 61.3 ± 5.5 | 78.5 ± 0.0 | 62.9 ± 1.7 | 65.0 ± 0.0 | 72.6 ± 1.3 | 81.3 ± 0.0 | 72.0 ± 2.4 | 77.5 ± 1.5 | 66.4 ± 0.3 | 87.1 ± 2.2 | 83.5 ± 1.4 |
| celeba | 67.3 ± 0.0 | 74.3 ± 0.0 | 50.0 ± 0.0 | 79.0 ± 0.0 | 79.5 ± 0.0 | 75.3 ± 1.2 | 76.5 ± 2.0 | 79.4 ± 0.6 | 55.8 ± 6.1 | 75.7 ± 0.0 | 69.5 ± 0.0 | 63.9 ± 1.0 | 50.0 ± 0.0 | 75.0 ± 0.0 | 81.9 ± 0.1 | 73.4 ± 5.4 | 79.4 ± 0.1 | 83.6 ± 2.1 | 80.3 ± 0.5 |
| census | 72.7 ± 0.0 | 71.9 ± 0.0 | 60.5 ± 0.0 | 70.2 ± 0.0 | 51.1 ± 0.0 | 72.5 ± 0.2 | 52.4 ± 1.9 | 72.8 ± 0.3 | 61.1 ± 2.2 | 65.9 ± 0.0 | 66.8 ± 0.0 | 66.9 ± 1.2 | 50.0 ± 0.0 | 68.1 ± 2.0 | 69.9 ± 0.3 | 67.9 ± 0.4 | 69.5 ± 0.4 | 78.4 ± 1.1 | 69.2 ± 2.6 |
| cover | 95.5 ± 0.0 | 98.5 ± 0.0 | 99.3 ± 0.0 | 98.8 ± 0.0 | 92.1 ± 0.0 | 98.4 ± 0.4 | 49.8 ± 0.5 | 83.8 ± 0.1 | 82.2 ± 7.6 | 92.0 ± 1.1 | 67.4 ± 3.4 | 73.1 ± 5.8 | 50.1 ± 0.0 | 95.1 ± 0.0 | 98.1 ± 0.1 | 98.7 ± 1.0 | 93.1 ± 0.1 | 99.9 ± 0.0 | 99.6 ± 0.1 |
| donors | 97.4 ± 0.0 | 99.9 ± 0.0 | 99.0 ± 0.0 | 93.5 ± 0.0 | 96.0 ± 0.0 | 92.0 ± 1.2 | 97.4 ± 0.4 | 96.6 ± 0.4 | 99.9 ± 0.0 | 88.8 ± 0.0 | 94.0 ± 0.0 | 87.7 ± 4.4 | 67.7 ± 10.9 | 100.0 ± 0.0 | 98.8 ± 0.1 | 99.8 ± 0.1 | 93.8 ± 0.3 | 99.6 ± 0.2 | 98.2 ± 8.2 |
| fault | 81.2 ± 0.0 | 79.2 ± 0.0 | 71.0 ± 0.0 | 61.9 ± 0.0 | 65.3 ± 0.0 | 76.1 ± 0.0 | 70.2 ± 2.0 | 50.8 ± 0.1 | 74.2 ± 0.9 | 47.4 ± 0.0 | 79.2 ± 0.6 | 77.2 ± 0.0 | 74.2 ± 0.9 | 77.1 ± 0.0 | 72.7 ± 1.1 | 73.8 ± 0.4 | 72.3 ± 0.0 | 71.4 ± 2.0 | 69.0 ± 1.8 |
| fraud | 95.8 ± 0.0 | 96.0 ± 0.0 | 94.8 ± 0.0 | 95.6 ± 0.0 | 95.1 ± 0.0 | 95.6 ± 0.1 | 54.3 ± 1.3 | 54.4 ± 0.1 | 93.6 ± 0.6 | 94.9 ± 0.0 | 93.3 ± 0.4 | 94.6 ± 0.3 | 54.8 ± 4.8 | 94.4 ± 0.0 | 94.6 ± 0.4 | 93.9 ± 0.1 | 95.2 ± 0.1 | 95.7 ± 0.6 | 92.6 ± 0.1 |
| glass | 86.8 ± 0.0 | 89.3 ± 0.0 | 84.0 ± 0.0 | 79.3 ± 0.0 | 84.1 ± 0.0 | 89.5 ± 0.5 | 83.0 ± 2.3 | 81.5 ± 1.3 | 97.8 ± 0.9 | 69.8 ± 0.0 | 93.6 ± 1.0 | 78.7 ± 2.4 | 84.1 ± 4.4 | 89.3 ± 0.0 | 86.7 ± 2.2 | 88.9 ± 0.3 | 90.1 ± 0.1 | 89.2 ± 0.4 | 91.3 ± 2.1 |
| Hepatitis | 84.8 ± 0.0 | 85.1 ± 0.0 | 86.0 ± 0.0 | 85.1 ± 0.0 | 87.6 ± 0.0 | 83.9 ± 0.7 | 83.3 ± 1.8 | 59.4 ± 0.8 | 72.6 ± 2.5 | 71.5 ± 0.0 | 70.2 ± 2.4 | 80.3 ± 1.5 | 72.4 ± 3.6 | 72.4 ± 0.0 | 84.8 ± 0.9 | 85.9 ± 1.0 | 84.4 ± 0.2 | 86.7 ± 1.5 | 85.1 ± 2.4 |
| http | 100.0 ± 0.0 | 100.0 ± 0.0 | 99.5 ± 0.0 | 100.0 ± 0.0 | 99.4 ± 0.0 | 99.9 ± 0.0 | 99.8 ± 0.0 | 99.6 ± 0.0 | 96.7 ± 0.2 | 97.8 ± 0.0 | 74.9 ± 24.8 | 99.9 ± 0.0 | 75.9 ± 23.9 | 100.0 ± 0.0 | 99.6 ± 0.1 | 100.0 ± 0.0 | 100.0 ± 0.0 | 100.0 ± 0.0 | 100.0 ± 0.0 |
| InternetAds | 85.6 ± 0.0 | 76.4 ± 0.0 | 83.8 ± 0.0 | 74.9 ± 0.0 | 59.9 ± 0.0 | 88.7 ± 0.0 | 77.9 ± 1.4 | 81.2 ± 0.3 | 86.7 ± 0.4 | 68.9 ± 0.0 | 79.3 ± 1.2 | 90.0 ± 0.4 | 79.2 ± 0.2 | 86.9 ± 0.0 | 87.0 ± 0.6 | 88.0 ± 0.0 | 71.1 ± 0.3 | 83.9 ± 0.3 | 74.7 ± 0.2 |
| Ionosphere | 97.4 ± 0.0 | 98.0 ± 0.0 | 96.3 ± 0.0 | 96.9 ± 0.0 | 91.9 ± 0.0 | 97.8 ± 0.3 | 96.7 ± 0.1 | 93.8 ± 0.2 | 96.6 ± 0.1 | 75.5 ± 0.0 | 96.7 ± 1.1 | 96.5 ± 0.1 | 89.3 ± 0.0 | 96.5 ± 0.0 | 96.7 ± 0.6 | 96.8 ± 0.1 | 96.0 ± 0.1 | 95.3 ± 0.3 | 87.6 ± 8.6 |
| landsat | 72.7 ± 0.0 | 75.3 ± 0.0 | 77.8 ± 0.0 | 47.0 ± 0.0 | 68.1 ± 0.0 | 63.1 ± 1.7 | 68.4 ± 1.1 | 59.7 ± 2.2 | 72.6 ± 0.3 | 36.6 ± 0.0 | 79.3 ± 0.5 | 72.6 ± 0.2 | 70.8 ± 0.5 | 64.7 ± 0.0 | 61.0 ± 0.0 | 61.2 ± 1.2 | 60.2 ± 0.4 | 70.4 ± 5.5 | 74.2 ± 2.5 |
| letter | 91.8 ± 0.0 | 87.2 ± 0.0 | 92.2 ± 0.0 | 61.0 ± 0.0 | 65.4 ± 0.0 | 88.5 ± 0.2 | 67.2 ± 3.7 | 83.1 ± 0.6 | 93.2 ± 1.4 | 56.0 ± 0.0 | 87.7 ± 0.2 | 90.2 ± 0.1 | 77.3 ± 1.2 | 89.5 ± 0.0 | 89.9 ± 0.6 | 93.9 ± 0.2 | 76.7 ± 0.4 | 81.6 ± 0.4 | 68.2 ± 1.4 |
| Lymphography | 98.6 ± 0.0 | 98.6 ± 0.0 | 98.6 ± 0.0 | 98.4 ± 0.0 | 98.1 ± 0.0 | 98.7 ± 0.1 | 98.7 ± 0.1 | 94.3 ± 0.5 | 92.4 ± 2.0 | 98.5 ± 0.0 | 99.5 ± 0.0 | 98.7 ± 0.1 | 98.6 ± 0.0 | 99.8 ± 0.0 | 98.5 ± 0.1 | 98.6 ± 0.0 | 98.6 ± 0.0 | 98.7 ± 1.0 | 98.9 ± 0.7 |
| magic.gamma | 75.7 ± 0.0 | 82.9 ± 0.0 | 84.0 ± 0.0 | 73.8 ± 0.0 | 77.2 ± 0.0 | 84.6 ± 0.2 | 80.7 ± 0.1 | 79.6 ± 0.5 | 78.0 ± 2.8 | 63.4 ± 0.0 | 78.5 ± 0.1 | 70.6 ± 0.8 | 76.6 ± 1.4 | 83.1 ± 0.0 | 88.1 ± 0.7 | 83.5 ± 0.5 | 70.2 ± 0.1 | 82.5 ± 0.1 | 80.3 ± 1.1 |
| mammography | 89.8 ± 0.0 | 87.8 ± 0.0 | 85.0 ± 0.0 | 88.9 ± 0.0 | 89.0 ± 0.0 | 91.5 ± 0.2 | 89.6 ± 0.2 | 89.5 ± 0.4 | 76.1 ± 0.3 | 90.7 ± 0.0 | 76.7 ± 1.8 | 76.0 ± 0.4 | 78.2 ± 8.9 | 91.6 ± 0.0 | 89.3 ± 0.8 | 86.3 ± 0.5 | 87.5 ± 0.0 | 91.7 ± 0.1 | 92.4 ± 0.7 |
| mnist | 95.1 ± 0.0 | 94.3 ± 0.0 | 94.1 ± 0.0 | 91.0 ± 0.0 | 73.5 ± 0.0 | 94.4 ± 0.5 | 91.8 ± 0.6 | 92.4 ± 0.4 | 88.2 ± 0.1 | 74.7 ± 0.0 | 86.2 ± 0.2 | 89.9 ± 0.2 | 82.2 ± 1.6 | 96.7 ± 0.0 | 90.7 ± 0.4 | 93.8 ± 0.1 | 91.1 ± 0.2 | 93.3 ± 0.4 | 88.9 ± 2.0 |
| musk | 100.0 ± 0.0 | 100.0 ± 0.0 | 100.0 ± 0.0 | 100.0 ± 0.0 | 100.0 ± 0.0 | 100.0 ± 0.0 | 100.0 ± 0.0 | 99.4 ± 0.3 | 100.0 ± 0.0 | 95.8 ± 0.0 | 100.0 ± 0.0 | 100.0 ± 0.0 | 100.0 ± 0.0 | 100.0 ± 0.0 | 100.0 ± 0.0 | 100.0 ± 0.0 | 100.0 ± 0.0 | 100.0 ± 0.0 | 83.0 ± 28.6 |
| optdigits | 97.4 ± 0.0 | 94.9 ± 0.0 | 98.7 ± 0.0 | 62.6 ± 0.0 | 86.8 ± 0.0 | 93.7 ± 0.0 | 78.4 ± 1.3 | 94.2 ± 0.7 | 98.8 ± 0.6 | 60.4 ± 0.0 | 96.3 ± 0.5 | 93.7 ± 2.4 | 59.9 ± 10.0 | 96.1 ± 0.0 | 89.0 ± 0.8 | 92.1 ± 2.2 | 63.3 ± 0.2 | 97.7 ± 0.8 | 90.5 ± 5.0 |
| PageBlocks | 95.0 ± 0.0 | 96.2 ± 0.0 | 97.5 ± 0.0 | 96.1 ± 0.0 | 94.1 ± 0.0 | 97.0 ± 0.5 | 96.4 ± 0.1 | 92.0 ± 0.9 | 92.9 ± 0.1 | 91.4 ± 0.0 | 91.7 ± 0.9 | 90.0 ± 1.6 | 90.8 ± 4.1 | 79.8 ± 0.0 | 96.7 ± 0.3 | 95.7 ± 1.4 | 94.4 ± 0.0 | 93.5 ± 0.5 | 97.2 ± 0.5 |
| pendigits | 99.8 ± 0.0 | 99.8 ± 0.0 | 99.7 ± 0.0 | 97.3 ± 0.0 | 97.5 ± 0.0 | 99.5 ± 0.0 | 96.9 ± 1.9 | 98.5 ± 0.6 | 97.1 ± 1.1 | 92.7 ± 0.0 | 97.6 ± 1.2 | 94.4 ± 1.9 | 97.6 ± 1.0 | 98.2 ± 0.0 | 98.3 ± 0.3 | 98.1 ± 0.3 | 97.6 ± 0.2 | 99.3 ± 0.2 | 99.5 ± 0.2 |
| Pima | 78.5 ± 0.0 | 79.0 ± 0.0 | 77.3 ± 0.0 | 76.5 ± 0.0 | 76.9 ± 0.0 | 76.2 ± 0.8 | 75.7 ± 0.4 | 79.3 ± 0.6 | 63.5 ± 1.2 | 61.5 ± 0.0 | 70.2 ± 0.3 | 64.7 ± 7.2 | 75.0 ± 0.7 | 70.3 ± 0.0 | 73.1 ± 1.5 | 74.9 ± 3.5 | 77.2 ± 0.2 | 80.0 ± 3.3 | 81.1 ± 0.8 |
| satellite | 87.3 ± 0.0 | 87.3 ± 0.0 | 86.8 ± 0.0 | 75.4 ± 0.0 | 83.2 ± 0.0 | 82.0 ± 0.2 | 87.8 ± 0.8 | 83.6 ± 1.3 | 79.7 ± 0.2 | 58.3 ± 0.0 | 90.1 ± 0.2 | 87.9 ± 0.3 | 86.5 ± 0.5 | 81.7 ± 0.0 | 86.6 ± 0.6 | 83.0 ± 0.6 | 81.1 ± 0.2 | 85.8 ± 1.2 | 83.6 ± 3.1 |
| satimage-2 | 99.9 ± 0.0 | 99.9 ± 0.0 | 99.6 ± 0.0 | 99.7 ± 0.0 | 99.4 ± 0.0 | 99.9 ± 0.0 | 99.9 ± 0.0 | 98.9 ± 0.0 | 88.9 ± 0.1 | 96.6 ± 0.0 | 99.4 ± 0.1 | 99.8 ± 0.1 | 99.2 ± 0.3 | 99.3 ± 0.0 | 99.0 ± 0.2 | 99.4 ± 0.2 | 99.9 ± 0.0 | 99.8 ± 0.0 | 99.4 ± 0.2 |
| shuttle | 99.8 ± 0.0 | 99.9 ± 0.0 | 100.0 ± 0.0 | 99.9 ± 0.0 | 99.7 ± 0.0 | 99.9 ± 0.0 | 99.8 ± 0.0 | 99.8 ± 0.0 | 100.0 ± 0.0 | 99.3 ± 0.0 | 99.8 ± 0.1 | 99.9 ± 0.0 | 99.1 ± 0.2 | 99.9 ± 0.0 | 99.7 ± 0.0 | 100.0 ± 0.0 | 99.8 ± 0.0 | 99.9 ± 0.0 | 99.8 ± 0.0 |
| skin | 89.1 ± 0.0 | 99.8 ± 0.0 | 94.4 ± 0.0 | 90.3 ± 0.0 | 90.0 ± 0.0 | 84.2 ± 0.2 | 70.4 ± 0.4 | 90.0 ± 0.1 | 94.0 ± 2.6 | 48.8 ± 0.0 | 50.0 ± 0.0 | 92.1 ± 1.8 | 87.7 ± 11.8 | 92.1 ± 0.0 | 92.3 ± 0.0 | 86.0 ± 2.8 | 91.2 ± 0.1 | 94.9 ± 0.6 | 96.2 ± 0.4 |
| smtp | 88.2 ± 0.0 | 93.6 ± 0.0 | 94.9 ± 0.0 | 87.2 ± 0.0 | 93.5 ± 0.0 | 94.6 ± 0.0 | 91.9 ± 0.1 | 93.4 ± 0.1 | 95.3 ± 0.2 | 87.9 ± 0.0 | 65.0 ± 15.0 | 96.2 ± 1.2 | 84.1 ± 0.8 | 88.4 ± 0.0 | 95.5 ± 0.0 | 91.8 ± 4.4 | 93.0 ± 0.3 | 90.6 ± 1.1 | 96.3 ± 0.3 |
| SpamBase | 86.0 ± 0.0 | 84.5 ± 0.0 | 82.4 ± 0.0 | 81.7 ± 0.0 | 79.1 ± 0.0 | 82.6 ± 0.4 | 83.3 ± 0.1 | 80.1 ± 0.3 | 80.9 ± 0.5 | 66.0 ± 0.0 | 82.3 ± 0.2 | 84.8 ± 0.8 | 71.2 ± 3.1 | 73.6 ± 0.0 | 85.4 ± 0.7 | 86.4 ± 0.0 | 80.8 ± 0.1 | 84.5 ± 0.4 | 82.0 ± 0.3 |
| speech | 71.4 ± 0.0 | 48.5 ± 0.0 | 57.7 ± 0.0 | 46.3 ± 0.0 | 49.2 ± 0.0 | 47.5 ± 0.0 | 47.6 ± 1.5 | 50.0 ± 0.0 | 58.5 ± 1.0 | 46.1 ± 0.0 | 54.9 ± 4.6 | 55.4 ± 0.3 | 54.3 ± 3.5 | 52.7 ± 0.0 | 52.9 ± 2.0 | 55.1 ± 0.3 | 49.2 ± 0.1 | 58.7 ± 2.7 | 62.8 ± 0.4 |
| Stamps | 95.1 ± 0.0 | 92.3 ± 0.0 | 91.9 ± 0.0 | 92.9 ± 0.0 | 95.2 ± 0.0 | 93.5 ± 0.2 | 90.6 ± 0.4 | 93.6 ± 1.4 | 79.7 ± 2.7 | 86.7 ± 0.0 | 89.0 ± 0.6 | 81.5 ± 1.1 | 92.5 ± 1.5 | 89.8 ± 0.0 | 89.6 ± 2.2 | 90.8 ± 1.2 | 91.9 ± 0.1 | 96.9 ± 1.6 | 97.4 ± 1.6 |
| thyroid | 98.3 ± 0.0 | 98.5 ± 0.0 | 98.7 ± 0.0 | 98.3 ± 0.0 | 99.4 ± 0.0 | 98.6 ± 0.2 | 98.4 ± 0.0 | 98.6 ± 0.1 | 75.6 ± 1.6 | 97.7 ± 0.0 | 85.8 ± 0.9 | 95.5 ± 1.3 | 98.0 ± 0.6 | 98.5 ± 0.0 | 99.2 ± 0.0 | 98.3 ± 0.4 | 98.3 ± 0.1 | 99.1 ± 0.2 | 99.4 ± 0.3 |
| vertebral | 43.5 ± 0.0 | 45.7 ± 0.0 | 58.0 ± 0.0 | 59.3 ± 0.0 | 49.6 ± 0.0 | 55.8 ± 0.7 | 50.9 ± 1.3 | 53.6 ± 4.8 | 61.5 ± 4.7 | 41.8 ± 0.0 | 60.8 ± 0.7 | 46.2 ± 3.4 | 48.3 ± 9.2 | 53.0 ± 0.0 | 66.3 ± 3.5 | 69.3 ± 5.0 | 36.7 ± 0.2 | 82.8 ± 4.3 | 80.0 ± 0.7 |
| vowels | 96.5 ± 0.0 | 97.3 ± 0.0 | 96.9 ± 0.0 | 85.0 ± 0.0 | 75.8 ± 0.0 | 97.8 ± 0.8 | 87.5 ± 0.0 | 90.4 ± 1.1 | 98.9 ± 0.4 | 59.5 ± 0.0 | 98.5 ± 0.3 | 95.8 ± 1.5 | 94.8 ± 0.0 | 95.9 ± 0.0 | 99.1 ± 0.2 | 97.0 ± 1.3 | 95.4 ± 0.0 | 88.7 ± 1.1 | 87.6 ± 2.5 |
| Waveform | 76.9 ± 0.0 | 76.4 ± 0.0 | 76.7 ± 0.0 | 72.7 ± 0.0 | 73.6 ± 0.0 | 71.6 ± 0.8 | 73.1 ± 2.9 | 72.5 ± 1.6 | 75.3 ± 1.0 | 60.0 ± 0.0 | 64.3 ± 1.5 | 52.2 ± 0.6 | 82.3 ± 1.9 | 77.8 ± 0.0 | 64.9 ± 0.1 | 75.7 ± 1.8 | 77.4 ± 0.4 | 86.7 ± 1.3 | 80.2 ± 2.8 |
| WBC | 99.5 ± 0.0 | 99.5 ± 0.0 | 99.5 ± 0.0 | 99.1 ± 0.0 | 100.0 ± 0.0 | 99.9 ± 0.0 | 99.5 ± 0.0 | 99.3 ± 0.0 | 67.3 ± 4.5 | 99.0 ± 0.0 | 85.7 ± 1.4 | 99.2 ± 0.5 | 99.4 ± 0.5 | 99.5 ± 0.0 | 98.2 ± 0.4 | 98.1 ± 0.6 | 99.0 ± 0.0 | 99.7 ± 0.1 | 99.6 ± 0.1 |
| WDBC | 99.4 ± 0.0 | 99.1 ± 0.0 | 99.7 ± 0.0 | 99.6 ± 0.0 | 99.9 ± 0.0 | 99.9 ± 0.0 | 99.5 ± 0.1 | 99.3 ± 0.0 | 41.5 ± 4.5 | 97.8 ± 0.0 | 95.1 ± 2.0 | 99.1 ± 0.1 | 99.7 ± 0.0 | 100.0 ± 0.0 | 99.1 ± 0.0 | 99.3 ± 0.1 | 99.4 ± 0.1 | 99.7 ± 0.1 | 99.2 ± 0.5 |
| Wilt | 37.1 ± 0.0 | 65.2 ± 0.0 | 84.0 ± 0.0 | 35.5 ± 0.0 | 58.5 ± 0.0 | 78.5 ± 0.3 | 64.7 ± 2.7 | 59.9 ± 0.5 | 83.4 ± 2.7 | 40.3 ± 0.0 | 77.0 ± 0.5 | 73.1 ± 6.7 | 73.8 ± 5.7 | 86.6 ± 0.0 | 91.7 ± 0.4 | 96.3 ± 0.0 | 42.5 ± 0.5 | 83.6 ± 0.1 | 83.5 ± 2.1 |
| wine | 92.2 ± 0.0 | 93.3 ± 0.0 | 92.2 ± 0.0 | 92.2 ± 0.0 | 93.8 ± 0.0 | 92.5 ± 2.7 | 90.2 ± 4.2 | 92.6 ± 1.6 | 97.4 ± 1.8 | 73.0 ± 0.0 | 89.2 ± 3.7 | 95.0 ± 4.0 | 84.2 ± 7.7 | 100.0 ± 0.0 | 94.7 ± 1.7 | 94.1 ± 0.8 | 92.8 ± 0.1 | 99.8 ± 0.3 | 99.4 ± 0.2 |
| WPBC | 52.5 ± 0.0 | 51.7 ± 0.0 | 53.5 ± 0.0 | 51.2 ± 0.0 | 54.6 ± 0.0 | 52.6 ± 1.2 | 49.8 ± 3.5 | 58.0 ± 1.0 | 57.5 ± 0.9 | 47.0 ± 0.0 | 57.8 ± 0.8 | 49.8 ± 1.8 | 51.0 ± 4.8 | 57.3 ± 0.0 | 53.8 ± 1.3 | 51.2 ± 0.6 | 50.1 ± 0.0 | 77.6e 5.0 | 64.7 ± 2.5 |
| yeast | 43.2 ± 0.0 | 47.6 ± 0.0 | 49.5 ± 0.0 | 48.3 ± 0.0 | 47.4 ± 0.0 | 50.0 ± 0.2 | 50.9 ± 0.3 | 51.0 ± 1.4 | 60.2 ± 2.3 | 45.3 ± 0.0 | 57.6 ± 3.4 | 55.3 ± 1.3 | 55.3 ± 4.2 | 50.1 ± 0.0 | 52.0 ± 2.4 | 59.0 ± 1.0 | 40.1 ± 0.2 | 61.2 ± 2.5 | 63.4 ± 0.9 |
| Avg. AUROC | 85.3 | 86.1 | 86.3 | 81.1 | 82.1 | 85.9 | 81.3 | 82.3 | 81.0 | 74.2 | 81.2 | 82.0 | 77.8 | 85.9 | 86.2 | 86.7 | 81.7 | 88.5 | 86.7 |
| Avg. Rank | 8.18 | 7.44 | 7.82 | 11.53 | 10.93 | 7.86 | 10.68 | 11.50 | 11.02 | 16.18 | 11.34 | 11.61 | 13.76 | 7.71 | 9.12 | 8.37 | 11.83 | 5.79 | 7.34 |
| *p*-value | 0.037 | 0.041 | 0.094 | 0.000 | 0.000 | 0.007 | 0.000 | 0.000 | 0.000 | 0.000 | 0.000 | 0.000 | 0.000 | 0.010 | 0.015 | 0.037 | 0.000 | - | 0.008 |
| **AUPRC** | | | | | | | | | | | | | | | | | | | |
| ALOI | 10.5 ± 0.0 | 11.4 ± 0.0 | 21.1 ± 0.0 | 7.5 ± 0.0 | 6.3 ± 0.0 | 7.9 ± 0.0 | 7.2 ± 0.0 | 7.1 ± 0.0 | 11.1 ± 0.4 | 6.4 ± 0.0 | 9.7 ± 0.8 | 7.1 ± 0.1 | 6.5 ± 0.3 | 9.2 ± 0.0 | 7.0 ± 0.1 | 8.3 ± 0.0 | 7.2 ± 0.1 | 7.3 ± 0.1 | 7.1 ± 0.1 |
| annthyroid | 66.2 ± 0.0 | 72.0 ± 0.0 | 77.0 ± 0.0 | 66.9 ± 0.0 | 65.8 ± 0.0 | 69.3 ± 7.3 | 69.5 ± 3.0 | 77.0 ± 2.8 | 43.4 ± 7.1 | 40.8 ± 0.0 | 48.1 ± 2.6 | 60.1 ± 0.5 | 54.7 ± 6.6 | 85.5 ± 0.0 | 84.3 ± 0.3 | 73.8 ± 0.8 | 65.1 ± 1.3 | 75.6 ± 3.0 | 85.7 ± 2.1 |
| backdoor | 44.6 ± 0.0 | 54.0 ± 0.0 | 70.4 ± 0.0 | 8.3 ± 0.0 | 10.1 ± 0.0 | 87.4 ± 0.0 | 75.9 ± 1.2 | 77.9 ± 5.8 | 88.9 ± 9.4 | 16.7 ± 0.0 | 88.5 ± 0.2 | 12.5 ± 1.1 | 77.2 ± 4.4 | 88.9 ± 0.0 | 62.1 ± 0.1 | 84.1 ± 0.5 | 12.3 ± 1.2 | 75.2 ± 1.9 | 72.8 ± 1.1 |
| breastw | 99.1 ± 0.0 | 99.1 ± 0.0 | 98.0 ± 0.0 | 98.8 ± 0.0 | 99.6 ± 0.0 | 99.2 ± 0.2 | 99.1 ± 0.1 | 96.7 ± 0.1 | 71.2 ± 4.5 | 99.3 ± 0.0 | 89.5 ± 1.0 | 99.3 ± 0.2 | 98.7 ± 0.4 | 99.4 ± 0.0 | 94.4 ± 2.9 | 99.0 ± 0.2 | 98.2 ± 0.1 | 98.7 ± 0.2 | 98.8 ± 0.5 |
| campaign | 47.7 ± 0.0 | 50.1 ± 0.0 | 41.9 ± 0.0 | 48.8 ± 0.0 | 58.1 ± 0.0 | 52.9 ± 0.7 | 47.6 ± 0.8 | 50.3 ± 0.1 | 41.4 ± 0.7 | 50.0 ± 0.0 | 49.5 ± 1.0 | 47.8 ± 0.3 | 24.8 ± 4.5 | 55.6 ± 0.0 | 47.4 ± 1.8 | 50.5 ± 1.0 | 48.8 ± 0.4 | 59.7 ± 1.3 | 55.1 ± 4.2 |
| cardio | 83.7 ± 0.0 | 78.0 ± 0.0 | 77.4 ± 0.0 | 86.2 ± 0.0 | 81.5 ± 0.0 | 74.7 ± 5.9 | 85.7 ± 1.3 | 71.0 ± 2.6 | 47.6 ± 6.6 | 70.9 ± 0.0 | 74.4 ± 1.1 | 70.6 ± 6.9 | 82.6 ± 0.0 | 69.7 ± 1.8 | 85.6 ± 0.7 | 81.1 ± 0.2 | 74.1 ± 1.3 | 80.5 ± 5.4 |
| Cardiotocography | 71.5 ± 0.0 | 63.9 ± 0.0 | 67.2 ± 0.0 | 75.6 ± 0.0 | 77.8 ± 0.0 | 71.1 ± 3.1 | 75.3 ± 2.0 | 62.6 ± 2.1 | 44.7 ± 2.2 | 65.7 ± 0.0 | 57.7 ± 1.1 | 59.2 ± 0.1 | 65.1 ± 0.3 | 68.1 ± 0.0 | 61.1 ± 1.9 | 69.9 ± 1.4 | 61.1 ± 0.3 | 79.5 ± 1.8 | 77.5 ± 3.7 |
| celeba | 8.8 ± 0.0 | 12.6 ± 0.0 | 4.3 ± 0.0 | 20.4 ± 0.0 | 18.2 ± 0.0 | 12.5 ± 0.0 | 10.5 ± 1.3 | 13.1 ± 0.6 | 4.8 ± 0.8 | 17.2 ± 0.0 | 8.9 ± 0.0 | 7.0 ± 0.2 | 4.4 ± 0.0 | 12.3 ± 0.0 | 15.8 ± 0.0 | 13.3 ± 6.0 | 18.6 ± 0.3 | 13.9 ± 1.4 | 13.4 ± 0.2 |
| census | 22.5 ± 0.0 | 21.2 ± 0.0 | 14.3 ± 0.0 | 20.5 ± 0.0 | 12.5 ± 0.0 | 22.4 ± 0.1 | 12.1 ± 0.3 | 20.5 ± 0.6 | 17.2 ± 0.9 | 15.5 ± 0.0 | 17.4 ± 0.0 | 19.2 ± 0.9 | 11.7 ± 0.0 | 18.8 ± 2.0 | 18.4 ± 0.8 | 17.9 ± 0.3 | 19.7 ± 0.2 | 28.4 ± 1.0 | 18.6 ± 0.9 |
| cover | 34.2 ± 0.0 | 72.0 ± 0.0 | 84.8 ± 0.0 | 48.2 ± 0.0 | 19.5 ± 0.0 | 59.1 ± 0.9 | 2.2 ± 0.0 | 9.3 ± 1.8 | 46.4 ± 8.6 | 18.4 ± 0.0 | 9.0 ± 3.7 | 4.6 ± 1.2 | 2.0 ± 0.0 | 21.1 ± 0.0 | 71.8 ± 0.7 | 93.4 ± 0.1 | 16.2 ± 0.1 | 96.5 ± 0.0 | 84.6 ± 6.8 |
| donors | 70.9 ± 0.0 | 96.8 ± 0.0 | 89.0 ± 0.0 | 46.6 ± 0.0 | 62.3 ± 0.0 | 45.1 ± 2.2 | 81.9 ± 2.1 | 66.2 ± 1.1 | 98.0 ± 0.5 | 41.2 ± 0.0 | 88.5 ± 2.3 | 45.0 ± 14.5 | 42.4 ± 19.2 | 98.7 ± 0.0 | 77.4 ± 0.5 | 98.0 ± 0.6 | 48.5 ± 1.0 | 94.8 ± 0.3 | 79.5 ± 2.1 |
| fault | 79.8 ± 0.0 | 76.0 ± 0.0 | 68.4 ± 0.0 | 63.8 ± 0.0 | 63.3 ± 0.0 | 75.7 ± 0.1 | 69.1 ± 2.1 | 51.5 ± 0.0 | 72.2 ± 0.9 | 49.4 ± 0.0 | 77.2 ± 0.8 | 77.0 ± 0.1 | 74.0 ± 0.4 | 75.5 ± 0.0 | 72.6 ± 0.8 | 73.1 ± 0.4 | 70.7 ± 0.0 | 72.9 ± 1.2 | 71.1 ± 0.9 |
| fraud | 36.9 ± 0.0 | 35.3 ± 0.0 | 16.9 ± 0.0 | 31.7 ± 0.0 | 24.3 ± 0.0 | 65.2 ± 0.1 | 27.6 ± 0.9 | 3.8 ± 0.0 | 69.4 ± 0.8 | 31.5 ± 0.0 | 23.7 ± 3.7 | 36.8 ± 0.3 | 6.3 ± 5.9 | 63.0 ± 0.0 | 73.1 ± 0.2 | 54.8 ± 2.6 | 27.1 ± 0.5 | 59.7 ± 5.5 | 20.2 ± 10.5 |
| glass | 27.9 ± 0.0 | 29.9 ± 0.0 | 32.3 ± 0.0 | 20.8 ± 0.0 | 28.4 ± 0.0 | 33.6 ± 0.1 | 25.7 ± 2.4 | 24.5 ± 1.4 | 81.4 ± 10.5 | 25.1 ± 0.0 | 43.3 ± 2.5 | 23.8 ± 4.7 | 25.2 ± 4.2 | 32.5 ± 0.0 | 38.9 ± 8.5 | 30.3 ± 0.6 | 31.9 ± 0.2 | 30.8 ± 3.0 | 43.4 ± 5.7 |
| Hepatitis | 62.6 ± 0.0 | 63.3 ± 0.0 | 64.1 ± 0.0 | 62.8 ± 0.0 | 68.0 ± 0.0 | 62.7 ± 1.2 | 67.1 ± 1.2 | 32.0 ± 0.4 | 46.7 ± 1.4 | 40.2 ± 0.0 | 55.7 ± 2.8 | 55.2 ± 4.6 | 58.2 ± 3.2 | 55.1 ± 0.0 | 64.6 ± 1.2 | 64.4 ± 3.9 | 56.9 ± 0.3 | 61.5 ± 1.8 | 67.2 ± 9.1 |
| http | 99.2 ± 0.0 | 99.9 ± 0.0 | 58.7 ± 0.0 | 99.5 ± 0.0 | 53.8 ± 0.0 | 91.3 ± 0.5 | 99.6 ± 0.1 | 55.1 ± 0.0 | 18.4 ± 1.2 | 25.2 ± 0.0 | 49.9 ± 49.1 | 91.3 ± 0.1 | 51.2 ± 46.6 | 92.2 ± 0.0 | 62.5 ± 6.3 | 97.3 ± 2.7 | 99.7 ± 0.2 | 97.6 ± 1.5 | 94.3 ± 2.4 |
| InternetAds | 82.0 ± 0.0 | 65.3 ± 0.0 | 70.1 ± 0.0 | 69.0 ± 0.0 | 46.7 ± 0.0 | 86.4 ± 0.1 | 65.9 ± 1.4 | 56.5 ± 0.5 | 81.8 ± 0.4 | 62.8 ± 0.0 | 61.2 ± 2.3 | 83.8 ± 0.8 | 68.4 ± 2.4 | 84.0 ± 0.0 | 80.6 ± 3.6 | 84.8 ± 0.1 | 62.1 ± 0.2 | 79.8 ± 1.7 | 65.3 ± 0.2 |
| Ionosphere | 97.0 ± 0.0 | 98.3 ± 0.0 | 96.9 ± 0.0 | 97.7 ± 0.0 | 92.4 ± 0.0 | 98.2 ± 0.3 | 97.3 ± 0.0 | 90.9 ± 0.3 | 97.4 ± 0.1 | 77.9 ± 0.0 | 96.7 ± 1.1 | 97.0 ± 0.1 | 90.7 ± 0.4 | 97.2 ± 0.0 | 97.2 ± 0.5 | 97.5 ± 0.4 | 96.8 ± 0.1 | 95.2 ± 0.5 | 91.6 ± 6.1 |
| landsat | 54.8 ± 0.0 | 58.0 ± 0.0 | 73.2 ± 0.0 | 36.3 ± 0.0 | 46.8 ± 0.0 | 46.2 ± 1.0 | 45.6 ± 0.5 | 46.4 ± 0.4 | 49.9 ± 0.2 | 27.8 ± 0.0 | 64.2 ± 1.5 | 49.4 ± 0.9 | 50.8 ± 2.8 | 47.8 ± 0.0 | 39.5 ± 0.0 | 43.9 ± 0.7 | 38.4 ± 0.2 | 51.7 ± 7.3 | 58.2 ± 6.3 |
| letter | 59.9 ± 0.0 | 44.4 ± 0.0 | 67.5 ± 0.0 | 18.9 ± 0.0 | 17.0 ± 0.0 | 48.9 ± 1.0 | 27.5 ± 0.9 | 41.2 ± 3.4 | 67.4 ± 6.6 | 13.8 ± 0.0 | 55.5 ± 1.0 | 60.0 ± 0.2 | 39.6 ± 1.8 | 44.7 ± 0.0 | 60.5 ± 2.4 | 70.6 ± 0.4 | 29.7 ± 0.3 | 39.1 ± 2.1 | 29.7 ± 1.4 |
| Lymphography | 80.0 ± 0.0 | 80.0 ± 0.0 | 80.0 ± 0.0 | 72.0 ± 0.0 | 84.3 ± 0.0 | 82.8 ± 2.8 | 82.8 ± 2.8 | 42.9 ± 2.1 | 46.2 ± 0.1 | 89.7 ± 0.0 | 95.8 ± 0.0 | 82.8 ± 2.8 | 80.0 ± 0.0 | 97.6 ± 0.0 | 77.6 ± 6.0 | 80.0 ± 0.0 | 75.0 ± 0.0 | 82.3 ± 15.7 | 80.1 ± 8.1 |
| magic.gamma | 80.4 ± 0.0 | 85.4 ± 0.0 | 86.8 ± 0.0 | 79.7 ± 0.0 | 79.5 ± 0.0 | 86.7 ± 0.0 | 83.7 ± 0.1 | 83.2 ± 0.3 | 80.1 ± 1.7 | 67.6 ± 0.0 | 83.1 ± 0.1 | 77.2 ± 0.4 | 81.4 ± 1.5 | 86.4 ± 0.0 | 89.5 ± 0.2 | 85.9 ± 0.7 | 76.6 ± 0.1 | 85.0 ± 0.5 | 81.9 ± 0.4 |
| mammography | 41.6 ± 0.0 | 41.9 ± 0.0 | 35.0 ± 0.0 | 41.2 ± 0.0 | 47.1 ± 0.0 | 59.7 ± 1.9 | 38.5 ± 0.8 | 44.8 ± 2.2 | 11.4 ± 0.4 | 54.0 ± 0.0 | 30.0 ± 0.8 | 20.2 ± 1.3 | 36.6 ± 7.2 | 43.9 ± 0.0 | 48.0 ± 0.3 | 39.0 ± 2.3 | 38.6 ± 0.3 | 49.6 ± 6.8 | 59.4 ± 3.5 |
| mnist | 81.2 ± 0.0 | 77.1 ± 0.0 | 73.2 ± 0.0 | 69.4 ± 0.0 | 36.4 ± 0.0 | 78.9 ± 1.3 | 69.7 ± 1.7 | 67.1 ± 1.2 | 64.4 ± 1.0 | 29.9 ± 0.0 | 62.6 ± 0.1 | 70.2 ± 0.1 | 58.2 ± 5.0 | 87.8 ± 0.0 | 63.9 ± 1.1 | 76.5 ± 0.2 | 69.7 ± 0.7 | 68.5 ± 1.2 | 59.7 ± 2.9 |
| musk | 100.0 ± 0.0 | 100.0 ± 0.0 | 100.0 ± 0.0 | 100.0 ± 0.0 | 100.0 ± 0.0 | 100.0 ± 0.0 | 100.0 ± 0.0 | 84.9 ± 6.7 | 100.0 ± 0.0 | 63.2 ± 0.0 | 100.0 ± 0.0 | 100.0 ± 0.0 | 100.0 ± 0.0 | 100.0 ± 0.0 | 100.0 ± 0.0 | 100.0 ± 0.0 | 100.0 ± 0.0 | 100.0 ± 0.0 | 36.7 ± 12.4 |
| optdigits | 49.7 ± 0.0 | 33.5 ± 0.0 | 66.6 ± 0.0 | 7.0 ± 0.0 | 20.9 ± 0.0 | 30.9 ± 0.4 | 11.5 ± 0.6 | 40.5 ± 3.6 | 79.2 ± 9.5 | 7.0 ± 0.0 | 52.6 ± 2.9 | 35.2 ± 9.3 | 8.8 ± 3.2 | 56.3 ± 0.0 | 22.4 ± 0.8 | 28.6 ± 5.1 | 6.8 ± 0.0 | 65.8 ± 6.6 | 36.7 ± 12.4 |
| PageBlocks | 84.8 ± 0.0 | 86.6 ± 0.0 | 89.0 ± 0.0 | 86.2 ± 0.0 | 76.8 ± 0.0 | 88.2 ± 1.2 | 87.5 ± 0.4 | 74.5 ± 1.8 | 78.5 ± 0.7 | 66.4 ± 0.0 | 81.4 ± 1.2 | 74.7 ± 2.3 | 78.8 ± 4.4 | 54.1 ± 0.0 | 86.6 ± 0.6 | 86.2 ± 2.7 | 79.6 ± 0.3 | 76.6 ± 4.2 | 90.1 ± 1.2 |
| pendigits | 96.7 ± 0.0 | 95.9 ± 0.0 | 91.9 ± 0.0 | 61.8 ± 0.0 | 56.7 ± 0.0 | 86.9 ± 4.4 | 55.4 ± 18.8 | 68.0 ± 8.1 | 44.4 ± 8.6 | 38.5 ± 0.0 | 66.0 ± 6.4 | 47.5 ± 8.4 | 71.9 ± 1.5 | 54.6 ± 0.0 | 53.1 ± 4.1 | 70.6 ± 8.6 | 60.9 ± 1.7 | 84.8 ± 2.7 | 90.0 ± 4.1 |
| Pima | 77.0 ± 0.0 | 77.9 ± 0.0 | 76.0 ± 0.0 | 76.0 ± 0.0 | 78.2 ± 0.0 | 78.1 ± 0.0 | 75.2 ± 1.0 | 77.6 ± 0.8 | 63.0 ± 1.5 | 65.7 ± 0.0 | 69.4 ± 2.0 | 66.6 ± 7.2 | 74.3 ± 0.5 | 73.8 ± 0.0 | 73.4 ± 1.4 | 74.1 ± 2.1 | 75.6 ± 0.3 | 79.5 ± 0.9 | 80.5 ± 1.0 |
| satellite | 89.2 ± 0.0 | 89.3 ± 0.0 | 89.5 ± 0.0 | 82.3 ± 0.0 | 85.6 ± 0.0 | 86.0 ± 0.0 | 88.2 ± 0.9 | 86.8 ± 1.1 | 75.9 ± 0.0 | 65.7 ± 0.0 | 90.6 ± 0.3 | 89.0 ± 0.2 | 86.7 ± 0.3 | 86.5 ± 0.0 | 88.1 ± 0.4 | 87.3 ± 0.4 | 84.3 ± 0.0 | 87.9 ± 0.3 | 86.5 ± 2.1 |
| satimage-2 | 98.3 ± 0.0 | 98.2 ± 0.0 | 91.1 ± 0.0 | 97.5 ± 0.0 | 91.7 ± 0.0 | 97.7 ± 0.2 | 97.6 ± 0.0 | 57.2 ± 2.5 | 16.7 ± 1.0 | 77.3 ± 0.0 | 91.3 ± 0.4 | 96.4 ± 0.0 | 88.0 ± 3.9 | 73.9 ± 0.0 | 58.3 ± 5.0 | 81.4 ± 10.5 | 98.3 ± 0.0 | 94.8 ± 1.8 | 94.0 ± 2.6 |
| shuttle | 98.0 ± 0.0 | 97.5 ± 0.0 | 99.4 ± 0.0 | 98.0 ± 0.0 | 99.0 ± 0.0 | 98.6 ± 0.2 | 97.0 ± 0.2 | 98.1 ± 0.2 | 99.9 ± 0.0 | 94.3 ± 0.0 | 99.2 ± 0.1 | 97.1 ± 0.2 | 97.6 ± 0.4 | 98.3 ± 0.0 | 94.3 ± 0.0 | 98.9 ± 0.2 | 97.1 ± 0.2 | 98.5 ± 0.0 | 98.3 ± 0.0 |
| skin | 65.0 ± 0.0 | 99.5 ± 0.0 | 76.8 ± 0.0 | 66.3 ± 0.0 | 66.0 ± 0.0 | 67.8 ± 1.8 | 48.4 ± 0.6 | 65.7 ± 2.5 | 80.7 ± 8.1 | 30.3 ± 0.0 | 34.4 ± 0.0 | 82.7 ± 4.3 | 83.2 ± 15.1 | 87.4 ± 0.0 | 70.5 ± 0.1 | 69.9 ± 2.1 | 68.0 ± 0.5 | 83.4 ± 3.0 | 83.4 ± 3.0 |
| smtp | 58.8 ± 0.0 | 42.0 ± 0.0 | 40.4 ± 0.0 | 61.4 ± 0.0 | 1.2 ± 0.0 | 40.8 ± 0.6 | 28.4 ± 3.8 | 32.0 ± 2.0 | 48.1 ± 1.9 | 52.6 ± 0.0 | 12.3 ± 12.2 | 43.9 ± 0.0 | 50.1 ± 10.9 | 65.0 ± 0.0 | 43.6 ± 1.5 | 52.8 ± 12.3 | 43.6 ± 6.1 | 57.6 ± 0.9 | 57.9 ± 0.5 |
| SpamBase | 88.3 ± 0.0 | 86.6 ± 0.0 | 83.8 ± 0.0 | 85.1 ± 0.0 | 84.1 ± 0.0 | 85.5 ± 0.3 | 86.5 ± 0.1 | 80.5 ± 0.4 | 83.8 ± 0.8 | 68.9 ± 0.0 | 87.0 ± 0.2 | 87.9 ± 0.2 | 74.3 ± 2.6 | 78.8 ± 0.0 | 87.2 ± 0.3 | 88.9 ± 0.2 | 84.2 ± 0.1 | 85.3 ± 0.6 | 80.9 ± 0.6 |
| speech | 7.5 ± 0.0 | 3.7 ± 0.0 | 6.9 ± 0.0 | 3.6 ± 0.0 | 4.0 ± 0.0 | 3.8 ± 0.1 | 3.1 ± 0.1 | 3.2 ± 0.0 | 4.6 ± 0.8 | 3.8 ± 0.0 | 4.2 ± 0.9 | 3.9 ± 0.3 | 4.9 ± 1.3 | 5.9 ± 0.0 | 4.0 ± 0.5 | 5.0 ± 0.1 | 3.9 ± 0.1 | 4.9 ± 0.7 | 5.1 ± 0.2 |
| Stamps | 63.7 ± 0.0 | 54.2 ± 0.0 | 55.4 ± 0.0 | 56.3 ± 0.0 | 69.4 ± 0.0 | 59.5 ± 4.2 | 49.6 ± 1.1 | 61.8 ± 6.1 | 33.2 ± 3.4 | 45.2 ± 0.0 | 61.3 ± 4.7 | 36.9 ± 2.0 | 57.2 ± 3.9 | 51.5 ± 0.0 | 54.1 ± 3.7 | 55.0 ± 2.5 | 53.6 ± 0.1 | 78.4 ± 13.0 | 88.3 ± 4.8 |
| thyroid | 73.8 ± 0.0 | 77.4 ± 0.0 | 80.7 ± 0.0 | 74.8 ± 0.0 | 88.6 ± 0.0 | 73.7 ± 1.8 | 80.0 ± 0.0 | 76.4 ± 1.9 | 9.1 ± 0.5 | 62.9 ± 0.0 | 34.9 ± 0.2 | 79.5 ± 2.6 | 71.8 ± 4.1 | 84.9 ± 0.0 | 86.8 ± 0.1 | 70.3 ± 4.5 | 76.4 ± 0.1 | 81.7 ± 1.3 | 87.6 ± 1.7 |
| vertebral | 19.7 ± 0.0 | 20.3 ± 0.0 | 29.6 ± 0.0 | 26.4 ± 0.0 | 22.2 ± 0.0 | 24.7 ± 0.1 | 23.1 ± 0.5 | 25.3 ± 2.2 | 33.8 ± 3.2 | 19.5 ± 0.0 | 30.3 ± 0.8 | 20.3 ± 1.0 | 23.7 ± 5.3 | 23.2 ± 0.0 | 33.0 ± 4.2 | 35.5 ± 5.4 | 17.4 ± 0.0 | 51.4 ± 7.4 | 54.9 ± 2.3 |
| vowels | 77.7 ± 0.0 | 76.3 ± 0.0 | 79.0 ± 0.0 | 52.7 ± 0.0 | 26.5 ± 0.0 | 85.3 ± 4.5 | 45.6 ± 2.4 | 63.6 ± 3.6 | 88.6 ± 4.4 | 14.2 ± 0.0 | 86.3 ± 4.3 | 65.6 ± 10.3 | 73.9 ± 4.6 | 78.8 ± 0.0 | 91.8 ± 1.7 | 77.8 ± 8.7 | 64.9 ± 0.6 | 48.5 ± 2.1 | 50.0 ± 7.8 |
| Waveform | 27.6 ± 0.0 | 25.4 ± 0.0 | 33.8 ± 0.0 | 11.2 ± 0.0 | 10.8 ± 0.0 | 15.8 ± 1.0 | 10.3 ± 0.8 | 11.3 ± 0.6 | 49.0 ± 0.3 | 7.6 ± 0.0 | 8.0 ± 1.0 | 6.8 ± 0.1 | 32.4 ± 9.8 | 20.4 ± 0.0 | 11.5 ± 0.0 | 19.6 ± 1.7 | 15.4 ± 0.0 | 32.3 ± 0.9 | 14.8 ± 1.4 |
| WBC | 96.7 ± 0.0 | 96.7 ± 0.0 | 96.7 ± 0.0 | 91.8 ± 0.0 | 100.0 ± 0.0 | 99.5 ± 0.0 | 97.3 ± 1.8 | 83.2 ± 4.1 | 14.2 ± 0.3 | 99.0 ± 0.0 | 25.2 ± 2.5 | 93.5 ± 3.8 | 96.3 ± 4.0 | 84.8 ± 4.0 | 86.9 ± 2.9 | 91.7 ± 0.5 | 95.6 ± 3.9 | 97.0 ± 0.3 | |
| WDBC | 90.9 ± 0.0 | 82.2 ± 0.0 | 94.3 ± 0.0 | 92.4 ± 0.0 | 99.1 ± 0.0 | 90.4 ± 3.7 | 91.0 ± 0.9 | 36.0 ± 1.4 | 4.7 ± 0.3 | 73.9 ± 0.0 | 60.9 ± 7.6 | 81.7 ± 1.9 | 94.3 ± 0.0 | 100.0 ± 0.0 | 79.0 ± 0.1 | 89.8 ± 0.9 | 90.1 ± 1.4 | 96.3 ± 2.0 | 88.0 ± 5.5 |
| Wilt | 7.4 ± 0.0 | 12.9 ± 0.0 | 35.4 ± 0.0 | 7.2 ± 0.0 | 10.8 ± 0.0 | 19.8 ± 0.2 | 17.2 ± 1.9 | 11.3 ± 0.1 | 53.8 ± 1.9 | 8.1 ± 0.0 | 36.5 ± 2.4 | 20.8 ± 4.6 | 26.1 ± 0.7 | 29.5 ± 0.0 | 38.9 ± 1.5 | 72.7 ± 4.5 | 8.8 ± 0.1 | 30.6 ± 1.8 | 25.8 ± 2.1 |
| wine | 58.2 ± 0.0 | 61.5 ± 0.0 | 52.7 ± 0.0 | 58.7 ± 0.0 | 69.1 ± 0.0 | 65.3 ± 10.9 | 59.0 ± 13.4 | 56.6 ± 7.1 | 88.4 ± 7.7 | 30.5 ± 0.0 | 60.3 ± 8.8 | 73.1 ± 21.0 | 47.1 ± 14.4 | 100.0 ± 0.0 | 72.3 ± 9.7 | 66.2 ± 4.8 | 59.8 ± 0.5 | 99.1 ± 1.5 | 97.2 ± 1.2 |
| WPBC | 38.4 ± 0.0 | 38.2 ± 0.0 | 39.0 ± 0.0 | 37.8 ± 0.0 | 39.9 ± 0.0 | 38.1 ± 1.8 | 39.0 ± 2.0 | 43.8 ± 1.0 | 45.5 ± 3.2 | 35.4 ± 0.0 | 47.2 ± 1.7 | 38.9 ± 1.1 | 40.2 ± 3.0 | 42.5 ± 0.0 | 40.1 ± 2.1 | 37.8 ± 0.6 | 36.9 ± 0.0 | 68.7 ± 6.3 | 52.7 ± 6.4 |
| yeast | 48.2 ± 0.0 | 49.6 ± 0.0 | 52.4 ± 0.0 | 51.1 ± 0.0 | 52.3 ± 0.0 | 52.1 ± 0.1 | 51.3 ± 0.0 | 51.7 ± 0.6 | 56.6 ± 1.1 | 50.0 ± 0.0 | 55.6 ± 1.9 | 54.4 ± 0.6 | 54.8 ± 3.1 | 51.7 ± 0.0 | 52.2 ± 0.8 | 56.9 ± 1.4 | 46.1 ± 0.4 | 58.5 ± 2.9 | 61.1 ± 1.3 |
| Avg. AUPRC | 63.0 | 63.6 | 64.5 | 56.9 | 54.3 | 63.9 | 57.8 | 52.8 | 54.2 | 45.0 | 56.8 | 56.5 | 55.6 | 65.1 | 61.8 | 66.0 | 55.6 | 68.1 | 66.3 |
| Avg. Rank | 8.29 | 8.27 | 8.37 | 11.19 | 10.73 | 7.74 | 10.88 | 12.85 | 10.90 | 15.34 | 10.09 | 11.43 | 12.35 | 7.37 | 9.79 | 7.71 | 12.51 | 6.82 | 7.36 |
| *p*-value | 0.021 | 0.015 | 0.078 | 0.000 | 0.000 | 0.041 | 0.000 | 0.000 | 0.003 | 0.000 | 0.001 | 0.000 | 0.000 | 0.162 | 0.005 | 0.249 | 0.000 | - | 0.149 |

## I.3. Anomaly Detection with Anomaly Contamination

Under this experimental setting, we conduct experiments on 10 datasets where the contamination ratio ranges in {1%, 3%, 5%, 10%}. Detailed results for each dataset are shown in Table 19 and Table 20, while the average performance under different anomaly contamination ratios is shown in Figure 4. As the anomaly ratio increases, the performance variation of our proposed method remains minimal, demonstrating its robustness to anomalous data in the training set.

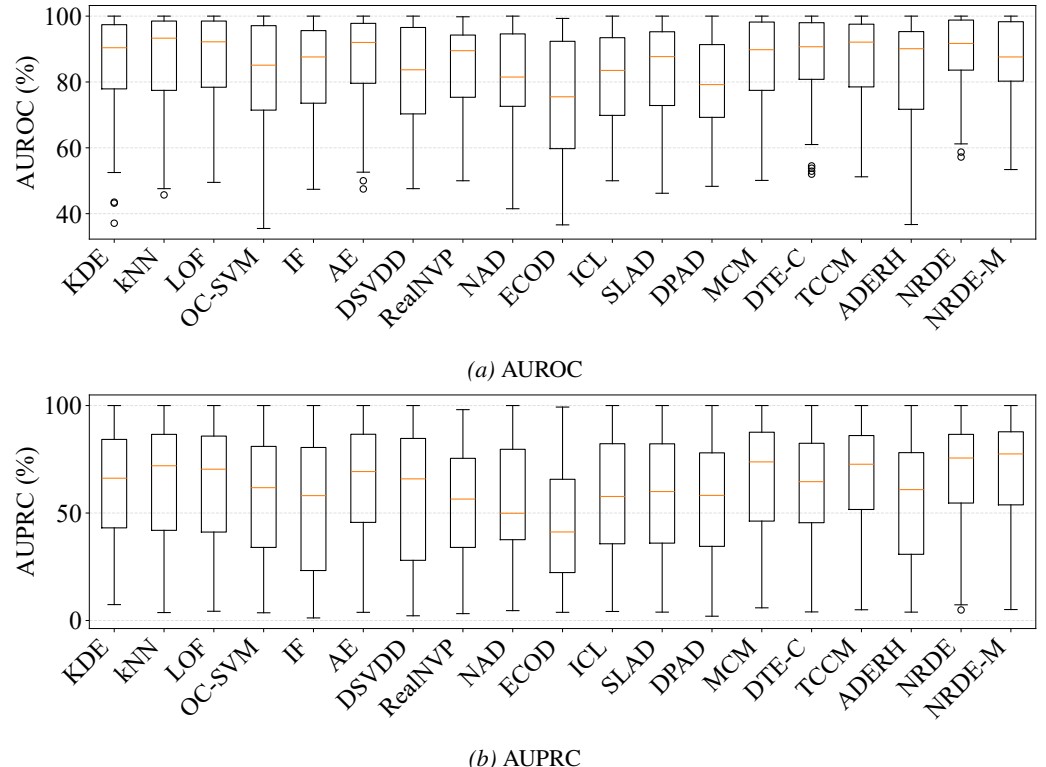

*(a)* AUROC

*(b)* AUPRC

*Figure 7.* Box plots of 19 methods on 47 benchmark datasets under the standard anomaly detection setting.

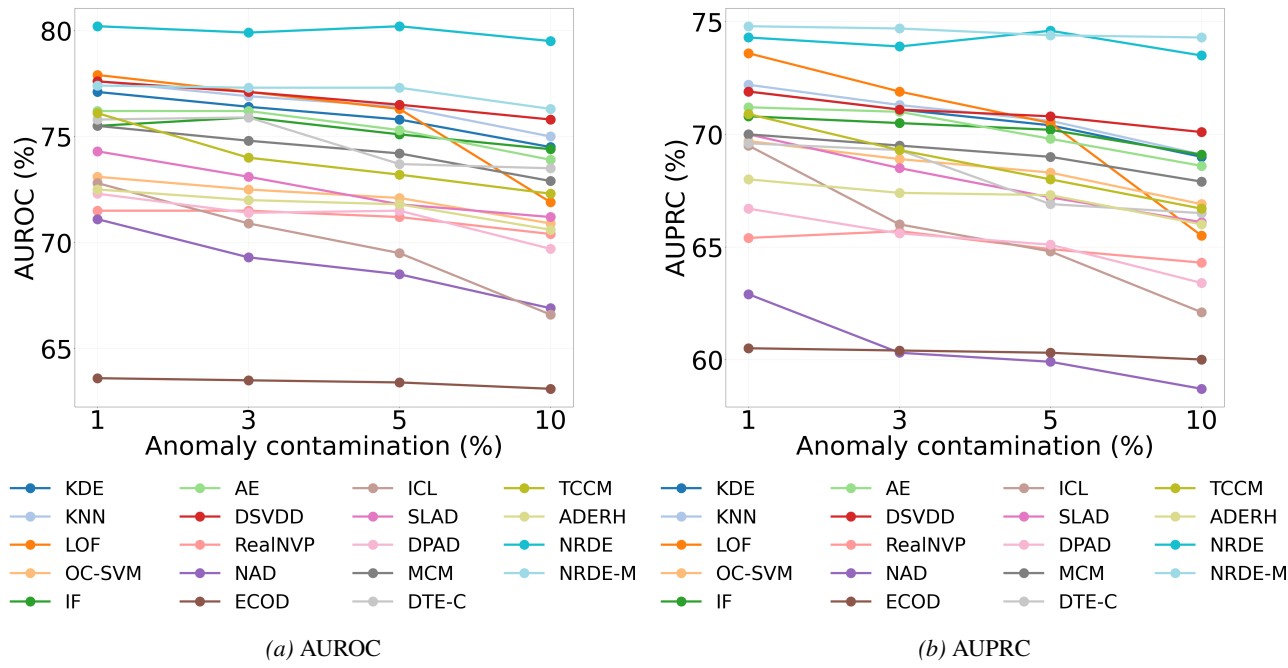

*(a)* AUROC

*(b)* AUPRC

*Figure 8.* Average performance of all methods under different nomaly contamination ratio.

## I.4. Anomaly Detection on Noisy Data

The detailed results of anomaly detection on noisy data are provided in Table 21. We also consider a symmetric variant where anomalous test samples receive stronger noise. In this setting, most methods exhibit improved performance compared to the standard anomaly detection setup, suggesting that adding stronger noise to anomalous samples increases the separability

*Table 15.* Average AUROC (%) of each method on 47 tabular datasets of ADBench. The best results are marked in **bold**. Each of these methods follow the recommended hyperparameter configuration in their original paper or code without dataset-specific tuning.

| Methods | KDE | KNN | LOF | OC-SVM | IF | AE | DSVDD | RealNVP | NeutralAD | ECOD | ICL | SLAD | DPAD | MCM | DTE-C | TCCM | ADERH | NRDE |
|---|---|---|---|---|---|---|---|---|---|---|---|---|---|---|---|---|---|---|
| ALOI | 56.4 | 63.4 | **74.6** | 55.0 | 54.4 | 55.8 | 51.8 | 56.2 | 55.1 | 53.0 | 50.2 | 54.8 | 51.7 | 63.2 | 54.2 | 54.3 | 55.1 | 54.9 |
| annthyroid | 91.4 | 94.1 | 92.9 | 90.9 | 91.8 | 83.4 | 79.4 | 96.1 | 78.9 | 78.7 | 64.0 | 90.4 | 91.2 | 83.9 | **97.8** | 82.4 | 90.6 | 85.5 |
| backdoor | 90.5 | 93.7 | 95.7 | 62.6 | 75.1 | 93.5 | 92.5 | 91.8 | 87.1 | 84.6 | 95.2 | 92.5 | 94.6 | **96.6** | 91.8 | 95.9 | 75.2 | 92.1 |
| breastw | 98.9 | 99.1 | 96.7 | 99.0 | **99.5** | 98.4 | 99.1 | 98.0 | 81.4 | 99.3 | 90.2 | 99.2 | 98.9 | 99.0 | 96.3 | 98.5 | 98.7 | 98.9 |
| campaign | 77.3 | 78.3 | 69.8 | 77.1 | 72.0 | 80.8 | 1.1 | 79.6 | 62.3 | 78.3 | 80.5 | 76.3 | 64.2 | 78.5 | 76.7 | 79.5 | 64.2 | **81.0** |
| cardio | 95.7 | 93.4 | 93.0 | **96.4** | 94.9 | 92.4 | 96.1 | 94.1 | 81.0 | 93.4 | 83.9 | 88.7 | 90.0 | 90.4 | 93.6 | 90.3 | 94.1 | 92.8 |
| Cardiotocography | 75.0 | 71.3 | 72.7 | 80.7 | 79.3 | 73.4 | **83.4** | 77.9 | 58.2 | 78.5 | 54.7 | 58.4 | 68.0 | 70.0 | 72.4 | 63.3 | 60.1 | 68.6 |
| celeba | 70.5 | 68.0 | 44.9 | 79.0 | 70.8 | 70.9 | 48.4 | 79.4 | 66.4 | 75.7 | 69.5 | 65.2 | 56.3 | 65.3 | **82.7** | 73.4 | 79.4 | 81.0 |
| census | 72.0 | 71.9 | 60.5 | 70.2 | 62.7 | 71.8 | 51.9 | 72.8 | 72.9 | 65.9 | 66.8 | 68.9 | 50.7 | 68.1 | 69.6 | 67.9 | 69.5 | **78.4** |
| cover | 95.5 | 98.5 | **98.9** | 96.2 | 14.0 | 98.3 | 47.6 | 83.8 | 85.1 | 92.0 | 67.4 | 79.2 | 87.9 | 96.4 | 96.7 | 97.2 | 92.4 | 87.8 |
| donors | 97.4 | 99.8 | 98.2 | 92.0 | 88.4 | 93.6 | 97.7 | 96.6 | 95.8 | 88.8 | 94.0 | 96.9 | 98.7 | **100.0** | 98.9 | 99.8 | 99.8 | **100.0** |
| fault | **81.2** | 76.9 | 67.2 | 61.4 | 65.2 | 73.3 | 71.9 | 50.8 | 73.2 | 47.4 | 77.9 | 79.2 | 73.5 | 71.2 | 71.8 | 64.7 | 70.2 | 68.4 |
| fraud | 95.8 | **96.0** | 78.3 | 95.6 | 95.0 | 95.7 | 50.6 | 54.4 | 94.9 | 94.9 | 83.4 | 94.6 | 64.1 | 95.8 | 94.4 | 93.9 | 95.2 | 95.7 |
| glass | 83.5 | 89.3 | 74.0 | 71.1 | 81.3 | 77.9 | 79.8 | 81.5 | **92.1** | 69.8 | 90.8 | 83.5 | 88.8 | 86.5 | 78.4 | 74.4 | 88.8 | 86.5 |
| Hepatitis | 79.4 | 85.0 | 84.6 | 84.2 | 77.8 | 83.9 | 80.3 | 59.4 | 62.2 | 71.5 | 60.1 | 77.6 | 83.7 | 81.2 | 80.8 | 79.1 | 83.5 | **85.5** |
| http | **100.0** | 99.9 | 93.0 | **100.0** | 99.1 | 99.8 | 99.8 | 99.6 | 99.0 | 97.8 | 50.0 | 99.9 | 99.8 | 99.9 | 99.5 | **100.0** | **100.0** | **100.0** |
| InternetAds | 85.7 | 73.8 | 78.7 | 73.8 | 43.6 | **88.2** | 52.4 | 81.2 | 81.6 | 68.9 | 78.9 | 86.7 | 83.2 | 82.3 | 87.1 | 87.7 | 70.2 | 83.9 |
| Ionosphere | **97.4** | 97.0 | 94.7 | 96.5 | 93.6 | 93.6 | 78.7 | 93.8 | 96.4 | 73.5 | 94.1 | 86.3 | 97.3 | 95.4 | 95.4 | 95.9 | 96.0 | 94.6 |
| landsat | 72.7 | 73.9 | **75.4** | 45.7 | 59.9 | 57.0 | 58.1 | 59.7 | 70.7 | 36.6 | 73.8 | 69.9 | 69.7 | 62.2 | 58.8 | 55.5 | 59.6 | 55.9 |
| letter | 91.8 | 84.1 | 86.1 | 60.9 | 61.7 | 80.1 | 34.2 | 83.1 | **92.5** | 56.0 | 87.5 | 90.3 | 80.2 | 89.0 | 89.6 | 85.0 | 72.6 | 77.3 |
| Lymphography | 98.6 | **98.6** | **98.6** | 98.4 | 97.7 | 98.5 | 95.4 | 94.3 | 82.9 | 98.5 | 92.9 | 98.5 | 98.5 | 98.5 | 97.7 | 98.4 | **98.6** | 95.4 |
| magic.gamma | 75.7 | 82.2 | 83.2 | 73.5 | 77.3 | 81.8 | 76.1 | 79.6 | 77.5 | 63.4 | 71.7 | 72.4 | 79.8 | 82.6 | **85.8** | 60.7 | 69.5 | 79.9 |
| mammography | 88.1 | 87.6 | 83.8 | 88.9 | 88.3 | 87.5 | 81.3 | 89.5 | 70.9 | **90.7** | 57.3 | 72.6 | 84.8 | **90.7** | 87.8 | 79.5 | 86.9 | 86.7 |
| mnist | **94.8** | 93.8 | 92.6 | 91.0 | 86.6 | 93.4 | 85.1 | 92.4 | 77.8 | 74.7 | 86.7 | 91.2 | 85.8 | 90.3 | 84.9 | 90.0 | 90.4 | 92.3 |
| musk | **100.0** | **100.0** | 1.0 | **100.0** | 95.8 | 1.0 | 99.9 | 99.4 | **100.0** | 95.8 | **100.0** | **100.0** | 99.9 | **100.0** | **100.0** | **100.0** | **100.0** | **100.0** |
| optdigits | 97.4 | 92.7 | 97.8 | 60.9 | 79.6 | 88.7 | 32.6 | 94.2 | 95.9 | 60.4 | 91.7 | 91.4 | 75.5 | 87.8 | 89.2 | 57.9 | 62.6 | **97.9** |
| PageBlocks | 95.0 | 95.8 | **96.7** | 94.4 | 92.4 | 94.8 | 94.7 | 92.0 | 93.7 | 91.4 | 79.5 | 87.7 | 95.4 | 96.3 | 96.2 | 93.6 | 93.9 | 93.4 |
| pendigits | **99.8** | **99.8** | 98.8 | 96.1 | 97.0 | 98.0 | 88.6 | 98.5 | 93.9 | 92.7 | 93.8 | 92.7 | 94.7 | 98.6 | 97.6 | 94.7 | 97.0 | 99.3 |
| Pima | 78.1 | 78.0 | 73.6 | 73.1 | 76.5 | 73.9 | 73.0 | **79.3** | 56.0 | 61.5 | 63.1 | 54.6 | 70.2 | 74.1 | 70.7 | 66.8 | 75.7 | 78.9 |
| satellite | 86.9 | **87.3** | 85.1 | 75.4 | 80.0 | 80.3 | 81.7 | 83.6 | 78.4 | 58.3 | 83.9 | 87.0 | 86.5 | 82.0 | 86.1 | 80.5 | 81.1 | 86.4 |
| satimage-2 | **99.9** | **99.9** | 99.6 | 99.7 | 99.4 | 99.8 | 98.5 | 98.9 | 84.9 | 96.6 | 96.2 | 99.7 | 98.7 | 98.9 | 98.9 | 98.0 | **99.9** | 99.7 |
| shuttle | 99.8 | 99.9 | 99.9 | 99.7 | 99.6 | 99.8 | 99.2 | 99.8 | 99.9 | 99.3 | 99.4 | 99.9 | 99.9 | **100.0** | 99.7 | 99.9 | 99.8 | 99.9 |
| skin | 89.1 | **99.8** | 92.5 | 90.3 | 88.8 | 83.9 | 69.6 | 90.0 | 88.4 | 48.8 | 50.0 | 91.2 | 99.3 | 79.4 | 92.4 | 86.0 | 91.2 | 92.9 |
| smtp | 88.2 | 93.5 | 94.2 | 85.5 | 90.1 | 91.4 | 89.5 | 93.4 | 91.0 | 87.9 | 53.0 | 91.9 | 93.4 | 85.1 | **95.2** | 87.0 | 92.8 | 80.0 |
| SpamBase | **85.7** | 83.0 | 81.7 | 81.6 | 83.6 | 82.0 | 79.4 | 80.1 | 79.2 | 66.0 | 81.4 | 85.3 | 76.8 | 81.3 | 84.5 | 76.2 | 79.0 | 81.1 |
| speech | 45.8 | 48.5 | 48.9 | 45.9 | 46.7 | 46.8 | 45.2 | 50.0 | 54.3 | 46.1 | 49.1 | 50.7 | 54.8 | 49.9 | **56.1** | 53.7 | 48.9 | 54.4 |
| Stamps | **95.1** | 90.8 | 87.2 | 91.2 | 91.9 | 89.2 | 91.9 | 93.6 | 74.2 | 86.7 | 88.1 | 73.0 | 90.9 | 88.6 | 67.5 | 74.5 | 91.4 | 94.5 |
| thyroid | 98.3 | 98.5 | 94.6 | 98.2 | 99.0 | 98.0 | 97.5 | 98.6 | 65.2 | 97.7 | 82.2 | 94.8 | 96.1 | 97.9 | 99.2 | 83.2 | 98.2 | **99.3** |
| vertebral | 43.5 | 42.5 | 40.0 | 52.7 | 42.6 | 48.0 | 43.7 | 53.6 | 53.9 | 41.8 | 54.2 | 44.4 | 46.4 | 47.2 | 59.2 | 49.1 | 33.6 | **71.0** |
| vowels | 96.5 | 97.3 | 96.8 | 83.0 | 77.7 | 95.3 | 41.3 | 90.4 | **98.7** | 59.5 | 98.2 | 97.2 | 93.4 | 91.5 | 97.3 | 85.8 | 93.6 | 90.1 |
| Waveform | 76.0 | 76.2 | 76.6 | 69.0 | 72.5 | 65.8 | 69.6 | 72.5 | 71.5 | 60.0 | 59.8 | 50.2 | 61.0 | 69.6 | 65.6 | 55.0 | 70.8 | **81.5** |
| WBC | 98.1 | 99.4 | 97.9 | 99.0 | **99.7** | 99.0 | 99.3 | 98.8 | 78.6 | 99.0 | 80.0 | 98.8 | 98.3 | 99.1 | 98.2 | 98.1 | 98.5 | 98.9 |
| WDBC | **99.4** | 99.1 | **99.4** | 99.3 | 99.3 | 99.0 | 99.3 | 95.0 | 32.5 | 97.8 | 83.7 | 97.8 | 97.2 | 97.9 | 38.9 | 99.3 | 99.3 | 98.7 |
| Wilt | 37.1 | 60.8 | 70.8 | 33.9 | 50.4 | 56.2 | 49.9 | 59.9 | 80.3 | 40.3 | 78.2 | 66.8 | 73.1 | 66.0 | 86.8 | 83.5 | 40.1 | **88.8** |
| wine | 92.2 | 93.2 | 92.2 | 91.2 | 88.5 | 85.6 | 90.2 | 92.6 | 78.5 | 73.0 | 82.6 | 92.7 | 85.3 | 95.8 | 92.3 | 94.1 | 92.0 | **99.0** |
| WPBC | 52.5 | 51.3 | 50.5 | 49.1 | 52.5 | 49.6 | 50.0 | 58.0 | 59.1 | 47.0 | 53.4 | 50.4 | 52.1 | 52.3 | 48.3 | 46.3 | 49.0 | **71.0** |
| yeast | 43.2 | 46.6 | 46.7 | 44.9 | 42.9 | 47.9 | 42.5 | 51.0 | **60.1** | 45.3 | 55.9 | 52.7 | 51.8 | 45.7 | 50.5 | 58.9 | 39.4 | 56.4 |
| AVG | 84.3 | 85.1 | 81.0 | 79.8 | 78.2 | 81.3 | 74.3 | 83.6 | 77.9 | 73.7 | 76.1 | 79.2 | 81.5 | 83.4 | 83.3 | 80.6 | 80.5 | **85.9** |

between normal and anomalous instances and thereby makes the task easier. The detailed results are shown in Table 22.

## I.5. Outlier Detection

To evaluate the effectiveness of our method in outlier detection (transductive learning), we conduct experiments on 10 datasets where all data are used for both training and testing, and compare our method with other AD methods. We provide the detailed experimental results for outlier detection on 10 datasets. We compare our proposed method with traditional and state-of-the-art outlier detection methods. The AUROC and AUPRC results are shown in Table 23.

## I.6. Ablation Studies

In this subsection, we investigate how changes in hyperparameters and the architecture of the normalizing flow affect the anomaly detection performance of NRDE. Table 24 and 25 present the performance results of NRDE with different hyperparameter configurations across five datasets. We observe that the regularization term $\mathcal{R}$ contributes positively to overall performance. We also observe that the method is not highly sensitive to changes in $\lambda$ and learning rate; however, in some datasets, large values of $\lambda$ may affect the training process and lead to a decrease in performance.

We also include additional experiments to evaluate the sensitivity of our method to architectural choices in Table 26 and Table 27. Overall, NRDE remains robust across different architectures.

Moreover, to validate the effectiveness of our strategy for dynamically selecting $\hat{m}$ across different datasets, we compare

*Table 16.* Average AUPRC (%) of each method on 47 tabular datasets of ADBench. The best results are marked in **bold**. Each of these methods follow the recommended hyperparameter configuration in their original paper or code without dataset-specific tuning.

| Methods | KDE | KNN | LOF | OC-SVM | IF | AE | DSVDD | RealNVP | NeutralAD | ECOD | ICL | SLAD | DPAD | MCM | DTE-C | TCCM | ADERH | NRDE |
|---|---|---|---|---|---|---|---|---|---|---|---|---|---|---|---|---|---|---|
| ALOI | 10.5 | 9.8 | **15.9** | 7.5 | 6.6 | 7.6 | 7.2 | 7.1 | 7.5 | 6.4 | 6.1 | 7.1 | 7.1 | 10.9 | 6.8 | 9.2 | 7.2 | 7.0 |
| annthyroid | 66.2 | 72.0 | 66.7 | 65.2 | 63.8 | 60.7 | 54.8 | 77.0 | 29.4 | 40.8 | 31.3 | 63.1 | 64.5 | 55.0 | **84.1** | 52.3 | 62.4 | 59.0 |
| backdoor | 44.7 | 45.0 | 59.9 | 7.8 | 9.6 | 86.8 | 71.4 | 77.9 | 55.8 | 16.7 | **89.6** | 86.0 | 65.1 | 81.8 | 63.2 | 84.5 | 12.2 | 75.2 |
| breastw | 98.8 | 99.1 | 93.7 | 98.8 | **99.5** | 98.1 | 99.1 | 96.7 | 71.2 | 99.3 | 86.3 | 99.2 | 98.7 | 99.0 | 92.1 | 98.4 | 98.2 | 98.7 |
| campaign | 47.7 | 48.8 | 39.6 | 48.9 | 43.7 | 49.2 | 42.5 | 50.3 | 28.9 | 50.0 | 49.5 | 48.4 | 33.0 | 50.0 | 48.7 | 50.5 | 42.6 | **57.6** |
| cardio | **84.0** | 76.8 | 69.3 | 82.8 | 78.4 | 74.7 | 83.0 | 71.0 | 48.9 | 70.9 | 60.7 | 72.7 | 73.5 | 73.1 | 69.5 | 74.4 | 80.8 | 79.6 |
| Cardiotocography | 68.1 | 62.4 | 59.9 | 71.0 | 67.6 | 65.0 | **75.1** | 62.6 | 40.3 | 65.7 | 45.4 | 54.7 | 61.5 | 61.3 | 61.1 | 57.4 | 57.8 | 55.4 |
| celeba | 8.9 | 9.8 | 3.7 | 20.4 | 12.5 | 9.5 | 4.0 | 13.1 | 6.6 | 17.2 | 8.9 | **76.1** | 5.8 | 7.3 | 15.7 | 13.3 | 18.6 | 13.7 |
| census | 21.6 | 21.2 | 14.3 | 20.5 | 14.2 | 21.6 | 11.9 | 20.5 | 23.3 | 15.5 | 17.4 | 19.8 | 12.1 | 18.8 | 18.0 | 17.9 | 19.7 | **28.4** |
| cover | 34.2 | 72.0 | 83.7 | 22.0 | 1.1 | 52.8 | 2.0 | 9.3 | 29.1 | 18.4 | 9.0 | 9.2 | 37.4 | 60.7 | 67.9 | **90.2** | 15.0 | 22.3 |
| donors | 70.9 | 95.3 | 76.3 | 42.4 | 37.6 | 49.8 | 82.3 | 66.2 | 64.3 | 41.2 | 88.5 | 65.6 | 96.6 | **99.7** | 77.8 | 98.0 | 47.0 | 94.8 |
| fault | **79.8** | 76.0 | 64.0 | 65.2 | 63.9 | 72.9 | 69.3 | 51.5 | 70.5 | 49.4 | 75.4 | 78.5 | 72.5 | 69.4 | 72.2 | 65.9 | 69.3 | 70.4 |
| fraud | 33.8 | 31.3 | 1.1 | 31.7 | 21.2 | 60.5 | 11.5 | 3.8 | 58.8 | 31.5 | 30.3 | 33.3 | 13.5 | **80.1** | 72.8 | 54.8 | 27.1 | 59.7 |
| glass | 27.9 | 29.9 | 18.8 | 19.2 | 22.6 | 20.8 | 21.2 | 24.5 | **49.7** | 25.1 | 38.7 | 26.1 | 33.3 | 26.1 | 23.4 | 19.3 | 29.6 | 42.1 |
| Hepatitis | 59.7 | 62.1 | 62.0 | 61.6 | 45.9 | 59.8 | 54.4 | 32.0 | 34.5 | 40.2 | 34.3 | 53.4 | 59.6 | 52.4 | 55.9 | 53.2 | 57.1 | **66.7** |
| http | 99.2 | **99.9** | 9.6 | 99.5 | 45.3 | 86.8 | 99.7 | 55.1 | 43.3 | 25.2 | 0.8 | 92.9 | 97.4 | 87.5 | 57.7 | 97.3 | 99.7 | 97.6 |
| InternetAds | 80.7 | 65.3 | 67.3 | 64.4 | 27.2 | **86.1** | 42.3 | 56.5 | 69.6 | 62.8 | 62.2 | 79.6 | 77.8 | 73.8 | 78.4 | 84.5 | 60.9 | 79.8 |
| Ionosphere | 97.9 | **98.3** | 95.2 | 97.3 | 94.0 | 95.5 | 80.6 | 90.9 | 96.9 | 77.9 | 94.4 | 97.1 | 97.7 | 96.6 | 96.1 | 96.9 | 96.8 | 92.7 |
| landsat | 54.8 | 58.0 | **70.4** | 32.1 | 43.2 | 40.0 | 42.5 | 46.4 | 49.4 | 27.8 | 68.9 | 48.3 | 50.6 | 44.0 | 39.4 | 40.8 | 37.1 | 43.6 |
| letter | 59.9 | 44.4 | 49.5 | 20.7 | 15.5 | 36.2 | 8.6 | 41.2 | **69.1** | 13.8 | 52.1 | 57.2 | 40.6 | 46.2 | 64.9 | 52.3 | 26.5 | 37.1 |
| Lymphography | 80.0 | 80.0 | 80.0 | 72.0 | 80.5 | 76.6 | 80.0 | 42.9 | 28.7 | **89.7** | 66.6 | 77.9 | 75.6 | 77.2 | 65.0 | 71.7 | 75.0 | 48.1 |
| magic.gamma | 80.4 | 85.4 | 86.2 | 78.5 | 80.2 | 84.9 | 80.3 | 83.2 | 80.5 | 67.6 | 77.6 | 77.9 | 84.0 | 86.4 | **87.9** | 70.3 | 76.1 | 83.5 |
| mammography | 43.7 | 41.9 | 32.7 | 41.2 | 38.8 | 37.2 | 34.9 | 44.8 | 10.0 | **54.0** | 15.9 | 15.4 | 36.9 | 41.2 | 39.3 | 42.4 | 38.4 | 25.4 |
| mnist | **78.7** | 77.1 | 72.3 | 69.4 | 55.8 | 74.0 | 58.1 | 67.1 | 49.7 | 29.9 | 63.0 | 74.0 | 64.8 | 69.1 | 58.7 | 72.4 | 68.4 | 66.6 |
| musk | **100.0** | 100.0 | 99.0 | 100.0 | 60.8 | 99.9 | 99.2 | 84.9 | 100.0 | 63.2 | 100.0 | 100.0 | 99.9 | 100.0 | 100.0 | 100.0 | 100.0 | 100.0 |
| optdigits | 49.7 | 33.6 | 53.2 | 6.5 | 14.1 | 19.8 | 4.0 | 40.5 | 55.5 | 7.0 | 34.0 | 26.0 | 16.7 | 19.9 | 22.4 | 7.1 | 6.7 | **69.2** |
| PageBlocks | 84.8 | 86.6 | **87.9** | 80.0 | 70.2 | 82.6 | 84.4 | 74.5 | 78.2 | 66.4 | 68.0 | 71.2 | 86.9 | 85.5 | 84.9 | 82.4 | 78.4 | 78.9 |
| pendigits | **96.7** | 95.9 | 69.7 | 47.4 | 48.9 | 56.7 | 27.5 | 68.0 | 30.2 | 38.5 | 48.3 | 32.6 | 58.1 | 66.6 | 45.9 | 65.2 | 60.5 | 88.0 |
| Pima | 77.0 | 76.9 | 73.0 | 74.3 | 71.0 | 73.9 | 73.4 | 77.6 | 56.8 | 65.7 | 63.0 | 58.7 | 71.0 | 74.1 | 70.2 | 69.6 | 74.7 | **79.3** |
| satellite | 89.2 | **89.3** | 88.5 | 82.3 | 84.3 | 85.8 | 84.4 | 86.8 | 74.5 | 65.7 | 87.2 | 87.8 | 87.6 | 85.7 | 87.7 | 84.9 | 84.2 | 88.2 |
| satimage-2 | 98.3 | 97.9 | **99.6** | 97.4 | 93.9 | 94.2 | 93.7 | 57.2 | 7.3 | 77.3 | 82.5 | 91.8 | 89.0 | 61.0 | 52.4 | 93.9 | 98.2 | 89.6 |
| shuttle | 98.1 | 97.5 | 99.4 | 97.5 | 97.3 | 97.0 | 97.0 | 98.1 | **99.9** | 94.3 | 98.3 | 97.5 | 99.2 | 99.2 | 94.0 | 98.3 | 96.9 | 98.6 |
| skin | 65.0 | **99.5** | 73.0 | 66.3 | 63.5 | 64.0 | 48.6 | 65.7 | 67.4 | 30.3 | 34.4 | 80.3 | 98.3 | 63.3 | 70.7 | 69.9 | 68.0 | 73.9 |
| smtp | 58.8 | 42.0 | 29.3 | 60.5 | 0.9 | 28.9 | 16.0 | 32.0 | **62.8** | 52.6 | 6.1 | 52.1 | 58.5 | 43.7 | 44.1 | 51.2 | 49.7 | 41.7 |
| SpamBase | 87.7 | 86.6 | 82.9 | 84.9 | 87.3 | 84.8 | 82.6 | 80.5 | 80.8 | 68.9 | 86.3 | **88.9** | 82.3 | 84.7 | 86.2 | 81.4 | 83.2 | 81.9 |
| speech | 3.7 | 3.7 | 4.5 | 3.6 | 3.5 | 3.6 | 3.0 | 3.2 | 4.0 | 3.8 | 3.3 | 3.8 | 4.4 | 4.4 | **4.9** | 4.6 | 4.4 | 4.3 |
| Stamps | 63.7 | 54.2 | 44.2 | 51.0 | 50.9 | 47.4 | 52.8 | 61.8 | 26.8 | 45.2 | 56.2 | 33.6 | 52.8 | 48.4 | 26.9 | 31.2 | 52.6 | **79.6** |
| thyroid | 73.8 | 77.4 | 58.8 | 73.9 | 83.7 | 78.3 | 78.9 | 76.4 | 6.2 | 62.9 | 28.8 | 67.6 | 60.6 | 71.9 | **86.4** | 58.5 | 74.8 | 84.6 |
| vertebral | 19.7 | 20.3 | 19.6 | 23.1 | 19.4 | 21.6 | 20.1 | 25.3 | 29.8 | 19.5 | 26.2 | 21.4 | 21.2 | 20.9 | 27.1 | 21.5 | 16.6 | **38.4** |
| vowels | 77.7 | 76.3 | 74.3 | 44.2 | 25.4 | 69.9 | 6.0 | 63.6 | **87.6** | 14.2 | 84.5 | 76.6 | 64.1 | 56.4 | 79.6 | 42.5 | 58.2 | 59.8 |
| Waveform | 27.6 | 27.0 | 31.7 | 10.7 | 10.8 | 11.1 | 9.5 | 11.3 | **47.4** | 7.6 | 29.6 | 5.7 | 12.0 | 20.0 | 10.3 | 8.0 | 15.8 | 18.4 |
| WBC | 85.5 | 95.7 | 82.3 | 91.2 | 97.8 | 92.1 | 94.3 | 83.2 | 26.7 | **99.0** | 24.3 | 91.7 | 86.0 | 93.7 | 77.2 | 86.9 | 87.2 | 94.1 |
| WDBC | **90.9** | 81.5 | 89.9 | 87.7 | 86.5 | 90.0 | 80.8 | 36.0 | 4.0 | 73.9 | 34.3 | 70.3 | 76.4 | 66.8 | 17.1 | 89.8 | 88.1 | 90.2 |
| Wilt | 7.4 | 12.9 | 17.0 | 7.0 | 9.2 | 10.8 | 10.1 | 11.3 | **51.9** | 8.1 | 38.6 | 17.8 | 17.3 | 13.7 | 29.5 | 43.8 | 7.7 | 48.1 |
| wine | 58.2 | 60.8 | 52.7 | 55.0 | 57.3 | 42.4 | 56.5 | 56.6 | 33.8 | 30.5 | 44.4 | 64.0 | 39.8 | 80.2 | 59.6 | 66.2 | 54.5 | **94.8** |
| WPBC | 38.3 | 37.5 | 37.5 | 36.9 | 37.8 | 37.1 | 37.9 | 43.8 | 50.6 | 35.4 | 41.3 | 39 | 39.2 | 39.0 | 36.5 | 35.2 | 36.6 | **65.4** |
| yeast | 48.2 | 49.5 | 49.8 | 48.6 | 48.10.4 | 40.3 | 47.0 | 51.7 | **57.5** | 50 | 55.0 | 53.1 | 52.3 | 49.0 | 51.2 | 56.2 | 46.1 | 56.1 |
| AVG | 62.3 | 62.2 | 57.6 | 54.6 | 48.9 | 58.2 | 51.5 | 53.6 | 48.0 | 45.0 | 49.9 | 58.4 | 58.2 | 59.9 | 57.0 | 59.9 | 54.6 | **64.4** |

NRDE using $\hat{m}$ sources with using different proportions (25%, 50%, 75%, and 100%) of sources with the smallest row norms treated as data sources. As shown in Table 28, the optimal proportions of data sources change for different datasets, yet the performance of NRDE is close to that of using optimal proportions of sources for pure data generation, illustrating the effectiveness of this selection strategy. Meanwhile, using all sources (100%) consistently yields the worst performance across all datasets, which further supports the underlying assumptions and motivations of NRDE.

### I.7. Training Time Cost Comparison between NRDE and RealNVP (Normalizing Flow)

Since NRDE and RealNVP share almost the same inference procedure, we report only their training times across datasets of varying dimensionality in Table 29. The main additional time consumption of NRDE compared to RealNVP is the time required for Jacobian matrix computation for each training sample. The results indicate that, in most cases, even for large-scale or high-dimensional datasets, NRDE's training time remains comparable to RealNVP without a substantial increase.

### I.8. Dynamics of Training Loss and AUROC across the Training Procedure

In this subsection, we provide the dynamics of training loss and AUROC across the training procedure for several datasets in Figure 9. As shown in the figure, the decrease in loss is consistent with the improvement in performance.

*Table 17.* AUROC performance using AutoUAD for hyperparameter tuning on benchmark datasets (best in **bold**).

| AUROC | ADERH | DTE | KNN | LOF | TCCM | NRDE |
|---|---|---|---|---|---|---|
| annthyroid | 90.7 | **97.8** | 94.1 | 92.9 | 89.5 | 95.9 |
| fault | 72.5 | 64.7 | 79.2 | 71.0 | **73.9** | 69.1 |
| glass | **89.8** | 86.8 | 83.5 | 84.0 | 83.3 | 87.4 |
| Ionosphere | 95.1 | 95.0 | **98.0** | 90.4 | 96.3 | 95.4 |
| magic.gamma | 69.8 | **87.7** | 82.9 | 82.7 | 78.0 | 80.5 |
| mnist | 91.3 | 90.1 | **94.3** | 90.6 | 91.2 | 92.3 |
| musk | **100.0** | **100.0** | **100.0** | **100.0** | **100.0** | **100.0** |
| optdigits | 59.6 | 90.0 | 94.9 | **98.7** | 83.1 | 95.6 |
| PageBlocks | 93.9 | 94.7 | 96.2 | **97.4** | 94.3 | 93.4 |
| pendigits | 97.5 | 96.3 | 99.8 | **99.4** | 98.4 | 99.2 |
| satimage-2 | **99.9** | 97.1 | **99.9** | 99.5 | 99.4 | 99.6 |
| shuttle | 99.8 | 99.7 | **99.9** | **99.9** | **99.9** | 99.8 |
| thyroid | 98.3 | 97.5 | 98.5 | 97.7 | 98.2 | **99.3** |
| vertebral | 37.6 | 67.0 | 45.7 | 49.3 | 65.6 | **80.8** |
| vowels | 95.2 | **98.5** | 97.3 | 96.6 | 93.7 | 92.0 |
| Waveform | 68.9 | 62.5 | 76.2 | **76.2** | 74.2 | **85.4** |
| WBC | 98.5 | 89.2 | **99.4** | 74.9 | 97.9 | **99.4** |
| WDBC | 99.3 | 99.2 | 99.1 | 99.7 | 97.7 | **99.8** |
| Wilt | 42.2 | **91.1** | 65.2 | 79.6 | 83.0 | 82.2 |
| wine | 92.2 | 94.3 | 93.3 | 89.2 | 88.5 | **98.5** |
| AVG | 88.6 | 90.0 | 88.8 | 88.5 | 89.3 | **92.2** |

*Table 18.* AUPRC performance using AutoUAD for hyperparameter tuning (best in **bold**).

| AUPRC | ADERH | DTE | KNN | LOF | TCCM | NRDE |
|---|---|---|---|---|---|---|
| annthyroid | 64.5 | **83.3** | 72.0 | 66.7 | 55.9 | 74.5 |
| fault | 71.0 | 64.8 | **76.0** | 68.4 | 73.9 | 72.7 |
| glass | 31.3 | 30.3 | 23.4 | 32.3 | 24.0 | **39.9** |
| Ionosphere | 96.2 | 96.4 | **98.3** | 90.7 | 97.1 | 95.7 |
| magic.gamma | 76.5 | **89.6** | 85.4 | 85.6 | 82.0 | 84.1 |
| mnist | 69.6 | 62.3 | **77.1** | 72.4 | 70.2 | 66.6 |
| musk | **100.0** | **100.0** | **100.0** | **100.0** | **100.0** | **100.0** |
| optdigits | 6.2 | 24.3 | 33.5 | **66.6** | 15.1 | 55.4 |
| PageBlocks | 79.2 | 83.5 | 86.6 | **89.5** | 84.8 | 78.8 |
| pendigits | 60.4 | 37.1 | 95.9 | **84.5** | 70.5 | 79.2 |
| satimage-2 | **98.3** | 31.2 | 98.0 | 90.3 | 88.9 | 95.7 |
| shuttle | 96.9 | 92.6 | 97.5 | 98.8 | **98.9** | 98.5 |
| thyroid | 75.9 | 62.4 | 77.4 | 71.1 | 72.7 | **84.5** |
| vertebral | 17.6 | 36.1 | 20.3 | 24.3 | 33.5 | **50.0** |
| vowels | 63.5 | **89.2** | 76.3 | 76.1 | 76.2 | 63.1 |
| Waveform | 15.5 | 8.6 | 27.0 | **34.8** | 12.7 | 27.4 |
| WBC | 87.4 | 39.3 | **95.7** | 18.3 | 86.2 | 94.3 |
| WDBC | 88.6 | 80.7 | 81.5 | 94.3 | 67.5 | **98.5** |
| Wilt | 8.0 | **38.1** | 12.9 | 26.4 | 31.9 | 37.4 |
| wine | 52.3 | 74.8 | 61.5 | 42.4 | 43.1 | **91.4** |
| AVG | 62.9 | 61.2 | 67.6 | 66.7 | 64.3 | **74.4** |

*Table 19.* Detailed AUROC (%) of each method under anomaly-contamination settings on 10 tabular datasets from ADBench, where the number following each dataset name denotes the anomaly contamination ratio (%). The best and second-best results are highlighted in red and orange, respectively.

| Methods | KDE | KNN | LOF | OC-SVM | IF | AE | DSVDD | RealNVP | NAD | ECOD | ICL | SLAD | DPAD | MCM | DTE-C | TCCM | ADERH | NRDE | NRDE-M |
|---|---|---|---|---|---|---|---|---|---|---|---|---|---|---|---|---|---|---|---|
| breastw1 | 99.2 | 99.2 | 98.5 | 98.9 | 99.5 | 99.1 | 99.3 | 97.0 | 75.6 | 99.4 | 89.7 | 99.1 | 99.2 | 99.4 | 97.3 | 98.8 | 99.1 | 98.8 | 98.9 |
| breastw3 | 99.2 | 99.2 | 98.4 | 98.9 | 99.5 | 99.1 | 99.1 | 97.7 | 73.0 | 99.4 | 91.6 | 99.0 | 98.7 | 99.5 | 96.7 | 99.0 | 99.1 | 99.0 | 99.1 |
| breastw5 | 99.2 | 99.2 | 98.2 | 98.9 | 99.5 | 99.2 | 99.3 | 98.4 | 72.5 | 99.4 | 91.1 | 99.0 | 98.7 | 99.5 | 96.3 | 98.7 | 99.1 | 99.2 | 99.2 |
| breastw10 | 99.2 | 99.2 | 97.0 | 99.0 | 99.5 | 98.7 | 99.0 | 98.5 | 75.0 | 99.3 | 89.9 | 97.3 | 98.5 | 99.0 | 95.0 | 97.5 | 99.1 | 99.5 | 99.4 |
| Cardiotocography1 | 81.6 | 72.0 | 78.0 | 83.3 | 84.7 | 81.8 | 83.9 | 63.0 | 58.2 | 79.8 | 58.9 | 67.8 | 65.6 | 79.9 | 71.8 | 78.8 | 67.9 | 88.7 | 85.4 |
| Cardiotocography3 | 81.5 | 70.3 | 76.5 | 81.3 | 84.9 | 81.6 | 84.2 | 62.8 | 51.2 | 79.7 | 61.3 | 56.2 | 63.1 | 79.6 | 74.5 | 74.0 | 66.1 | 89.2 | 88.1 |
| Cardiotocography5 | 81.1 | 69.3 | 75.2 | 80.2 | 84.0 | 81.1 | 84.4 | 59.5 | 48.3 | 79.5 | 53.5 | 54.8 | 59.3 | 79.2 | 71.3 | 69.2 | 65.8 | 88.1 | 85.4 |
| Cardiotocography10 | 79.5 | 65.4 | 71.5 | 77.7 | 80.0 | 77.6 | 81.4 | 62.5 | 44.6 | 78.9 | 48.8 | 51.4 | 58.6 | 77.1 | 69.9 | 70.9 | 62.7 | 86.9 | 85.1 |
| Ionosphere1 | 97.4 | 98.0 | 96.2 | 96.9 | 92.1 | 98.0 | 96.1 | 90.4 | 96.7 | 75.9 | 96.2 | 96.3 | 88.7 | 96.3 | 96.8 | 96.6 | 96.2 | 96.0 | 81.3 |
| Ionosphere3 | 97.4 | 98.0 | 96.1 | 96.6 | 91.3 | 97.9 | 96.0 | 91.1 | 97.2 | 75.9 | 95.9 | 97.8 | 88.8 | 96.0 | 96.9 | 97.3 | 96.4 | 95.3 | 79.4 |
| Ionosphere5 | 96.7 | 97.7 | 95.7 | 96.3 | 92.0 | 97.5 | 95.6 | 90.9 | 96.1 | 75.8 | 95.2 | 96.0 | 90.9 | 96.2 | 96.8 | 96.5 | 95.4 | 95.1 | 77.6 |
| Ionosphere10 | 96.7 | 97.5 | 95.3 | 95.6 | 90.3 | 96.9 | 95.0 | 91.3 | 95.8 | 75.6 | 94.3 | 96.4 | 91.3 | 95.9 | 96.1 | 96.6 | 95.1 | 94.7 | 75.0 |
| landsat1 | 72.2 | 74.5 | 75.6 | 47.0 | 65.7 | 60.6 | 69.7 | 55.5 | 70.6 | 37.7 | 76.0 | 69.9 | 66.8 | 67.3 | 58.6 | 60.8 | 62.5 | 67.1 | 69.2 |
| landsat3 | 70.8 | 71.9 | 72.4 | 46.4 | 64.3 | 59.9 | 68.8 | 54.0 | 70.0 | 37.5 | 73.8 | 68.4 | 67.9 | 62.1 | 58.2 | 57.4 | 61.7 | 66.4 | 67.5 |
| landsat5 | 69.6 | 70.3 | 69.8 | 46.0 | 64.1 | 59.8 | 67.9 | 53.6 | 68.6 | 37.4 | 71.0 | 69.3 | 67.3 | 59.2 | 58.6 | 56.5 | 62.2 | 67.3 | 73.7 |
| landsat10 | 67.2 | 67.2 | 62.6 | 45.0 | 65.0 | 57.1 | 65.6 | 53.1 | 64.8 | 37.0 | 72.3 | 68.9 | 66.7 | 57.1 | 58.7 | 54.0 | 59.9 | 70.6 | 71.0 |
| magic.gamma1 | 75.6 | 83.0 | 83.9 | 73.6 | 75.7 | 84.2 | 80.2 | 77.4 | 75.8 | 63.6 | 71.7 | 70.4 | 73.9 | 82.1 | 87.3 | 80.0 | 70.8 | 80.3 | 79.4 |
| magic.gamma3 | 75.4 | 82.9 | 83.2 | 72.9 | 77.1 | 83.1 | 79.4 | 75.5 | 74.5 | 63.6 | 69.2 | 71.9 | 71.8 | 81.8 | 87.2 | 77.0 | 70.8 | 80.4 | 77.3 |
| magic.gamma5 | 75.2 | 82.8 | 82.3 | 72.6 | 74.9 | 82.8 | 80.1 | 75.6 | 73.7 | 63.5 | 63.7 | 68.8 | 75.2 | 80.7 | 85.1 | 77.6 | 71.3 | 80.5 | 77.3 |
| magic.gamma10 | 74.7 | 82.5 | 80.5 | 71.6 | 74.5 | 81.7 | 78.7 | 74.0 | 72.5 | 63.4 | 63.0 | 69.4 | 71.0 | 80.0 | 84.6 | 77.6 | 71.1 | 81.1 | 76.5 |
| Pima1 | 78.0 | 78.5 | 76.5 | 75.7 | 76.1 | 73.5 | 73.8 | 76.4 | 62.2 | 61.2 | 67.8 | 61.1 | 68.9 | 68.6 | 71.9 | 72.7 | 76.7 | 79.8 | 80.8 |
| Pima3 | 77.9 | 78.3 | 76.4 | 75.5 | 75.8 | 74.0 | 72.2 | 77.7 | 61.6 | 61.1 | 63.5 | 60.7 | 68.9 | 69.0 | 73.1 | 72.6 | 76.6 | 79.8 | 80.2 |
| Pima5 | 77.7 | 78.2 | 76.2 | 75.2 | 74.4 | 73.2 | 73.0 | 77.0 | 64.1 | 61.0 | 66.8 | 56.6 | 72.3 | 68.6 | 69.8 | 70.6 | 76.6 | 79.2 | 80.7 |
| Pima10 | 77.3 | 78.0 | 75.6 | 74.7 | 74.4 | 72.5 | 72.6 | 75.4 | 63.4 | 60.5 | 63.6 | 59.8 | 68.9 | 68.4 | 70.5 | 70.5 | 76.6 | 78.3 | 80.5 |
| satellite1 | 86.7 | 87.1 | 84.9 | 75.3 | 82.1 | 80.5 | 86.2 | 74.9 | 76.7 | 58.9 | 77.5 | 87.9 | 79.4 | 77.8 | 87.0 | 80.9 | 81.7 | 84.1 | 81.2 |
| satellite3 | 85.9 | 86.4 | 84.4 | 74.7 | 82.5 | 78.8 | 84.0 | 76.1 | 75.3 | 58.8 | 68.6 | 87.7 | 78.1 | 76.1 | 83.5 | 78.6 | 81.8 | 83.7 | 82.3 |
| satellite5 | 85.0 | 85.8 | 83.4 | 74.4 | 82.1 | 77.1 | 83.7 | 77.7 | 73.0 | 58.7 | 64.4 | 87.7 | 78.0 | 75.3 | 79.4 | 77.4 | 81.4 | 84.1 | 79.1 |
| satellite10 | 83.4 | 84.5 | 60.9 | 73.1 | 81.0 | 77.6 | 79.7 | 74.4 | 70.6 | 58.4 | 55.9 | 86.6 | 75.7 | 73.8 | 77.2 | 75.7 | 80.3 | 81.3 | 74.2 |
| SpamBase1 | 84.4 | 84.1 | 81.9 | 81.1 | 79.7 | 81.7 | 82.7 | 79.0 | 79.2 | 66.8 | 78.5 | 83.2 | 74.8 | 74.6 | 84.8 | 85.6 | 80.0 | 81.3 | 70.8 |
| SpamBase3 | 81.2 | 82.9 | 80.4 | 79.4 | 80.4 | 80.2 | 81.8 | 75.4 | 76.2 | 66.7 | 75.2 | 81.2 | 71.4 | 75.6 | 83.0 | 83.6 | 78.4 | 80.9 | 72.3 |
| SpamBase5 | 79.2 | 81.7 | 78.9 | 77.9 | 79.5 | 78.9 | 79.6 | 75.5 | 75.3 | 66.6 | 72.2 | 78.7 | 68.7 | 75.0 | 80.4 | 81.5 | 77.0 | 80.2 | 73.4 |
| SpamBase10 | 74.7 | 78.3 | 74.5 | 73.6 | 78.5 | 74.6 | 75.2 | 73.5 | 70.8 | 66.2 | 65.3 | 73.0 | 63.6 | 73.0 | 77.1 | 77.9 | 72.6 | 78.4 | 73.9 |
| WPBC1 | 52.3 | 51.8 | 53.6 | 51.2 | 53.1 | 52.7 | 49.7 | 54.9 | 55.9 | 46.7 | 56.2 | 52.1 | 50.3 | 56.8 | 51.5 | 52.0 | 48.9 | 64.7 | 65.0 |
| WPBC3 | 51.6 | 52.1 | 52.9 | 50.5 | 55.5 | 55.5 | 52.1 | 54.4 | 54.9 | 46.6 | 52.9 | 53.8 | 50.3 | 57.2 | 51.6 | 49.3 | 49.1 | 64.3 | 64.5 |
| WPBC5 | 51.4 | 52.4 | 53.3 | 50.7 | 54.4 | 52.9 | 52.4 | 55.4 | 55.0 | 46.8 | 60.5 | 53.3 | 51.0 | 56.3 | 50.9 | 51.3 | 49.6 | 67.2 | 64.0 |
| WPBC10 | 50.4 | 51.6 | 51.5 | 49.9 | 53.2 | 52.0 | 51.4 | 54.2 | 54.9 | 46.6 | 55.5 | 53.4 | 49.7 | 53.7 | 53.4 | 49.9 | 48.7 | 65.1 | 63.4 |
| yeast1 | 43.3 | 47.8 | 49.7 | 48.3 | 46.3 | 50.0 | 54.1 | 54.8 | 59.7 | 45.7 | 55.9 | 55.0 | 55.0 | 51.9 | 50.6 | 54.4 | 40.8 | 61.3 | 61.6 |
| yeast3 | 43.0 | 47.1 | 50.6 | 48.7 | 48.0 | 51.4 | 53.6 | 50.3 | 58.9 | 45.7 | 56.6 | 54.2 | 54.6 | 51.2 | 54.4 | 51.1 | 40.3 | 60.4 | 62.7 |
| yeast5 | 42.6 | 46.5 | 49.7 | 48.4 | 45.7 | 50.5 | 49.3 | 48.7 | 58.2 | 45.7 | 56.9 | 53.6 | 54.1 | 52.4 | 48.3 | 52.8 | 40.1 | 61.1 | 62.9 |
| yeast10 | 42.0 | 45.9 | 49.3 | 48.5 | 47.2 | 50.0 | 59.2 | 46.9 | 56.9 | 45.6 | 57.4 | 55.4 | 52.8 | 51.4 | 52.9 | 52.1 | 40.0 | 59.4 | 64.4 |
| AVG1 | 77.1 | 77.6 | 77.9 | 73.1 | 75.5 | 76.2 | 77.6 | 71.5 | 71.1 | 63.6 | 72.8 | 74.3 | 72.3 | 75.5 | 75.8 | 76.1 | 72.5 | 80.2 | 77.4 |
| AVG3 | 76.4 | 76.9 | 77.1 | 72.5 | 75.9 | 76.2 | 77.1 | 71.5 | 69.3 | 63.5 | 70.9 | 73.1 | 71.4 | 74.8 | 75.9 | 74.0 | 72.0 | 79.9 | 77.3 |
| AVG5 | 75.8 | 76.4 | 76.3 | 72.1 | 75.1 | 75.3 | 76.5 | 71.2 | 68.5 | 63.4 | 69.5 | 71.8 | 71.5 | 74.2 | 73.7 | 73.2 | 71.8 | 80.2 | 77.3 |
| AVG10 | 74.5 | 75.0 | 71.9 | 70.9 | 74.4 | 73.9 | 75.8 | 70.4 | 66.9 | 63.1 | 66.6 | 71.2 | 69.7 | 72.9 | 73.5 | 72.3 | 70.6 | 79.5 | 76.3 |

*(a)* glass      *(b)* optdigits      *(c)* pendigits

*Figure 9.* Dynamics of training loss and AUROC across the training procedure for several datasets.

*Table 20.* Detailed AUPRC (%) of each method under anomaly-contamination settings on 10 tabular datasets from ADBench, where the number following each dataset name denotes the anomaly contamination ratio (%). The best and second-best results are highlighted in red and orange, respectively.

| Methods | KDE | KNN | LOF | OC-SVM | IF | AE | DSVDD | RealNVP | NAD | ECOD | ICL | SLAD | DPAD | MCM | DTE-C | TCCM | ADERH | NRDE | NRDE-M |
|---|---|---|---|---|---|---|---|---|---|---|---|---|---|---|---|---|---|---|---|
| breastw1 | 99.1 | 99.1 | 98.1 | 98.6 | 99.4 | 99.0 | 99.3 | 94.3 | 64.2 | 99.4 | 83.7 | 99.0 | 99.1 | 99.4 | 94.7 | 98.5 | 98.9 | 98.0 | 98.5 |
| breastw3 | 99.1 | 99.1 | 97.5 | 98.7 | 99.5 | 99.0 | 99.1 | 95.6 | 60.7 | 99.4 | 85.8 | 99.0 | 98.3 | 99.4 | 92.2 | 98.8 | 99.0 | 98.7 | 98.7 |
| breastw5 | 99.1 | 99.1 | 95.9 | 98.7 | 99.4 | 99.1 | 99.2 | 97.0 | 59.1 | 99.3 | 86.2 | 98.9 | 98.4 | 99.5 | 91.5 | 98.3 | 99.0 | 98.8 | 98.9 |
| breastw10 | 99.1 | 99.1 | 91.6 | 98.9 | 99.4 | 98.3 | 99.1 | 97.1 | 61.0 | 99.3 | 82.4 | 96.7 | 98.1 | 98.9 | 90.0 | 96.3 | 98.9 | 99.4 | 99.2 |
| Cardiotocography1 | 67.6 | 59.3 | 62.6 | 71.3 | 72.3 | 68.3 | 72.1 | 42.4 | 39.5 | 63.2 | 52.1 | 59.5 | 53.8 | 62.8 | 51.4 | 67.3 | 57.7 | 81.7 | 75.2 |
| Cardiotocography3 | 67.2 | 57.1 | 60.0 | 67.9 | 70.0 | 67.7 | 71.5 | 45.5 | 33.4 | 62.9 | 46.0 | 48.7 | 49.3 | 62.2 | 59.1 | 61.6 | 55.0 | 80.0 | 79.5 |
| Cardiotocography5 | 66.6 | 56.1 | 58.3 | 65.7 | 70.3 | 66.8 | 71.3 | 39.4 | 32.6 | 62.7 | 40.5 | 43.6 | 44.3 | 61.7 | 53.6 | 56.8 | 55.3 | 75.2 | 72.8 |
| Cardiotocography10 | 64.0 | 52.1 | 53.3 | 61.2 | 65.8 | 61.3 | 69.4 | 40.6 | 31.5 | 61.9 | 34.6 | 42.2 | 43.0 | 60.2 | 50.5 | 56.5 | 51.4 | 77.2 | 75.6 |
| Ionosphere1 | 97.7 | 98.2 | 96.5 | 97.4 | 92.0 | 98.4 | 96.3 | 85.2 | 97.1 | 76.8 | 96.2 | 96.9 | 89.4 | 96.8 | 97.0 | 97.2 | 96.7 | 95.8 | 85.0 |
| Ionosphere3 | 97.7 | 98.2 | 96.4 | 97.3 | 90.2 | 98.1 | 96.5 | 86.1 | 97.2 | 76.8 | 95.9 | 98.1 | 89.2 | 96.7 | 97.5 | 97.6 | 96.8 | 93.8 | 83.8 |
| Ionosphere5 | 97.1 | 97.8 | 96.0 | 97.0 | 92.5 | 97.8 | 96.3 | 85.9 | 96.3 | 76.7 | 95.6 | 96.6 | 92.0 | 96.8 | 97.1 | 96.9 | 96.1 | 94.9 | 82.6 |
| Ionosphere10 | 97.1 | 97.7 | 95.5 | 96.4 | 90.3 | 97.2 | 94.9 | 86.4 | 96.0 | 76.5 | 95.0 | 96.8 | 91.7 | 96.6 | 96.2 | 97.2 | 95.7 | 93.6 | 80.8 |
| landsat1 | 48.0 | 51.0 | 65.2 | 31.4 | 41.1 | 38.1 | 41.9 | 40.0 | 43.3 | 24.3 | 64.6 | 43.4 | 42.3 | 44.6 | 33.6 | 36.1 | 35.0 | 41.1 | 54.8 |
| landsat3 | 45.5 | 47.1 | 56.6 | 30.6 | 39.2 | 37.5 | 41.0 | 37.9 | 43.1 | 24.2 | 58.3 | 40.7 | 43.1 | 39.3 | 33.5 | 34.9 | 34.3 | 41.8 | 51.9 |
| landsat5 | 43.7 | 43.4 | 51.3 | 30.0 | 39.2 | 35.8 | 40.6 | 36.3 | 42.2 | 24.1 | 57.0 | 43.3 | 41.5 | 36.6 | 33.2 | 34.0 | 35.7 | 47.0 | 57.7 |
| landsat10 | 40.8 | 40.1 | 42.3 | 28.8 | 37.7 | 33.8 | 38.1 | 35.8 | 39.7 | 23.9 | 57.5 | 40.0 | 42.0 | 35.1 | 33.6 | 30.9 | 33.2 | 43.8 | 58.7 |
| magic.gamma1 | 78.9 | 84.3 | 85.5 | 77.5 | 77.4 | 85.7 | 83.0 | 80.4 | 76.2 | 65.8 | 75.7 | 75.1 | 76.0 | 84.6 | 88.4 | 82.7 | 75.5 | 82.8 | 83.3 |
| magic.gamma3 | 78.7 | 84.2 | 84.1 | 76.6 | 78.0 | 84.6 | 81.7 | 78.3 | 74.5 | 65.8 | 72.7 | 76.1 | 73.7 | 84.0 | 87.7 | 79.9 | 75.5 | 82.2 | 81.4 |
| magic.gamma5 | 78.5 | 84.1 | 83.4 | 76.4 | 76.6 | 84.1 | 82.6 | 78.6 | 72.1 | 65.8 | 66.0 | 73.2 | 77.9 | 83.2 | 86.0 | 79.1 | 75.7 | 82.7 | 81.3 |
| magic.gamma10 | 78.0 | 83.7 | 80.9 | 75.8 | 76.3 | 82.8 | 81.3 | 77.6 | 71.0 | 65.6 | 65.0 | 73.7 | 73.4 | 82.3 | 85.1 | 79.3 | 75.6 | 82.6 | 80.8 |
| Pima1 | 74.8 | 75.8 | 73.5 | 73.6 | 75.3 | 71.9 | 72.6 | 72.0 | 59.0 | 63.7 | 66.6 | 61.1 | 66.8 | 70.5 | 70.9 | 70.7 | 73.7 | 77.7 | 79.3 |
| Pima3 | 74.6 | 75.5 | 73.2 | 73.4 | 74.6 | 73.0 | 71.9 | 73.5 | 57.2 | 63.6 | 61.0 | 61.2 | 67.1 | 69.0 | 71.3 | 73.0 | 73.6 | 77.9 | 79.1 |
| Pima5 | 74.3 | 75.3 | 72.8 | 73.0 | 74.0 | 70.3 | 72.5 | 71.5 | 61.2 | 63.4 | 66.3 | 58.2 | 68.5 | 69.5 | 67.7 | 68.3 | 73.1 | 76.3 | 80.2 |
| Pima10 | 73.7 | 74.9 | 72.1 | 72.5 | 73.5 | 71.0 | 71.5 | 71.4 | 59.4 | 63.0 | 62.3 | 60.3 | 66.1 | 69.5 | 68.9 | 69.1 | 73.3 | 76.8 | 79.5 |
| satellite1 | 87.0 | 87.9 | 87.2 | 81.1 | 83.6 | 81.8 | 85.0 | 79.5 | 68.6 | 64.2 | 79.1 | 86.9 | 75.3 | 77.8 | 86.1 | 83.6 | 83.6 | 85.0 | 84.7 |
| satellite3 | 85.5 | 86.8 | 86.2 | 80.7 | 84.1 | 80.1 | 81.3 | 80.9 | 66.1 | 64.0 | 69.4 | 86.7 | 72.7 | 76.2 | 83.2 | 80.7 | 83.4 | 85.2 | 84.2 |
| satellite5 | 84.2 | 86.6 | 84.7 | 80.3 | 83.6 | 78.8 | 81.5 | 81.7 | 64.6 | 63.9 | 66.0 | 86.6 | 70.8 | 75.3 | 77.3 | 78.7 | 83.1 | 85.0 | 81.9 |
| satellite10 | 81.8 | 85.0 | 63.2 | 79.3 | 82.7 | 79.7 | 80.9 | 78.5 | 61.4 | 63.6 | 61.8 | 85.0 | 69.9 | 73.0 | 75.2 | 76.1 | 82.2 | 84.4 | 77.9 |
| SpamBase1 | 85.7 | 85.2 | 82.0 | 83.3 | 83.6 | 83.4 | 84.5 | 77.6 | 79.0 | 68.2 | 81.8 | 85.8 | 77.8 | 77.2 | 86.0 | 87.1 | 82.1 | 79.6 | 74.8 |
| SpamBase3 | 83.4 | 83.9 | 80.0 | 81.3 | 84.6 | 81.9 | 82.8 | 73.9 | 77.5 | 68.1 | 80.0 | 83.9 | 74.9 | 78.5 | 83.9 | 85.4 | 80.3 | 79.5 | 75.8 |
| SpamBase5 | 81.1 | 82.3 | 77.8 | 79.4 | 83.4 | 80.0 | 80.3 | 73.8 | 75.9 | 67.9 | 75.8 | 81.7 | 71.6 | 78.0 | 80.4 | 83.1 | 78.7 | 78.7 | 77.2 |
| SpamBase10 | 76.8 | 78.5 | 72.9 | 74.7 | 82.5 | 75.3 | 77.7 | 71.7 | 71.2 | 67.5 | 70.1 | 75.8 | 66.3 | 77.0 | 77.5 | 79.2 | 74.1 | 77.4 | 78.4 |
| WPBC1 | 34.2 | 34.0 | 35.1 | 33.7 | 34.5 | 34.3 | 33.6 | 36.5 | 47.2 | 31.3 | 41.2 | 40.4 | 34.7 | 37.1 | 37.9 | 34.3 | 32.6 | 46.3 | 52.9 |
| WPBC3 | 33.9 | 34.1 | 34.6 | 33.2 | 35.5 | 38.0 | 35.3 | 37.0 | 38.6 | 31.2 | 37.3 | 39.4 | 35.4 | 39.9 | 34.2 | 33.4 | 32.5 | 43.6 | 51.8 |
| WPBC5 | 33.9 | 34.4 | 34.4 | 33.5 | 34.7 | 35.3 | 35.9 | 37.1 | 41.8 | 31.3 | 41.0 | 38.8 | 33.7 | 38.5 | 34.4 | 33.9 | 32.9 | 51.5 | 49.9 |
| WPBC10 | 33.3 | 33.8 | 34.0 | 32.8 | 33.8 | 36.7 | 34.3 | 36.8 | 42.6 | 31.2 | 37.6 | 38.0 | 34.3 | 36.3 | 36.4 | 33.4 | 32.2 | 45.9 | 50.8 |
| yeast1 | 46.0 | 47.3 | 50.3 | 48.8 | 48.8 | 50.6 | 50.7 | 46.2 | 54.7 | 48.0 | 54.5 | 52.0 | 51.7 | 49.6 | 49.6 | 51.1 | 44.2 | 55.5 | 59.4 |
| yeast3 | 45.8 | 47.0 | 50.4 | 49.0 | 49.8 | 50.1 | 50.1 | 48.4 | 54.7 | 48.0 | 53.6 | 51.5 | 52.1 | 49.6 | 50.2 | 50.0 | 44.0 | 56.2 | 61.2 |
| yeast5 | 45.6 | 46.8 | 50.0 | 48.8 | 48.5 | 49.9 | 48.3 | 47.8 | 53.6 | 47.9 | 53.9 | 50.7 | 51.9 | 51.0 | 48.1 | 50.4 | 43.8 | 55.8 | 61.7 |
| yeast10 | 45.2 | 46.4 | 49.5 | 48.7 | 49.2 | 49.6 | 54.1 | 47.1 | 53.0 | 47.9 | 55.1 | 52.3 | 49.4 | 49.8 | 51.4 | 49.7 | 43.8 | 54.4 | 61.4 |
| AVG1 | 71.9 | 72.2 | 73.6 | 69.7 | 70.8 | 71.2 | 71.9 | 65.4 | 62.9 | 60.5 | 69.5 | 70.0 | 66.7 | 70.0 | 69.6 | 70.9 | 68.0 | 74.3 | 74.8 |
| AVG3 | 71.1 | 71.3 | 71.9 | 68.9 | 70.5 | 71.0 | 71.1 | 65.7 | 60.3 | 60.4 | 66.0 | 68.5 | 65.6 | 69.5 | 69.3 | 69.3 | 67.4 | 73.9 | 74.7 |
| AVG5 | 70.4 | 70.6 | 70.5 | 68.3 | 70.2 | 69.8 | 70.8 | 64.9 | 59.9 | 60.3 | 64.8 | 67.2 | 65.1 | 69.0 | 66.9 | 68.0 | 67.3 | 74.6 | 74.4 |
| AVG10 | 69.0 | 69.1 | 65.5 | 66.9 | 69.1 | 68.6 | 70.1 | 64.3 | 58.7 | 60.0 | 62.1 | 66.1 | 63.4 | 67.9 | 66.5 | 66.7 | 66.0 | 73.5 | 74.3 |

*Table 21.* Detailed AUROC (%) and AUPRC (%) of each method on anomaly detection with injected noise experiments of 10 tabular datasets of ADBench where test normal data are injected with stronger noise. The best and second-best results are highlighted in red and orange, respectively.

| AUROC | KDE | KNN | LOF | OC-SVM | IF | AE | DSVDD | RealNVP | NAD | ECOD | ICL | SLAD | DPAD | MCM | DTE-C | TCCM | ADERH | NRDE | NRDE-M |
|---|---|---|---|---|---|---|---|---|---|---|---|---|---|---|---|---|---|---|---|
| breastw | 99.1 | 99.2 | 98.8 | 98.9 | 99.5 | 99.0 | 99.3 | 96.7 | 66.4 | 76.7 | 94.4 | 98.4 | 99.0 | 99.2 | 97.5 | 98.8 | 98.9 | 98.7 | 99.0 |
| Cardiotocography | 77.7 | 68.7 | 71.6 | 74.7 | 70.4 | 75.1 | 83.6 | 50.6 | 40.8 | 64.2 | 60.2 | 59.0 | 59.5 | 67.0 | 56.4 | 75.5 | 62.7 | 84.3 | 80.7 |
| Ionosphere | 94.3 | 96.8 | 95.9 | 95.4 | 89.4 | 95.9 | 77.1 | 77.0 | 91.6 | 73.9 | 93.3 | 93.4 | 87.8 | 92.3 | 96.5 | 93.6 | 92.6 | 94.4 | 84.3 |
| landsat | 47.2 | 47.1 | 44.8 | 35.2 | 46.6 | 26.6 | 49.0 | 50.0 | 55.2 | 30.7 | 62.8 | 61.5 | 54.4 | 39.8 | 12.8 | 42.9 | 35.9 | 70.1 | 69.2 |
| magic.gamma | 68.5 | 68.7 | 68.2 | 67.2 | 67.1 | 68.0 | 69.5 | 67.6 | 65.0 | 59.1 | 64.6 | 65.2 | 67.2 | 68.9 | 67.5 | 67.0 | 62.9 | 82.2 | 76.1 |
| Pima | 74.5 | 73.0 | 72.3 | 71.4 | 70.5 | 70.1 | 70.8 | 69.2 | 54.9 | 56.7 | 61.3 | 55.2 | 71.5 | 69.8 | 67.4 | 68.5 | 71.8 | 79.6 | 76.7 |
| satellite | 74.6 | 75.1 | 70.6 | 67.0 | 71.5 | 67.2 | 74.8 | 65.2 | 64.6 | 55.0 | 77.6 | 80.6 | 75.0 | 66.7 | 56.9 | 70.9 | 68.4 | 84.7 | 79.5 |
| SpamBase | 80.8 | 80.5 | 79.4 | 78.8 | 58.1 | 79.3 | 78.6 | 58.9 | 68.3 | 11.5 | 77.4 | 79.9 | 72.3 | 76.2 | 78.8 | 77.4 | 78.9 | 83.6 | 78.0 |
| WPBC | 47.3 | 45.2 | 47.9 | 44.5 | 45.5 | 44.8 | 51.0 | 50.0 | 45.5 | 39.7 | 50.6 | 51.3 | 50.5 | 34.1 | 41.8 | 35.9 | 44.1 | 75.7 | 70.5 |
| yeast | 38.6 | 37.5 | 39.5 | 40.0 | 35.9 | 42.9 | 40.4 | 39.2 | 53.1 | 33.5 | 51.0 | 50.1 | 49.1 | 40.7 | 41.5 | 47.5 | 35.2 | 64.3 | 61.3 |
| AVG | 70.3 | 69.2 | 68.9 | 67.3 | 65.4 | 66.9 | 69.4 | 62.4 | 60.8 | 50.1 | 69.3 | 69.5 | 68.6 | 65.5 | 61.7 | 67.8 | 65.1 | 81.8 | 77.5 |
| **AURPC** | | | | | | | | | | | | | | | | | | | |
| breastw | 99.1 | 99.2 | 98.7 | 98.8 | 99.5 | 99.0 | 99.2 | 94.4 | 64.1 | 77.9 | 93.2 | 98.4 | 98.9 | 99.1 | 94.9 | 98.7 | 98.8 | 98.6 | 99.0 |
| Cardiotocography | 68.6 | 62.2 | 62.3 | 66.7 | 58.7 | 67.1 | 76.2 | 36.4 | 31.6 | 50.8 | 51.3 | 52.6 | 49.3 | 61.7 | 50.6 | 68.2 | 58.6 | 76.1 | 67.0 |
| Ionosphere | 95.9 | 97.4 | 96.8 | 96.6 | 91.0 | 96.8 | 78.2 | 71.2 | 92.3 | 78.8 | 92.2 | 95.0 | 89.4 | 95.1 | 97.2 | 95.7 | 94.2 | 95.0 | 89.0 |
| landsat | 34.2 | 33.4 | 38.6 | 28.1 | 31.9 | 24.4 | 35.4 | 34.3 | 37.5 | 25.6 | 55.7 | 40.7 | 39.3 | 27.8 | 20.7 | 33.4 | 26.1 | 55.7 | 51.6 |
| magic.gamma | 76.1 | 76.2 | 75.9 | 74.7 | 73.1 | 75.5 | 76.2 | 74.6 | 67.6 | 64.9 | 71.3 | 72.6 | 73.9 | 76.0 | 75.5 | 73.0 | 72.1 | 85.3 | 79.1 |
| Pima | 73.6 | 73.5 | 71.9 | 72.0 | 71.9 | 69.5 | 71.2 | 69.6 | 55.2 | 56.6 | 61.3 | 56.9 | 72.2 | 70.7 | 67.8 | 69.1 | 71.9 | 79.8 | 77.8 |
| satellite | 81.3 | 81.3 | 79.8 | 77.9 | 78.7 | 76.8 | 79.2 | 64.2 | 64.3 | 64.8 | 81.6 | 82.8 | 75.5 | 74.2 | 70.1 | 77.4 | 75.8 | 87.3 | 83.3 |
| SpamBase | 85.2 | 84.7 | 83.2 | 83.6 | 69.6 | 83.8 | 81.5 | 62.8 | 71.9 | 38.2 | 84.1 | 84.7 | 76.0 | 84.2 | 83.4 | 83.8 | 83.2 | 85.3 | 79.9 |
| WPBC | 36.2 | 35.7 | 36.3 | 34.9 | 35.0 | 35.9 | 39.1 | 38.2 | 37.4 | 32.5 | 40.5 | 40.4 | 38.9 | 30.0 | 34.8 | 30.9 | 34.5 | 67.5 | 64.2 |
| yeast | 44.8 | 44.0 | 45.4 | 45.9 | 43.3 | 48.5 | 45.1 | 44.6 | 51.8 | 41.5 | 51.8 | 50.9 | 50.8 | 44.9 | 45.7 | 48.6 | 43.6 | 60.4 | 57.6 |
| AVG | 69.5 | 68.8 | 68.9 | 67.9 | 65.3 | 67.7 | 68.2 | 59.0 | 57.4 | 53.2 | 68.3 | 67.5 | 66.4 | 66.4 | 64.1 | 67.9 | 65.9 | 79.1 | 74.8 |

*Table 22.* Detailed AUROC (%) of each method on anomaly detection with injected noise experiments of 10 tabular datasets of ADBench where test anomalous data are injected with stronger noise. The best and second-best results are highlighted in red and orange, respectively. * indicates that we report the performance in their paper.

| Methods | KDE | KNN | LOF | OC-SVM | IF | AE | DSVDD | RealNVP | NAD | ECOD | ICL | SLAD | DPAD | MCM | DTE-C | NRDE | NRDE-M |
|---|---|---|---|---|---|---|---|---|---|---|---|---|---|---|---|---|---|
| breastw | 99.1 | 99.1 | 98.6 | 98.9 | 99.6 | 99.1 | 99.2 | 94.8 | 64.8 | 86.9 | 92.1 | 98.9 | 99.0 | 99.3 | 97.8 | 98.9 | 99.2 |
| Cardiotocography | 73.4 | 68.8 | 78.1 | 78.5 | 79.1 | 75.1 | 75.8 | 36.1 | 40.5 | 74.3 | 59.9 | 62.7 | 51.1 | 76.9 | 72.1 | 83.6 | 86.0 |
| Ionosphere | 98.4 | 98.7 | 97.3 | 98.1 | 93.2 | 98.5 | 77.6 | 88.0 | 95.6 | 85.0 | 94.6 | 97.7 | 89.5 | 98.3 | 98.2 | 95.4 | 84.0 |
| landsat | 72.2 | 72.3 | 89.4 | 51.0 | 40.0 | 76.7 | 44.7 | 52.6 | 55.4 | 29.4 | 65.4 | 57.6 | 54.6 | 72.9 | 85.7 | 77.6 | 70.0 |
| magic.gamma | 82.6 | 86.3 | 88.4 | 83.1 | 79.7 | 87.9 | 82.8 | 84.4 | 75.0 | 70.7 | 74.2 | 76.3 | 76.7 | 86.0 | 88.7 | 85.2 | 81.7 |
| Pima | 79.2 | 79.5 | 76.6 | 76.8 | 80.2 | 75.3 | 75.9 | 51.7 | 61.2 | 70.7 | 68.3 | 58.5 | 75.1 | 76.6 | 76.7 | 80.5 | 79.1 |
| satellite | 94.1 | 94.3 | 95.6 | 87.9 | 84.3 | 94.7 | 82.8 | 48.1 | 74.3 | 68.1 | 79.4 | 90.9 | 87.2 | 95.0 | 95.6 | 91.2 | 82.8 |
| SpamBase | 89.5 | 88.0 | 87.4 | 86.6 | 96.5 | 87.6 | 84.4 | 59.8 | 78.7 | 91.2 | 87.9 | 87.6 | 74.4 | 92.4 | 89.5 | 90.5 | 84.2 |
| WPBC | 42.6 | 41.5 | 44.1 | 41.3 | 41.1 | 46.7 | 44.0 | 81.0 | 47.2 | 37.6 | 45.5 | 43.7 | 39.3 | 51.1 | 64.1 | 75.6 | 69.1 |
| yeast | 52.9 | 55.3 | 61.4 | 57.1 | 64.1 | 63.4 | 50.6 | 55.9 | 63.9 | 61.2 | 58.9 | 57.7 | 56.6 | 58.5 | 65.5 | 61.1 | 60.3 |
| AVG | 78.4 | 78.4 | 81.7 | 75.9 | 75.8 | 80.5 | 71.8 | 65.2 | 65.6 | 67.5 | 72.6 | 73.2 | 70.4 | 80.7 | 83.4 | 84.0 | 79.6 |

*Table 23.* Detailed AUROC and AUPRC performance of outlier detection on 10 datasets. The best and second-best results are highlighted in red and orange, respectively. * indicates that we report the experimental results from their original paper.

| AUROC | KDE | KNN | LOF | OC-SVM | IF | AE | DSVDD | RealNVP | NAD | ECOD | ICL | SLAD | DPAD | MCM | DTE-C | TCCM | ADERH | ODIM* | NRDE | NRDE-M |
|---|---|---|---|---|---|---|---|---|---|---|---|---|---|---|---|---|---|---|---|---|
| breastw | 3.8 | 98.4 | 48.5 | 99.1 | 99.1 | 95.9 | 84.5 | 98.2 | 90.3 | 99.1 | 66.7 | 80.6 | 90.9 | 99.4 | 93.3 | 93.3 | 98.9 | 99.1 | 99.2 | 98.9 |
| Cardiotocography | 49.7 | 51.9 | 54.7 | 74.1 | 75.2 | 59.5 | 75.2 | 65.7 | 61.7 | 78.5 | 45.4 | 50.5 | 54.3 | 78.9 | 58.2 | 65.6 | 59.2 | 61.0 | 78.1 | 79.3 |
| Ionosphere | 19.9 | 92.8 | 89.1 | 85.2 | 75.9 | 93.9 | 57.7 | 92.6 | 88.7 | 72.8 | 69.8 | 85.1 | 68.1 | 96.4 | 95.0 | 94.4 | 91.7 | 84.8 | 95.4 | 90.4 |
| landsat | 63.6 | 61.8 | 54.7 | 42.4 | 57.0 | 53.9 | 65.3 | 51.3 | 62.8 | 36.8 | 64.1 | 67.6 | 57.5 | 66.0 | 58.8 | 54.5 | 61.5 | 46.2 | 72.9 | 68.2 |
| magic.gamma | 33.0 | 81.1 | 71.7 | 68.6 | 72.3 | 76.9 | 72.1 | 69.9 | 68.3 | 63.8 | 58.7 | 66.2 | 62.9 | 83.2 | 77.7 | 71.2 | 75.0 | 74.5 | 80.8 | 73.8 |
| Pima | 33.9 | 72.7 | 65.1 | 68.6 | 67.8 | 65.9 | 62.8 | 67.9 | 66.1 | 59.4 | 56.1 | 52.3 | 61.2 | 70.7 | 65.0 | 68.6 | 73.6 | 71.9 | 75.0 | 78.8 |
| satellite | 39.3 | 73.1 | 56.4 | 66.4 | 73.9 | 71.3 | 68.5 | 72.1 | 65.9 | 58.3 | 54.5 | 80.5 | 65.0 | 81.6 | 74.7 | 76.1 | 80.2 | 69.8 | 81.1 | 78.7 |
| SpamBase | 50.5 | 56.7 | 47.9 | 54.7 | 73.4 | 55.7 | 57.5 | 50.0 | 58.6 | 65.6 | 47.8 | 53.1 | 50.6 | 76.3 | 59.2 | 63.6 | 58.5 | 55.0 | 78.8 | 76.9 |
| WPBC | 51.1 | 52.9 | 52.0 | 49.8 | 52.4 | 51.3 | 49.2 | 56.8 | 49.3 | 48.1 | 54.8 | 53.3 | 52.9 | 56.6 | 52.2 | 55.6 | 54.1 | 51.0 | 73.8 | 76.2 |
| yeast | 61.7 | 40.6 | 46.8 | 45.2 | 44.5 | 46.8 | 44.7 | 41.6 | 50.2 | 44.4 | 57.9 | 55.2 | 51.4 | 51.9 | 43.0 | 48.8 | 38.7 | 65.3 | 58.5 | 61.2 |
| AVG | 40.7 | 68.2 | 58.7 | 65.4 | 69.2 | 67.1 | 63.8 | 66.6 | 66.2 | 62.7 | 57.6 | 64.4 | 61.5 | 76.1 | 67.7 | 69.2 | 69.1 | 65.3 | 79.4 | 78.2 |
| **AUPRC** | | | | | | | | | | | | | | | | | | | | |
| breastw | 21.5 | 95.5 | 31.6 | 98.1 | 98.1 | 88.1 | 86.4 | 94.1 | 71.8 | 98.4 | 63.1 | 71.3 | 79.5 | 99.4 | 79.0 | 83.4 | 97.3 | 98.8 | 98.4 | 97.9 |
| Cardiotocography | 22.9 | 33.6 | 29.5 | 41.3 | 48.5 | 35.0 | 47.9 | 31.6 | 31.8 | 50.5 | 20.7 | 34.0 | 25.2 | 66.2 | 31.1 | 37.2 | 36.9 | 38.9 | 51.7 | 49.4 |
| Ionosphere | 23.3 | 92.9 | 85.0 | 82.4 | 65.6 | 93.2 | 45.3 | 84.9 | 78.7 | 64.6 | 57.9 | 62.9 | 58.9 | 97.1 | 92.7 | 93.2 | 90.5 | 86.7 | 92.1 | 86.4 |
| landsat | 37.1 | 25.8 | 25.2 | 17.5 | 22.8 | 21.7 | 28.8 | 22.7 | 28.5 | 16.3 | 42.7 | 29.1 | 24.9 | 49.0 | 23.9 | 22.0 | 24.2 | 17.9 | 37.1 | 35.5 |
| magic.gamma | 27.1 | 74.1 | 56.4 | 62.1 | 62.8 | 69.2 | 63.1 | 63.4 | 52.3 | 53.4 | 44.7 | 59.2 | 50.8 | 86.7 | 66.5 | 59.3 | 68.2 | 69.3 | 71.8 | 62.9 |
| Pima | 27.7 | 52.9 | 45.1 | 50.4 | 51.3 | 48.6 | 47.0 | 48.2 | 47.9 | 46.4 | 41.4 | 38.1 | 43.1 | 73.9 | 45.4 | 51.5 | 53.7 | 49.1 | 59.4 | 61.6 |
| satellite | 34.2 | 59.2 | 39.3 | 65.5 | 65.8 | 58.3 | 54.7 | 72.2 | 42.7 | 52.6 | 41.3 | 63.6 | 48.1 | 86.5 | 57.1 | 64.5 | 69.3 | 65.2 | 74.0 | 67.2 |
| SpamBase | 37.6 | 41.7 | 38.9 | 40.8 | 62.2 | 41.7 | 42.9 | 39.9 | 43.3 | 51.8 | 37.0 | 40.2 | 39.6 | 80.4 | 44.2 | 49.0 | 42.8 | 41.0 | 68.0 | 63.7 |
| WPBC | 23.8 | 23.9 | 23.2 | 22.9 | 24.3 | 26.1 | 23.4 | 26.3 | 25.9 | 21.8 | 30.9 | 25.1 | 25.2 | 42.0 | 27.7 | 26.1 | 24.2 | 23.6 | 52.8 | 49.6 |
| yeast | 48.0 | 30.1 | 32.7 | 31.5 | 34.3 | 32.0 | 30.8 | 29.3 | 34.4 | 33.3 | 41.2 | 38.2 | 36.8 | 52.4 | 30.8 | 32.9 | 29.3 | 28.7 | 38.7 | 40.1 |
| AVG | 30.3 | 53.0 | 40.7 | 51.3 | 53.6 | 51.4 | 47.0 | 51.3 | 45.7 | 48.9 | 42.1 | 46.2 | 43.2 | 73.3 | 49.9 | 51.9 | 53.6 | 51.9 | 64.4 | 61.4 |

*Table 24.* Detailed AUROC and AUPRC performance of NRDE with different values of learning rate $lr$.

| AUROC | Cardiotocography | SpamBase | Pima | Satellite | WPBC |
|---|---|---|---|---|---|
| $lr = 0.001$ | 85.4 | 80.8 | 80.0 | 85.8 | 59.2 |
| $lr = 0.005$ | 87.1 | 79.1 | 78.3 | 77.3 | 64.4 |
| $lr = 0.01$ | 81.6 | 76.8 | 76.4 | 61.3 | 57.2 |
| **AUPRC** | **Cardiotocography** | **SpamBase** | **Pima** | **Satellite** | **WPBC** |
| $lr = 0.001$ | 79.0 | 80.9 | 79.5 | 87.9 | 45.2 |
| $lr = 0.005$ | 79.6 | 77.7 | 79.0 | 77.8 | 49.3 |
| $lr = 0.01$ | 67.5 | 77.1 | 77.0 | 59.5 | 42.8 |

## I.9. Experimental results of using dimension reduction methods for high-dimensional data before NRDE

The main computational bottleneck of NRDE compared to RealNVP is the Jacobian matrix computation for each training sample, and this time-cost gap increases with data dimensionality. However, NRDE's inference time remains virtually identical to standard RealNVP. The overhead may become prohibitive on high-dimensional data. However, for such high-dimensional cases, the computational cost can be easily mitigated by employing dimensionality reduction methods like PCA before applying NRDE. Table 30 demonstrates that NRDE remains effective even on low-dimensional (128) data processed via Kernel PCA.

*Table 25.* Detailed AUROC and AUPRC performance of NRDE with different values of hyperparameter $\lambda$.

| AUROC | Cardiotocography | SpamBase | Pima | Satellite | WPBC |
|---|---|---|---|---|---|
| $\lambda = 0$ | 85.9 | 84.2 | 76.7 | 79.4 | 61.6 |
| $\lambda = 0.01$ | 86.9 | 80.7 | 80.0 | 85.8 | 63.8 |
| $\lambda = 0.1$ | 87.1 | 80.2 | 79.2 | 83.1 | 64.4 |
| $\lambda = 1$ | 82.1 | 77.1 | 79.1 | 84.0 | 64.0 |
| AUPRC | Cardiotocography | SpamBase | Pima | Satellite | WPBC |
| $\lambda = 0$ | 78.3 | 85.2 | 80.4 | 78.2 | 47.0 |
| $\lambda = 0.01$ | 80.9 | 80.9 | 79.5 | 87.9 | 51.2 |
| $\lambda = 0.1$ | 79.5 | 80.7 | 79.8 | 85.8 | 49.3 |
| $\lambda = 1$ | 66.3 | 76.8 | 79.9 | 86.4 | 48.5 |

*Table 26.* Detailed performance of NRDE with different numbers of coupling layers (T).

| AUROC | Cardiotocography | SpamBase | Pima | Satellite | WPBC |
|---|---|---|---|---|---|
| $T = 2$ | 87.1 | 80.8 | 80.0 | 85.8 | 64.4 |
| $T = 3$ | 83.3 | 83.8 | 80.4 | 85.3 | 71.1 |
| $T = 4$ | 82.2 | 84.2 | 80.6 | 83.3 | 62.6 |
| AUPRC | Cardiotocography | SpamBase | Pima | Satellite | WPBC |
| $T = 2$ | 79.5 | 80.9 | 79.5 | 87.9 | 49.4 |
| $T = 3$ | 71.9 | 85.1 | 81.7 | 87.2 | 60.0 |
| $T = 4$ | 65.3 | 84.8 | 80.1 | 86.0 | 52.2 |

*Table 27.* Detailed performance of NRDE with different width of coupling layers (b).

| AUROC | Cardiotocography | SpamBase | Pima | Satellite | WPBC |
|---|---|---|---|---|---|
| $b = 512$ | 85.3 | 81.0 | 78.8 | 84.4 | 71.5 |
| $b = 1024$ | 84.2 | 81.6 | 80.2 | 84.2 | 73.6 |
| $b = 2048$ | 87.1 | 80.8 | 80.0 | 85.8 | 64.4 |
| AUPRC | Cardiotocography | SpamBase | Pima | Satellite | WPBC |
| $b = 512$ | 77.2 | 81.1 | 78.5 | 86.8 | 64.8 |
| $b = 1024$ | 75.2 | 81.5 | 80.8 | 86.9 | 67.1 |
| $b = 2048$ | 79.5 | 80.9 | 79.5 | 87.9 | 49.4 |

*Table 28.* Detailed performance of the proposed NRDE using different proportions of sources as data sources. The best performance is marked in red.

| AUROC | Cardiotocography | SpamBase | Pima | Satellite | WPBC |
|---|---|---|---|---|---|
| NRDE (25%) | 86.7 | 81.5 | 79.8 | 82.5 | 63.4 |
| NRDE (50%) | 87.3 | 82.3 | 81.3 | 84.3 | 61.4 |
| NRDE (75%) | 86.2 | 81.8 | 81.1 | 83.0 | 62.0 |
| NRDE (100%) | 81.1 | 81.3 | 78.1 | 81.7 | 60.2 |
| NRDE | 87.1 | 84.5 | 80.0 | 85.7 | 64.4 |
| AUPRC | Cardiotocography | SpamBase | Pima | Satellite | WPBC |
| NRDE (25%) | 80.3 | 82.6 | 79.4 | 84.9 | 48.0 |
| NRDE (50%) | 76.3 | 82.2 | 81.4 | 86.9 | 45.7 |
| NRDE (75%) | 73.9 | 82.0 | 79.7 | 86.1 | 47.1 |
| NRDE (100%) | 61.9 | 81.4 | 75.8 | 84.9 | 45.0 |
| NRDE | 79.5 | 85.3 | 79.5 | 87.9 | 49.3 |

*Table 29.* 100-epoch training time cost ($s$) comparison between NRDE and RealNVP on different dimensional datasets from ADBench.

| Dataset | NRDE | RealNVP |
|---|---|---|
| annthyroid | 4.12 | 2.03 |
| glass | 1.58 | 0.59 |
| mammography | 5.46 | 2.72 |
| Pima | 1.68 | 0.61 |
| vertebral | 1.45 | 0.59 |
| Cardiotocography | 4.29 | 0.90 |
| fraud | 388.80 | 75.21 |
| satellite | 30.31 | 18.8 |
| satimage-2 | 31.40 | 19.04 |
| shuttle | 46.26 | 30.05 |
| mnist | 71.34 | 20.84 |
| musk | 70.41 | 18.03 |

*Table 30.* Experimental results of using dimension reduction methods for high-dimensional data before NRDE

| Data (Dimension) | Original | Low-dimensional (128) |
|---|---|---|
| Speech (400) | 58.7 | 56.5 |
| InternetAds (1555) | 83.9 | 80.0 |

