# OpenReview forum: "Noise-Robust Density Estimation for Tabular Data Anomaly Detection"
_ICML.cc/2026/Conference — ICML 2026 regular_

### Official Review · Reviewer_PoAr · 2026-02-24

**Soundness:** 2
**Presentation:** 2
**Significance:** 3
**Originality:** 3
**Overall Recommendation:** 4
**Confidence:** 3

**Summary:**

This paper proposes a robust anomaly detection method that can learns the densities of pure normal data to detect anomalies while excluding the influence of noise. To realize this, assuming the variance of the pure data source is greater than that of the noise, the authors developed a Jacobian-regularized normalizing flow that incorporates sparse regularization into the normalizing flow to further emphasize the difference between the pure data and noise sources. The experiments using 47 benchmark datasets and 17 baselines show the effectiveness of the proposed approach.

**Compliance With Llm Reviewing Policy:**

Affirmed.

**Final Justification:**

The additional experiments have largely resolved my main concern about the evaluation protocol, and thus, I have raised my rating from 3 to 4. On the other hand, if the experimental results in the submitted paper were based on hyperparameter tuning on test data, the revision required after the rebuttal may be too substantial.

**Key Questions For Authors:**

- Although the proposed method appears to be a general-purpose technique, why did the authors focus only on tabular data? Is this related to the assumptions in Eq. 6?
- Does the computational cost become high when using high-dimensional datasets, since the proposed method is based on a normalizing flow?

**Limitations:**

The limitations are not explicitly discussed. There are points to discuss such as the cases where Eq. 6 is not hold.

**Strengths And Weaknesses:**

Strengths
- Robust anomaly detection is an important topic in both academia and industry.
- The proposed approach, which eliminates the influence of noise during training for pure data's density by penalizing the Jacobian of the normalized flow, is interesting.
- The experiments were conducted using a wide variety of baselines and datasets.

Weaknesses
 - My primary concern is the experimental setup. Reading Appendix D, there is a description "we use grid search to determine the best hyperparameter configuration on each different dataset". Does this refer to reporting the best results on test data for the proposed method? If this is true, this experimental setup is problematic due to the leakage. Typically, hyperparameters such as $\lambda$ should be tuned on data different from the test data. Note that using anomalous data for tuning is undesirable in this paper's setting, which assumes only (noisy) normal data for training.
- The proposed method relies heavily on the assumption of Eq. 6. As described in the last paragraph of Section 3.1, this assumption seems reasonable to some extent. However, I am uncertain about the scope of tasks or domains to which it holds.
- Some mathematical descriptions are unclear. For example, in Eq. 16, $| \cdot |$ in the regularization term does not appear to be clearly defined.

---

> ### Author Rebuttal · Authors · 2026-03-31
>
> Response to Reviewer PoAr:
> Dear Reviewer PoAr, we are very grateful for your valuable feedback and suggestions. Our responses are as follows:
>
> **Weaknesses:**
>
> 1. We thank the reviewer for such an insightful question. Hyperparameter tuning for methods in unsupervised domains like AD and clustering remains a profound open challenge [1]. And such hyperparameter tuning strategy is commonly used  in prior work [2,3]. Under such settings, we use the same hyperparameter-tuning strategy for all baseline methods to prevent any unfair advantage. However, we completely agree with the reviewer that demonstrating practical utility without label leakage is essential. To address this, we integrated AutoUAD [1]—a SOTA hyperparameter tuning framework designed solely for unlabeled data—to tune our method. As shown in the table below, NRDE* (tuned via AutoUAD) achieves performance remarkably close to the original results. This empirically proves our method's practical utility in true unsupervised scenarios.
>
> Data |NRDE|NRDE*|
> |:-|:-|:-|
> Cardiotocography|87.1|86.3|
> SpamBase|84.5|82.7|
> Pima|80.0|79.5|
> Satellite|85.8|84.9|
> WPBC|77.6|74.9|
>
>
>
> [1] Dai and Fan. AutoUAD: Hyper-parameter Optimization for Unsupervised Anomaly Detection. ICLR 2025.
>
> [2] Yin, Jiaxin, et al. Mcm: Masked cell modeling for anomaly detection in tabular data. ICLR 2024.
>
> [3] Xu, Hongzuo, et al. Fascinating supervisory signals and where to find them: Deep anomaly detection with scale learning. ICML 2023
>
> 2. We thank the reviewer for this question. We agree that the assumption  may not hold universally. However, we have proactively investigated the boundaries of this assumption through extensive experiments in **Appendix H**.
> (i) in **Appendix H.4**, we provide an empirical method to verify the variance difference on real datasets. Table 12 demonstrates a strong correlation between performance gains and larger variance differences, thus the performance gains in most datasets shows that our assumption practically holds across the majority of benchmark datasets. (ii) To find out scenarios where it fails, **Appendix H.1** presents synthetic experiments explicitly violating this assumption; as expected, NRDE's performance declines as the variance difference decreases, mapping out the method's boundaries. (iii)Furthermore, in **Appendix H.3**, we designed an ablation model (NRDE-CON) based on the opposite assumption. The consistent superiority of NRDE over NRDE-CON on real datasets further corroborates our design. A discussion related to these verification and limitations is now provided in the revised manuscript.
>
> 3. We thank the reviewer for catching this notational ambiguity. We respectfully clarify that $| \cdot |$ refers to taking the element-wise absolute value of the matrix. We have updated the revised paper to include a clear explanation below Eq. 16.
>
> **Key Questions For Authors**
> 1. We thank the reviewer for this question. We respectfully clarify that this is not related to our assumption. As mentioned in our paper, data of other types like image or text can be converted into  semantic tabular-type representations using powerful pre-trained foundational models (e.g., CLIP, ViT, and LLMs).  Once converted, the downstream AD task is essentially reduced to finding anomalies in a multi-dimensional tabular dataset.  For this reason, we focus on tabular data.
>
> 2. We thank the reviewer for this insightful question. We acknowledge that the computational cost increases for high-dimensional data. As shown in Table 25 of Appendix I.7, the training time becomes relatively high primarily due to the Jacobian matrix computation required for training samples.
> While standard normalizing flows can scale efficiently, our specific noise-separation formulation introduces this training overhead. Fortunately, for very high-dimensional cases, this computational challenge can be straightforwardly mitigated. By employing dimensionality reduction methods such as Kernel PCA prior to NRDE, we significantly reduce the Jacobian computation burden.
> As shown in the following table, NRDE strongly maintains its effectiveness on the low-dimensional data processed via Kernel PCA. This empirical evidence proves that our framework can be practically adapted for high-dimensional datasets without sacrificing its core performance.
>
> Data (Dimension)|Original|Low-dimensional (128)|
> |:-|:-|:-|
> Speech (400)|58.7|56.5|
> InternetAds (1555)|83.9|80.0|
>
> **Limitations**
>
> We thank the reviwer for pointing out such oversight. In **Appendix H**, we have proactively investigated the boundaries of this assumption through extensive experiments, including the scenarios where assumptions do not hold. And we have included a detailed discussion of the limitations related to  assumption in our revised paper.
>
> **We are looking forward to your feedback, and please do not hesitate to let us know if there are any concerns or questions still not properly addressed.**

---

> > ### Author Rebuttal · Reviewer_PoAr · 2026-04-01
> >
> > Thank you for your response.
> > The hyperparameter selection using AutoUAD is appropriated, but results on only a subset of the data are insufficient.
> > In addition, if a large part of the results on the submitted paper are based on tuning on the test data, substantial revision will be necessary.

---

> > > ### Author Response · Authors · 2026-04-02
> > >
> > > Dear Reviweer PoAr,
> > > We are grateful for your further engagement. Thanks again for your time and effort throughout the review process. Our responses are as follows:
> > >
> > > ## **More results using AutoUAD for hyperparameter tuning**
> > >
> > > We thank the reviewer for recognizing the appropriateness of our hyperparameter selection strategy using AutoUAD.  Given the time and space constraints of the rebuttal period, we have currently expanded our evaluation using AutoUAD for hyperparameter tuning to 20 datasets. As shown in the table below, NRDE (AutoUAD) achieves performance remarkably close to the original results. This empirically demonstrates our method's practical utility and robustness in truly unsupervised scenarios. We fully agree that comprehensive evaluation is crucial, and we will include the complete tuning results on all 47 datasets in the revised manuscript.
> > > | **AUROC** | NRDE | NRDE (AutoUAD) |
> > > |---------|------|----------|
> > > | annthyroid | 97.3 | 95.9 |
> > > | fault | 71.4 | 69.1 |
> > > | glass | 89.2 | 87.4 |
> > > | Ionosphere | 95.3 | 95.4 |
> > > | magic.gamma | 82.5 | 80.5 |
> > > | mnist | 93.2 | 92.3 |
> > > | musk | 100.0 | 100.0 |
> > > | optdigits | 97.7 | 95.6 |
> > > | PageBlocks | 93.5 | 93.4 |
> > > | pendigits | 99.3 | 99.2 |
> > > | satimage-2 | 99.8 | 99.6 |
> > > | shuttle | 99.9 | 99.8 |
> > > | thyroid | 99.1 | 99.3 |
> > > | vertebral | 82.8 | 80.8 |
> > > | vowels | 88.7 | 92.0 |
> > > | Waveform | 86.7 | 85.4 |
> > > | WBC | 99.7 | 99.4 |
> > > | WDBC | 99.7 | 99.8 |
> > > | Wilt | 83.6 | 82.2 |
> > > | wine | 99.8 | 98.5 |
> > >
> > > | **AUPRC** | NRDE | NRDE (AutoUAD) |
> > > |---------|------|----------|
> > > | annthyroid | 75.6 | 74.5 |
> > > | fault | 72.9 | 72.7 |
> > > | glass | 30.8 | 39.9 |
> > > | Ionosphere | 95.2 | 95.7 |
> > > | magic.gamma | 85.0 | 84.1 |
> > > | mnist | 68.5 | 66.6 |
> > > | musk | 100.0 | 100.0 |
> > > | optdigits | 65.8 | 55.4 |
> > > | PageBlocks | 76.6 | 78.8 |
> > > | pendigits | 84.8 | 79.2 |
> > > | satimage-2 | 94.8 | 95.7 |
> > > | shuttle | 98.5 | 98.5 |
> > > | thyroid | 81.7 | 84.5 |
> > > | vertebral | 51.4 | 50.0 |
> > > | vowels | 48.5 | 63.1 |
> > > | Waveform | 32.3 | 27.4 |
> > > | WBC | 95.6 | 94.3 |
> > > | WDBC | 96.3 | 98.5 |
> > > | Wilt | 30.6 | 37.4 |
> > > | wine | 99.1 | 91.4 |
> > >
> > > ## **Performances of NRDE and baselines on 47 datasets under default hyperparameter setting**
> > >
> > > We thank the reviewer for raising this critical point. We fully agree that evaluation under more realistic, dataset-agnostic hyperparameter settings is essential to provide a comprehensive and fair comparison between methods. To address your concern, we have now evaluated the performance of all methods across all 47 datasets under their **default hyperparameter** settings. For NRDE, we strictly used the default configuration ($lr=0.001,\lambda=0.1$). Similarly, each baseline method was evaluated using the exact recommended hyperparameter configuration provided in its original paper or official codebase, without dataset-specific tuning. Detailed results for all 47 datasets are provided in **Table 1** and **Table 2** via our anonymous link (https://anonymous.4open.science/r/NRDE-7AEB/README.md), and the average performance is summarized in the table below. Under this strictly untuned and realistic setting, NRDE continues to achieve the best average performance, outperforming the second-best deep-learning-based method by more than 2%. We have incorporated these comprehensive default-setting experimental results into our revised manuscript to ensure our evaluation is both rigorous and practical.
> > >
> > > | Methods | KDE | KNN | LOF | OC-SVM | IF | AE | DSVDD | RealNVP | NeutralAD | ECOD | ICL | SLAD | DPAD | MCM | DTE-C | TCCM | ADERH | NRDE |
> > > |---|---|---|---|---|---|---|---|---|---|---|---|---|---|---|---|---|---|---|
> > > | AVG AUROC| 84.3 | 85.1 | 81.0 | 79.8 | 78.2 | 81.3 | 74.3 | 83.6 | 77.9 | 73.7 | 76.1 | 79.2 | 81.5 | 83.4 | 83.3 | 80.6 | 80.5 | **85.9** |
> > > | AVG AUPRC| 62.3 | 62.2 | 57.6 | 54.6 | 48.9 | 58.2 | 51.5 | 53.6 | 48.0 | 45.0 | 49.9 | 58.4 | 58.2 | 59.9 | 57.0 | 59.9 | 54.6 | **64.4** |
> > >
> > >
> > > **We sincerely look forward to your feedback and hope our response addresses your remaining concerns.**
> > >
> > > ## **Update:**
> > >  We have now provided the experimental results for the SOTA baseline methods using AutoUAD for hyperparameter tuning. These results are detailed in **Table 3 and Table 4** via our anonymous link (https://anonymous.4open.science/r/NRDE-7AEB/README.md).
> > >
> > > The findings demonstrate that NRDE consistently outperforms the other baselines under realistic hyperparameter tuning (AutoUAD). Combined with our evaluations using the best-performing and default recommended hyperparameters, it is clear that NRDE maintains a significant advantage across all three tuning settings. Given the time constraints of the rebuttal period, we commit to including the comprehensive results for all baselines across all 47 datasets in the revised manuscript.

---

### Official Review · Reviewer_sAoz · 2026-03-10

**Soundness:** 3
**Presentation:** 2
**Significance:** 3
**Originality:** 2
**Overall Recommendation:** 4
**Confidence:** 4

**Summary:**

This paper proposed an anomaly detection method, NRDE, for noisy data, and considers both normal data and noisy normal data as the target class. The method considers that the data is mixed from a noise-free source and a noise source. NRDE eliminates the noise source to obtain the pure data representation through Jacobian-regularized normalizing flow. The experimental results show the effectiveness of the method.

**Compliance With Llm Reviewing Policy:**

Affirmed.

**Final Justification:**

The refinement of the theoretical guarantee and analysis of experimental results addresses my concerns.

**Key Questions For Authors:**

- Figure 1 should be explained more clearly. Is that Anomaly Score mean how anomalous the data point is? Is NRDE better than other methods, because it assigns a similar score to normal data and noisy normal data, but a higher score for anomalous data? If so, the author should explain this in the caption.

- The original data $x$ is generated from some source $s$ through a transformation $G(\cdot)$, the dimensions $d$ of $x$ and $s$ are the same (lines 185-186), and the dimension of the pure data source is $m$. Does this assume that the pure data can be represented by fewer dimensions than the original data, and the remaining dimensions are noise? If so, why is the noisy dimension $d-m$? Will it be possible that the noisy dimension be fewer or more?

- In Section 3.2, lines 211-213, why is that when the density of pure data on original distribution $p_ \mathcal{X}(x_ {pure})$ or the density of pure data on pure distribution $p_ {\bar{\mathcal{X}}}(x_ {pure})$ is smaller, the data is more likely to be anomalous? What is the rationale? And what is the small or large density $p$ compared with?

- Does Assumption 3.2 already assume the method can detect anomalies if the $\varphi$ is small? If the $\phi$ is very large, this assumption is meaningless.

- Table 14 shows that NRDE achieves the best or second-best AUROC in 14 out of 47 datasets, and NRDE-M achieves this in 13 out of 47 datasets. Why do these results support the effectiveness of NRDE?

**Limitations:**

The limitation of this paper is not discussed. The potential limitation can be the assumptions in the method and theorems, such as the same dimension of the $x$ and $s$, and Assumption 3.2.

**Strengths And Weaknesses:**

Strengths:
- The method addresses the issue of anomaly detection with noisy data. Unlike traditional anomaly detection methods, this paper considers a practical scenario, where the noisy normal data should also be considered as the target class, since there will also be noisy data in testing procedures.

- This paper proposes a novel method for identifying the noise-free source and a noise source using Jacobian-regularized normalizing flow.

Weaknesses:
- The "tabular data" is only mentioned in left hand side of lines 102-104 in the Introduction. The importance or challenge of tabular data is not explained.

- The following citation is listed twice. The reference list should be checked to ensure it is formatted correctly.

  > Ruff, L., Vandermeulen, R., Goernitz, N., Deecke, L., Siddiqui, S. A., Binder, A., Müller, E., and Kloft, M. Deep one-class classification. In International conference on machine learning, pp. 4393–4402. PMLR, 2018b.

- The writing should be checked carefully. For example, there are two "as" in the right-hand side of page 2, line 59: "be described as as being generated".

- According to the proof of Theorem 3.3, the term $\varsigma_2$ in Theorem 3.3 should be the density of the point $c$. Or the conclusion should be $||x^b - x^a|| > \cdots$.

- In the main text, some parts use "AD" and others use "anomaly detection." Consistent terminology should be used.

- The main challenge addressed by this paper is anomaly detection on noisy data. The experimental evaluation should focus on Section 4.4, including all 47 datasets and various levels of noise injection.

---

> ### Author Rebuttal · Authors · 2026-03-31
>
> Dear Reviewer sAoz, we are very grateful for your valuable feedback and suggestions. Our responses are as follows:
>
> **Weaknesses:**
>
> 1.&2.&3. We thank the reviewer for their careful reading and constructive feedback. We have updated our revised paper to include a detailed discussion of challenges on tabular data, including heterogeneity  of structure and vulnerability to inherent noise. Besides, we have removed the duplicated citation and duplicated 'as'.
>
> 4. We thank the reviewer for detailed check of our theoritical analysis. We have carefully revisited the proof of **Theorem 3.3**. We respectfully clarify that there might be a notational misunderstanding regarding $\varsigma_2$ and the points involved. The core distance condition is actually constructed around $x^c$ and $x^a$. Specifically when $||x^c-x^a\||>\frac{(\varsigma_1-\varsigma_2)}{\varphi}$, we will have $p(x^c_{pure})< \min_{x \in \mathcal{D}}p(x_{pure})$. Thus, it can be detected as anomaly.
>
> 5. We appreciate the reviewer's attention to detailed writing. We now use 'AD' throughout the text following its initial definition.
>
> 6. We respectfully clarify that our core motivation is to tackle inherent noise naturally present in real-world data. Such intrinsic noise exists in the original data without any artificial noise injection and is fundamentally different from artificial noise.
> For this reason, our primary evaluation rightly focuses on standard AD setting across all 47 original datasets. The experiments with noise injection in Section 4.4 are secondary and designed to evaluate robustness under artificial perturbations. We appreciate the reviewer's comment and have clarified this distinction in the revised manuscript.
>
> **Key Questions For Authors:**
> 1. We thank the reviewer for the constructive feedback. Your interpretation is correct. Following your constructive suggestion, we have updated the caption in our revised paper.
> 2. We thank the reviewer for this insightful question. Your understanding of pure data generation is correct. Because $G(⋅)$ preserves the total dimension $d$ of the original data, and the pure data is generated from an intrinsic manifold of dimension $m$, the remaining latent source dimensions must strictly equal $d−m$. By definition, we attribute these remaining $d−m$ sources to noise generation. Regarding whether the noise dimension could be fewer or more: by mathematical definition it is strictly $d-m$ and is highly dataset-dependent. Practically, we dynamically approximate $m$ as $\hat{m}$. Consequently, the noise dimension $d-\hat{m}$ could be fewer or more than the actual $d-m$. We show the effectiveness of this selection strategy in **Appendix I.6**.
> 3. The rationale is rooted in the probabilistic definition of anomalies. Since $\mathcal{X}$ represents the distribution of normal data, true anomalies naturally fall into low-probability regions of this distribution. In our specific framework, our goal is to detect anomalies caused by fundamental changes in the pure data sources. Therefore, a significantly lower $p_ \mathcal{X}(x_ {pure})$ strongly indicates that the pure underlying structure of the data deviates from the normal manifold, making it likely to be anomalous. Regarding the comparison, these densities are used to comapre and rank all the test samples. This relative ranking yields performance metrics such as AUROC.
> 4. We respectfully clarify that Assumption 3.2 does not trivially assume all anomalies can be detected; rather, it specifies a lower bound on the density variation rate to establish  which anomalies are theoretically detectable. As mentioned in our paper (line 298), $\varphi$ could be calculated as $\inf_{x \in \text{dom}(p_{\mathcal{X}})} \|\nabla_x p_{\mathcal{X}}(\hat{x}_{pure})\|$ and is usually small. However, when it is large, it is still meaningful that as mentioned in Theorem 3.3, more anomalies closer to normal data could also be detectable.
> 5. We thank the reviewer for this insightful question. As dictated by the No Free Lunch theorem, no single algorithm can achieve SOTA performance across all heterogeneous data distributions. Therefore, given the extreme diversity of the 47 datasets in ADBench, absolute dominance on all 47 datasets is unattainable for any method. But compared with other 17 baselines, NRDE achieves top-2 AUROC on more datasets. Furthermore, when evaluating the overall performance, NRDE consistently achieves the highest Average AUROC and the best Average Rank across all methods. This comprehensive aggregate superiority validates the effectiveness of NRDE.
>
> **Limitations**
> We thank the reviewer for pointing out this oversight. We conduct experiments in **Appendix H** to evaluate NRDE when the assumption does not hold. And we have included a detailed discussion of the limitations in our revised paper.
>
> **We are looking forward to your feedback, and please do not hesitate to let us know if there are any concerns or questions still not properly addressed.**

---

> > ### Author Rebuttal · Reviewer_sAoz · 2026-04-01
> >
> > Thank you for your response. I have follow-up questions for the authors.
> >
> > - Regarding the challenges with tabular data, which part of the proposed method is developed to address the heterogeneity of structure and vulnerability to inherent noise?
> >
> > - Regarding the statement of Theorem 3.3, is that correct if we replace all $x^b$ with $x^c$?
> >
> > - Regarding the $p _\mathcal{X}(x _{pure})$ or $p _{\bar{\mathcal{X}}} (x _{pure})$, from Algorithm 1, is it correct that the smaller the anomaly score is, the larger the probability that the data is an anomaly? If so, this contradicts the description in Figure 1, where the anomaly score is higher for anomalous data than for normal data.
> >
> > - Regarding Assumption 3.2, if the $\varphi$ is defined on $p _\mathcal{X}(x _{pure})$, the assumption is reasonable. However, this assumption is for $p _\mathcal{X}(\hat{x} _{pure})$, which is learned by the Jacobian-Regularized Normalizing Flow in Section 3.3. In this case, does it already assume the trained model can detect anomalies?
> >
> > - Regarding the effectiveness of NRDE shown in Table 14, what statistical and intrinsic properties of datasets is NRDE suited for?
> >
> > ---
> >
> > **Update**:
> >
> > Thank you for the response. I have updated the score.

---

> > > ### Author Response · Authors · 2026-04-02
> > >
> > > We are grateful for your further engagement. Thanks again for your time and effort throughout the review process.
> > > 1. As mentioned in the introduction, density-based methods like normalizing flow make no assumptions about shape or distribution of the data and are capable of modeling complex data structures. For this reason, NRDE, which utilizes normalizing flows as the backbone, can address the heterogeneity of structure. As for vulnerability to inherent noise, as mentioned in EQ.(17), NRDE generates pure data by isolating the inherent noise sources and estimates the density of pure data to obtain the anomaly score. For this reason, it is robust to inherent noise.
> > > 2. We thank the reviewer for this insightful question. We respectfully clarify that if we replace all $x^b$ with $x^c$, we would obtain the following inequality: $||x^c-x^a||>\frac{\varsigma_1-\varsigma_2}{\varphi}=\frac{p_{\mathcal{X}}(\hat{{x}}^a_{{pure}})-p_{\mathcal{X}}(\hat{{x}}^c_{{pure}})}{\varphi}$. However, according to Assumption 3.2, the following must hold: $\vert p_{\mathcal{X}}(\hat{{x}}^a_{{pure}})-p_{\mathcal{X}}(\hat{{x}}^c_{{pure}}) \vert= p_{\mathcal{X}}(\hat{{x}}^a_{{pure}})-p_{\mathcal{X}}(\hat{{x}}^c_{{pure}}) \geq \varphi||{x}^a-{x}^b||$. Because these two mathematical statements directly contradict each other, the theorem would no longer hold if this replacement were made. Furthermore, the reviewer's careful examination prompted us to double-check the statement, and we did identify a typo in the original theorem regarding the definitions of $x^a$ and $x^b$. The rigorous definitions should be:
> > >  $\arg \max\_{{x} \in \mathcal{D}} p\_{\mathcal{X}}(\hat{{x}}\_{{pure}})={x}^a$ and $\arg\min_{{x}\in \mathcal{D}} p_{\mathcal{X}}(\hat{{x}}_{{pure}})={x}^b$. We have corrected this typo in our revised paper.
> > > 3. We thank the reviewer for this careful observation. Your understanding of Algorithm 1 is correct. Regarding Figure 1, we sincerely thank you for pointing out our oversight. The score plotted in Figure 1 is actually the negative log-likelihood $-\log p_{\mathcal{X}}(\hat{{x}}\_{{pure}})$ instead of $p_{\mathcal{X}}(\hat{{x}}_{\text{pure }})$. We applied this transformation so that all methods in Figure 1 share a consistent convention: a larger anomaly score (strictly >0) corresponds to a higher likelihood of being an anomaly, allowing for a intuitive visual comparison. We have updated the caption  in the revised manuscript to clearly provide this explanation and prevent any further confusion. Regarding Algorithm 1, we have updated the anomaly score to the negative log-likelihood to maintain consistency.
> > > 4. We respectfully clarify that Assumption 3.2 does not presuppose the anomaly detection capability of the trained model. Instead, it specifies a lower bound on the density variation rate for the learned density function, $p_{\mathcal{X}}(\hat{{x}}_{\text{pure}})$, which is the property of probabilistic density. This variation rate is strictly a mathematical/geometric property of the modeled probability density surface (which can be influenced and bounded by network architectures), rather than a direct measure of downstream anomaly detection performance. Our analysis demonstrates that if the learned density function satisfies this fundamental variation property (Assumption 3.2), it consequently possesses the capability to distinguish specific anomalies. This causal relationship is exactly what we formalize and prove in Theorem 3.3.
> > > 5. We thank the reviewer for this insightful question. We have specifically investigated this aspect in **Appendix H.4** and **Appendix H.5**. In short, NRDE is particularly well-suited for datasets that exhibit a large variance difference between the data sources and the noise sources. We demonstrate this through two key analyses:
> > > (i) **Empirical Correlation (Appendix H.4)**: We introduce an empirical method to verify the variance differences on real datasets. As shown in Table 12, there is a strong positive correlation between NRDE's performance gains and the magnitude of the variance differences.
> > > (ii)  **Enhanced Separability (Appendix H.5)**: We conducted experiments applying baseline methods to $\hat{x}\_{pure}$ generated by NRDE. The results show that the separability between normal and anomalous samples is substantially enhanced on the pure data space. Crucially, baseline methods exhibit larger performance improvements on datasets with more significant variance differences, indicating that NRDE's noise-isolation capability is most effective in these scenarios. We attribute this effectiveness to the fact that a more significant variance gap between the data sources and the noise sources allows for a much more accurate estimation of the parameter $\hat{m}$. Consequently, the generated pure data $\hat{x}\_{pure}$ is substantially closer to the true pure data $x\_{pure}$, leading to the superior performance observed in Table 14.
> > >
> > > **Update:**
> > >
> > > We sincerely appreciate your feedback and recognition!

---

### Official Review · Reviewer_BGe8 · 2026-03-12

**Soundness:** 3
**Presentation:** 3
**Significance:** 2
**Originality:** 3
**Overall Recommendation:** 4
**Confidence:** 5

**Summary:**

NRDE is a density-based anomaly detection method for tabular data that isolates inherent noise by separating latent sources into data sources and noise sources via a Jacobian-regularized normalizing flow.

**Compliance With Llm Reviewing Policy:**

Affirmed.

**Final Justification:**

All my comments have been addressed, I have raised my rating.

**Key Questions For Authors:**

1. In Figure 1, the scores of DSVDD appear to show two well-separated distribution, yet its AUROC is more than 20 points lower than NRDE. Can the authors clarify whether this is because noisy normal data and anomalous data are scored similarly by DSVDD, and if so, does this reflect a property of the synthetic data construction rather than a genuine failure of DSVDD?
2. The noisy data experiment injects isotropic Gaussian noise with small variance, which directly satisfies the paper's assumption that noise sources have lower variance than data sources, have the authors evaluated NRDE under noise distributions that violate this assumption, such as structured or high-variance noise, to test whether the robustness claim generalizes?
3. I am confused abouth the assumption in Eq6 which requires a significant variance gap between data sources and noise sources controlled by an unknown constant, which can neither be verified nor guaranteed on real-world datasets.
4. The training time for NRDE is slower than RealNVP according to Table 25, can the authors provide a more detailed analysis of when the computational overhead is and is not acceptable relative to the performance gain?

**Limitations:**

Yes

**Strengths And Weaknesses:**

Strengths:
1. The motivation of isolating noisy normal data from anomalous data is practically meaningful.
2. The Jacobian regularization strategy for separating data sources and noise sources is a clean and self-contained design.
3. The experimental scope is broad, covering 47 datasets and 17 baselines across different evaluation settings.

Weaknesses:
1. The absolute performance gain in the standard anomaly detection setting is modest and comes with substantial training time overhead, limiting the practical impact of the method.
2. The core assumption that data sources have larger variance than noise sources is neither theoretically justified nor empirically verified on real datasets, undermining the reliability of the method's foundations.

---

> ### Author Rebuttal · Authors · 2026-03-31
>
> Dear Reviewer BGe8, we are very grateful for your valuable feedback and suggestions. Our responses are as follows:
>
> **Weaknesses:**
> 1. We respectfully disagree with your statement. As shwon in **Table 1** and **Table 25**, in most cases, NRDE’s training time remains comparable to RealNVP with no substantial increase, but the average AUPRC performance improvement over $15$% can be observed.
> 2. We thank the reviewer for such an insightful question. Although such assumption verification on real datasets can be challenging, we still provide sveral experimental reuslts to verify assumptions and motivations in real datasets in **Appendix H**. For instance, **Appendix H.3** provides performance of NRDE under opposite assumptions, where performance decline supports the assumption in our paper. **Appendix H.4**  provides an empirical method to verify the variance difference on real datasets. Table 12 demonstrates a strong correlation between performance gains and larger variance differences, thus the performance gains in most datasets shows that our assumption practically holds across the majority of benchmark datasets. In **Appendix H.5**, we conduct experiments using baselines on clean data from NRDE where  separability between normal and anomalous samples is substantially enhanced, and the performance improvement compared to original data verifies our assumption and motivation.
>
> **Key Questions For Authors**
> 1. We thank the reviewer for this insightful question. The reviewer's interpretation is exactly correct.  However, we use this synthetic example to show that methods like DSVDD could distingush noisy normal data as anomalies. We would like to clarify that this is a genuine property (and limitation) of methods like DSVDD, rather than a mere artifact of our synthetic construction. Methods like DSVDD is not designed to handle such inherent noise.  We utilized this synthetic toy example specifically to isolate and visualize this exact failure mode. The synthetic data serves as a controlled environment to demonstrate why conventional methods fail under inherent noise. This motivation is strongly corroborated by our extensive empirical results on real datasets, where NRDE consistently achieves significantly lower FPR and higher AUROC precisely because it isolates such noise.
>
> 2. We thank the reviewer for this insightful question. First, we would like to briefly clarify that NRDE is designed to handle inherent noise naturally present in the data, rather than artificially injected perturbations. Perturbations are in the feature space and our assumption is on the source space, they are not related. However, we fully agree testing under assumption-violating conditions is crucial for evaluating robustness. We have explored this extensively in **Appendix H.1**.   Regarding the variance of the injected noise, since features are z-score normalized to unit variance, injecting large-variance noise would fundamentally obscure the separation between normal and anomalous samples. However, to further strengthen our empirical validation, we  conduct new experiments injecting large standard Gaussian noise ($\mathcal{N}(0,I_d)$) into datasets.  As shown in the table below, NRDE maintains an overall performance advantage.
>
> | Data | AE | KNN |TCCM|MCM | DTE | NRDE |
> | :--- | :--- | :--- | :--- | :--- | :--- | :--- |
> | breastw | 98.7 | **99.0** | 98.8 | **99.0** | 98.3 | 98.1 |
> | Cardiotocography | 69.4 | 71.4 | 76.5 | 76.6 | 73.2 | **76.8** |
> | Ionosphere | **95.3** | 93.5 | 91.6 | 92.6 | 91.0 | 92.3 |
> | landsat|51.7 | 56.2 | **58.9** | 53.3 | 51.9 | 58.1 |
> | magic.gamma | **69.7** | 68.8 | 68.1 | 67.1 | 69.5 | **69.7** |
> | Pima |66.4| **69.1** | 68.6 | 66.3 | 67.5 | 66.8 |
> | satellite | 75.7 | **78.6** | 77.9 | 75.5 | 73.3 | 75.7 |
> | SpamBase | 80.6 | 81.5 | 78.5 | 80.0 | **81.6** | 74.3 |
> | WPBC | 55.0 | 49.5 | 58.3 | 53.5 | 53.2 | **68.2** |
> | yeast | 56.1 | 49.3 | 49.8 | 49.6 | 50.8 | **56.6** |
> | **AVG** | 71.8 | 71.7 | 72.7 | 71.4 | 71.0 | **73.7** |
>
>
> 3. Please refer to our response for **Weakness 2**.
>
> 4. We thank the reviewer for this constructive question. The main computational bottleneck of NRDE compared to RealNVP is the Jacobian matrix computation for each training sample, and this time-cost gap increases with data dimensionality. But NRDE's inference time remains virtually identical to standard RealNVP. The overhead may become prohibitive on high-dimensional data. However, for such high-dimensional cases, the computational cost can be easily mitigated by employing dimensionality reduction methods like PCA prior to conducting NRDE. The following table demonstrates that NRDE remains effective even on low-dimensional (128) data processed via Kernel PCA.
>
> Data (Dimension)|Original|Low-dimensional (128)|
> |:-|:-|:-|
> Speech (400)|58.7|56.5|
> InternetAds (1555)|83.9|80.0|
>
> **We are looking forward to your feedback, and please do not hesitate to let us know if there are any concerns or questions still not properly addressed.**

---

> > ### Author Rebuttal · Reviewer_BGe8 · 2026-04-01
> >
> > Thanks for the rebuttal, I have updated my score.

---

> > > ### Author Response · Authors · 2026-04-01
> > >
> > > Dear Reviewer  BGe8,
> > >
> > > We are sincerely grateful for your valuable feedback and for your recognition of our work. Thanks again for your time and effort throughout the review process.

---

### Official Review · Reviewer_97ko · 2026-03-12

**Soundness:** 2
**Presentation:** 1
**Significance:** 2
**Originality:** 2
**Overall Recommendation:** 3
**Confidence:** 4

**Summary:**

The paper proposes NRDE, a noise-robust density estimation approach for tabular anomaly detection built on normalizing flows with a novel Jacobian row-sparsity regularization. The key idea is to align the learned flow so that latent components separate into high-variance data sources and low-variance noise sources, then zero out the latter to obtain a pure sample and score anomalies via its density.
The authors justify the approach with analysis linking Jacobian row norms to source variances, provide error bounds for the density approximation, and demonstrate empirical performance on 47 datasets and several robustness settings.

**Compliance With Llm Reviewing Policy:**

Affirmed.

**Final Justification:**

The paper demonstrates solid empirical performance but is limited by modest novelty and a strong core assumption. While the rebuttal partially addresses these concerns with additional evidence, it does not fully resolve them, and I have therefore slightly increased my score while maintaining a cautious overall assessment.

**Key Questions For Authors:**

pls see the above weaknesses part

**Limitations:**

The paper does not clearly discuss the limitations or potential societal impacts of the proposed method. It would be helpful for the authors to discuss the assumptions underlying the approach, particularly the variance structure between signal and noise sources, and how the method may behave when these assumptions do not hold. In addition, since the experiments rely on synthetic noise injection, the authors could comment on how the method might perform under more realistic noise conditions. The authors could also comment on potential failure cases where meaningful variations might be incorrectly treated as noise.

**Strengths And Weaknesses:**

Strengths
1. Evaluates on a large, modern tabular benchmark with 17 baselines, including shallow and deep AD models and a normalizing-flow baseline (RealNVP).
2. Provides evidence via a synthetic example that the approach reduces false positives from inherent noise variations.

Weaknesses
1. The overall framework mainly combines existing components such as normalizing flows and Jacobian regularization, and the methodological novelty appears somewhat limited.
2. The core assumption that data sources have substantially larger variances than noise sources (in source space before mixing) is strong and problem-dependent. Many real tabular settings can exhibit high-variance nuisance/noise or correlated noise-signal structure.
3. The experiments rely on injecting synthetic noise into real tabular datasets to simulate noisy observations. Since the proposed method assumes that noise sources have significantly smaller variance than signal sources, it would be helpful to clarify how the injected noise distributions relate to this assumption. In particular, the noise generation process appears to follow the same variance structure assumed by the method, which may favor the proposed approach and incur unfair issues. Additional experiments with more realistic or heterogeneous noise distributions would strengthen the empirical validation.
4. The presentation and organization of the paper could be improved. For example, the introduction includes a relatively long paragraph reviewing general anomaly detection methods, but the connection between these methods and the proposed approach is not clearly articulated, which makes the motivation less focused.

---

> ### Author Rebuttal · Authors · 2026-03-31
>
> Dear Reviewer 97ko, we are very grateful for your valuable feedback and suggestions. Our responses are as follows:
>
> **Weaknesses:**
> 1. We thank the reviewer for the opportunity to highlight our methodological contributions. We respectfully clarify that our primary novelty is a new perspective on data generation: explicitly separating inherent noise sources from data to estimate the pure data density. Existing flows are fundamentally unable to perform such separation. Therefore, combining normaglizing flows with Jacobian regularization in our framework is not a simple concatenation, but a deliberately designed mechanism to achieve this noise-isolated density estimation.
> 2. We thank the reviewer for this insightful comment. We agree that the assumption  may not hold universally. However, we have proactively investigated the boundaries of this assumption through extensive experiments in **Appendix H**.
> (i) in **Appendix H.4**, we provide an empirical method to verify the variance difference on real datasets. Table 12 demonstrates a strong correlation between performance gains and larger variance differences, thus the performance gains in most datasets shows that our assumption practically holds across the majority of benchmark datasets. (ii) To find out scenarios where it fails, **Appendix H.1** presents synthetic experiments explicitly violating this assumption; as expected, NRDE's performance declines as the variance difference decreases, mapping out the method's boundaries. (iii)Furthermore, in **Appendix H.3**, we designed an ablation model (NRDE-CON) based on the opposite assumption. The consistent superiority of NRDE over NRDE-CON on real datasets further corroborates our design. A discussion related to these verification and limitations is now provided in the revised manuscript.
>
> 3. We thank the reviewer for this question. We would like to clarify a potential misunderstanding: our primary evaluation on the 47 benchmark datasets does not involve any synthetic noise injection. Our core focus is on handling the inherent noise naturally present in real data. Since the artifical noise is injected in the feature space and our assumption is on the source space, they are not related. The strong performance across these 47 original datasets demonstrates that our method does not rely on artificially constructed variance structures to gain an unfair advantage. The noise injection experiments are purely supplementary robustness checks. Regarding the scale of the injected noise, since features are z-score normalized to unit variance, injecting large-variance noise would fundamentally obscure the separation between normal and anomalous samples. However, to comprehensively address your concern, we have evaluated NRDE under larger standard Gaussian noise ($\mathcal{N}(0,I_d)$) injections. As shown in the table below, NRDE  maintains a overall performance advantage.
>
> |Data|AE|KNN|TCCM|MCM|DTE|NRDE|
> |:-|:-|:-|:-|:-|:-|:-|
> |breastw|98.7|**99.0**|98.8|**99.0**|98.3|98.1|
> |Cardiotocography|69.4|71.4|76.5|76.6|73.2|**76.8**|
> |Ionosphere|**95.3**|93.5|91.6|92.6|91.0|92.3|
> |landsat|51.7|56.2|**58.9**|53.3|51.9|58.1|
> |magic.gamma|**69.7**|68.8|68.1|67.1|69.5|**69.7**|
> |Pima|66.4|**69.1**|68.6|66.3|67.5|66.8|
> |satellite|75.7|**78.6**|77.9|75.5|73.3|75.7|
> |SpamBase|80.6|81.5|78.5|80.0|**81.6**|74.3|
> |WPBC|55.0|49.5|58.3|53.5|53.2|**68.2**|
> |yeast|56.1|49.3|49.8|49.6|50.8|**56.6**|
> |**AVG**|71.8|71.7|72.7|71.4|71.0|**73.7**|
>
> 4. We sincerely thank the reviewer for the constructive suggestion on writing. We agree that the motivation was less focused due to the lengthy review of general methods. In the revised paper, we contrast general AD methods (which often require rigid structural assumptions) with density-based methods (which are highly flexible and make no such assumptions) in one short paragraph. We then clearly point out the critical problem in conventional density-based methods: their sensitivity to inherent noise. This directly sets up the motivation for NRDE, which is designed to solve this noise-sensitivity issue.
>
> **Limitations:**
>
> We sincerely thank the reviewer for the constructive suggestion regarding limitations. We have included a detailed discussion of limitations in our revised paper. In **Appendix H**, we discuss more failure cases related to our assumptions. For instance, **Appendix H.1** presents synthetic experiments explicitly violating this assumption; as expected, NRDE's performance declines as the variance difference decreases. **Appendix H.4** analysiss the failure cases in real datasets and its relation to the variance difference. As for synthetic noise injection, the motivation and main focus of our evaluation is not related to it and we have provided experimental results under more realistic noise conditions.
>
> **We are looking forward to your feedback, and please do not hesitate to let us know if there are any concerns or questions still not properly addressed.**

---

> > ### Author Rebuttal · Reviewer_97ko · 2026-04-04
> >
> > Thank you for the detailed rebuttal and additional clarifications. The rebuttal offers useful additional analysis and experiments that partially address my concerns, especially on the empirical side. However, the core variance assumption remains strong and may limit applicability in more complex real-world scenarios. I have therefore slightly increased my score.

---

> > > ### Author Response · Authors · 2026-04-04
> > >
> > > Dear Reviewer 97ko,
> > >
> > > Thank you very much for your continued engagement and for carefully considering our rebuttal. We sincerely appreciate your time and thoughtful feedback throughout the review process, and we are encouraged that the additional analysis and experiments helped address part of your concerns.
> > >
> > > We fully agree that the variance assumption may not hold universally, and there may indeed exist practical scenarios in which NRDE is less effective. We would like to respond from two perspectives:
> > >
> > > **1. Applicability of NRDE to complex real-world scenarios**
> > >
> > > We respectfully clarify that our empirical evaluation already covers a broad range of complex and realistic settings.
> > >
> > > First, we evaluate NRDE on **47 highly diverse datasets spanning multiple domains** in ADBench, which already reflects substantial heterogeneity in real-world data distributions.
> > >
> > > Second, our experiments include several challenging realistic settings beyond standard anomaly detection, including **AD with anomaly contamination, AD with artificially injected noise, and outlier detection (transductive AD)**. These settings cover most widely used anomaly detection scenarios in the literature and introduce substantial practical complexity.
> > >
> > > Taken together, the strong performance of NRDE across diverse datasets and multiple realistic challenging settings provides empirical evidence that the method can generalize to complex real-world scenarios.
> > >
> > > **2. Rationality of our variance assumption**
> > >
> > > We agree that this assumption is an important theoretical point and appreciate the opportunity to clarify it further.
> > >
> > > Unlike prior approaches that often rely on external structural information, metadata, or labeled data to identify informative patterns, our method operates solely on the observed signal itself.
> > >
> > > The intuition behind our assumption is that a meaningful signal typically contains structured and correlated patterns, whereas noise is more often generated by random and uncorrelated processes. As a result, the signal generally exhibits more concentrated energy and stronger variability, which is naturally reflected in higher variance.
> > >
> > > This assumption is also widely adopted in prior machine learning and statistical methods. For example, **PCA and KPCA** assume that the most informative data structure lies along directions with higher variance, while low-variance components are often treated as noise during denoising and dimensionality reduction. Similarly, in signal processing, many denoising filters suppress low-power (low-variance) frequency components that are assumed to correspond to noise while preserving dominant high-power components as signal.
> > >
> > > Importantly, although such assumptions are not universally valid, these methods have demonstrated strong practical effectiveness across many real-world applications. In the same spirit, while NRDE relies on a related assumption, both the empirical results and additional experiments support its practical utility in diverse complex scenarios.
> > >
> > > A brief discussion of this intuition is currently included around Line 175 of the paper. In the revised version, we will further expand this discussion and explicitly include a more detailed analysis of the variance assumption as well as potential failure cases of NRDE.
> > >
> > > Thank you again for your valuable feedback.
> > >
> > > ---
> > >
> > > ## Update: Clarification on Novelty Concerns
> > >
> > > We note that Reviewer 97ko has now submitted their final justification. We appreciate Reviewer 97ko's acknowledgment of our solid empirical performance and the score increase. However, the novelty concern was **not raised in the Rebuttal Acknowledgement**, depriving us of the chance to respond. We now clarify:
> > >
> > > 1. **Novel problem formulation.** We formulate Noise-Robust Anomaly Detection, which detects changes in data sources rather than all sources, targeting noise-induced false positives — a challenge largely overlooked in existing literature.
> > >
> > > 2. **New perspective on data generation.** We explicitly separate noise sources from data sources to obtain pure data for scoring. Existing density-based methods, including flows, cannot perform such separation, as they model overall density without distinguishing signal from noise.
> > >
> > > 3. **General framework.** NRDE does not necessarily depend on flows. As shown in **Table 13** of our paper, applying k-NN and KDE to obtained pure data remains effective, confirming our framework's generality and flexibility.
> > >
> > > 4. **Novel use of Jacobian regularization.** Although Jacobian regularization exists in prior work, our motivation — enlarging the variance gap between data and noise sources — is entirely new, also the regularizer is novel and purposefully designed for this goal.
> > >
> > > In summary, the novelty lies in the new formulation, new perspective, general framework design, and novel motivation behind each technical choice.

---

### Decision · Program_Chairs · 2026-04-30

**Decision:**

Accept (regular)

**Comment:**

The authors address the ongoing challenge of anomaly detection in tabular data. A topic with a broad interest in the ML community. The proposed method is a density estimator designed to be robust to noise, using a Jacobian-regularized normalizing flow. Most reviewers were positive about the paper and found the empirical results strong. Specifically, I like that the authors include an evaluation of performance on a contaminated training set, which is a realistic setting. The method shines in such a setting. During the rebuttal, concerns were raised about the hyperparamter tuning of the method, but I believe the authors properly addressed them. Remaining concerns involve presentation of the paper, which I believe are addressable. Overall, I see this as a valid contribution with an idea that could find interest in the ML community. I recommend acceptance of the paper.